# On the Convergence of Encoder-only Shallow Transformers

**Yongtao Wu**
LIONS, EPFL
yongtao.wu@epfl.ch

**Fanghui Liu**[*]
University of Warwick
fanghui.liu@warwick.ac.uk

**Grigorios G Chrysos**[*]
LIONS, EPFL
University of Wisconsin-Madison
chrysos@wisc.edu

**Volkan Cevher**
LIONS, EPFL
volkan.cevher@epfl.ch

## Abstract

In this paper, we aim to build the global convergence theory of encoder-only shallow Transformers under a *realistic* setting from the perspective of architectures, initialization, and scaling under a finite width regime. The difficulty lies in how to tackle the softmax in self-attention mechanism, the core ingredient of Transformer. In particular, we diagnose the scaling scheme, carefully tackle the input/output of softmax, and prove that quadratic overparameterization is sufficient for global convergence of our shallow Transformers under commonly-used He/LeCun initialization in practice. Besides, neural tangent kernel (NTK) based analysis is also given, which facilitates a comprehensive comparison. Our theory demonstrates the *separation* on the importance of different scaling schemes and initialization. We believe our results can pave the way for a better understanding of modern Transformers, particularly on training dynamics.

## 1 Introduction

Transformers [Vaswani et al., 2017] have demonstrated unparalleled success in influential applications [Devlin et al., 2019, Brown et al., 2020, Wang et al., 2018, Dosovitskiy et al., 2021, Liu et al., 2022b]. A fundamental theoretical topic concerns the global convergence, i.e., the training dynamics of Transformers, which would be helpful for further analysis, e.g., in-context learning [von Oswald et al., 2022, Akyürek et al., 2023], generalization [Li et al., 2023]. In fact, even within a simplified Transformer framework under certain specific regimes, the global convergence guarantees still remain an elusive challenge.

To theoretically understand this, let us first recall the exact format of the self-attention mechanism, the core ingredient of the Transformer. Given the input $\boldsymbol{X} \in \mathbb{R}^{d_s \times d}$ ($d_s$ is the number of tokens and $d$ is the feature dimension of each token), a self-attention mechanism is defined as:

$$\text{Self-attention}(\boldsymbol{X}) \triangleq \sigma_s \left( \tau_0 (\boldsymbol{X} \boldsymbol{W}_Q^\top) \left( \boldsymbol{X} \boldsymbol{W}_K^\top \right)^\top \right) \left( \boldsymbol{X} \boldsymbol{W}_V^\top \right) = \sigma_s \left( \tau_0 \boldsymbol{X} \boldsymbol{W}_Q^\top \boldsymbol{W}_K \boldsymbol{X}^\top \right) \left( \boldsymbol{X} \boldsymbol{W}_V^\top \right) ,$$

where $\sigma_s$ is the row-wise softmax function, $\tau_0 \in \mathbb{R}^+$ is the scaling factor, and the learnable weights are $\boldsymbol{W}_Q, \boldsymbol{W}_K, \boldsymbol{W}_V \in \mathbb{R}^{d_m \times d}$ with the width $d_m$. Given $\boldsymbol{X}$, the input of softmax depends on $\tau_0 \boldsymbol{W}_Q^\top \boldsymbol{W}_K$, including the scaling factor $\tau_0$ and initialization schemes for learnable parameters, and thus determines the output of softmax and then affects the performance of Transformers in both

---

[*]Work done at LIONS, EPFL.

theory and practice. There are several scaling schemes in previous literature. For instance, given $\boldsymbol{W}_Q$ and $\boldsymbol{W}_K$ initialized by standard Gaussian, the scaling factor $\tau_0$ is chosen by

- $\tau_0 = d_m^{-1/2}$ in the original Transformer [Vaswani et al., 2017]: each element in $\tau_0 \boldsymbol{W}_Q^\top \boldsymbol{W}_K$ is a random variable with mean 0 and variance 1. This scaling avoids the blow-up of value inside softmax as $d_m$ increases [Hron et al., 2020].
- $\tau_0 = d_m^{-1}$: This scaling stems from the neural tangent kernel (NTK) analysis [Jacot et al., 2018], a commonly used technical tool for convergence analysis of fully-connected (or convolutional) networks under an infinite-width setting $d_m \to \infty$. However, for Transformer, if one uses this scaling under the infinite-width setting, then by the law of large numbers, we have $\lim_{d_m \to \infty} \tau_0 [\boldsymbol{W}_Q^\top \boldsymbol{W}_K]^{(ij)} = 0$. As a result, the input of softmax is zero and the softmax degenerates to a pooling layer. That means, the non-linearity is missing, which motivates researchers to carefully rethink this setting.

For instance, under the $\tau_0 = d_m^{-1}$ setting, Yang [2020] use the same query and key matrices to prevent the softmax from degenerating into a pooling layer. Besides, to avoid the analytic difficulty of softmax due to the fact that each element of the output depends on all inputs, Hron et al. [2020] substitute softmax with ReLU under the $\tau_0 = d_m^{-1/2}$ and infinite width setting for simplicity.

Clearly, there exists a gap between theoretical analysis and practical architectures on the use of softmax, and accordingly, this leads to the following open question:

*How can we ensure the global convergence of Transformers under a realistic setting?*

The primary contribution of this work is to establish the convergence theory of shallow Transformer under a *realistic* setting. Despite its shallow and encoder-only architecture, our Transformer model captures all the fundamental components found on typical Transformers, including the self-attention mechanism with the softmax activation function, one feedforward ReLU layer, one average pooling layer, and a linear output layer, *cf.* Eq. (1.2). We adopt the $\tau_0 = d_m^{-1/2}$ scaling under the finite-width setting and compare the results of LeCun/He initializations, which are commonly used in practical applications. Besides, the convergence result under the $\tau_0 = d_m^{-1}$ setting (as well as the NTK based analysis) is also studied, which facilitates a comprehensive comparison. Our theoretical results demonstrate *notable separations* among scaling settings, initializations, and architectures as below:

- *Scaling:* The global convergence can be achieved under both $\tau_0 = d_m^{-1/2}$ and $\tau_0 = d_m^{-1}$. Nevertheless, as suggested by our theory: for a small $d_m$, there is no significant difference for these two scaling schemes on the convergence; but for a large enough $d_m$, the $\tau_0 = d_m^{-1/2}$ scaling admits a faster convergence rate of Transformers than that with $\tau_0 = d_m^{-1}$. Interestingly, under this $\tau_0 = d_m^{-1}$ setting, our theory also demonstrates the *separation* on the convergence result, depending on whether the input is formed along sequence dimension ($d = 1$) or embedding dimension ($d_s = 1$).
- *Initialization:* Under LeCun and He initialization, our shallow Transformer admits a faster convergence rate than the NTK initialization. This could be an explanation for the seldom usage of NTK initialization for Transformer training in practice.
- *Architecture:* Quadratic over-parameterization is enough to ensure the global convergence of our shallow Transformer. As a comparison, if the self-attention mechanism is substituted by a feed-forward ReLU layer, our shallow Transformer is close to a three-layer fully-connected ReLU neural networks to some extent, requiring cubic over-parameterization for global convergence.

We firmly believe that our theoretical analysis takes a significant step towards unraveling the mysteries behind Transformers from the perspective of global convergence. We hope that our analytical framework and insights on various initialization and scaling techniques would be helpful in training modern, large-scale Transformer-based models [Radford et al., 2018, Brown et al., 2020].

## 2   Related work

**Self-attention, Transformer:** Regarding training dynamics, Snell et al. [2021] explain why single-head attention focuses on salient words by analyzing the evolution throughout training. Hron et al. [2020] show that the output of Transformer converges to Gaussian process kernel and provide the

NTK formulation of Transformer. Recently, Li et al. [2023] provide sample complexity of shallow Transformer to study its generalization property under a good initialization from pretrained model. The separation between the Transformer and CNN is recently explored. Jelassi et al. [2022] provably demonstrate that Vision Transformer (ViT) has the ability to learn spatial structure without additional inductive bias such as the spatial locality in CNN. Chen et al. [2022] study the loss landscape of ViT and find that ViT converges at sharper local minima than ResNet. Park and Kim [2022] show that ViT is a low-pass filter while CNN is a high-pass filter, thus, these two models can be complementary.

**NTK, lazy training, Hessian:** The NTK was introduced by Jacot et al. [2018] to connect the infinite-width neural network trained by gradient descent and the kernel regression. The roles of such kernel include analysis of the training dynamics of the neural network in the over-parameterization regime [Allen-Zhu et al., 2019a, Chizat et al., 2019, Du et al., 2019a,b, Zou et al., 2020]. The global convergence, generalization bound, and memorization capacity largely depend on the minimum eigenvalue of the NTK [Cao and Gu, 2019, Zhu et al., 2022, Nguyen et al., 2021, Bombari et al., 2022]. Even though the NTK is extended from FCNN to several typical networks including Transformer [Tirer et al., 2020, Huang et al., 2020, Arora et al., 2019b, Alemohammad et al., 2021, Nguyen and Mondelli, 2020], it has not been used to analyze the global convergence of Transformer. On the other hand, the stability of the tangent kernel during training is required when connecting to kernel regression, but such stability can not be explained by the phenomenon of lazy training [Chizat et al., 2019], which indicates a small change of the parameters from initialization. The hessian spectral bound is the main reason for the stability of kernel, as mentioned in Liu et al. [2020].

**Over-parameterization for convergence analysis:** Due to over-parameterization, neural networks (NNs) can fit arbitrary labels with zero training loss when trained with (stochastic) gradient descent (SGD), both theoretically Li and Liang [2018], Du et al. [2019b] and empirically [Zhang et al., 2017]. This leads to an interesting question in theory: *how much overparameterization is enough to ensure global convergence of NNs?* A common recipe for the proof of global convergence relies on the variant of Polyak-Lojasiewicz condition [Polyak, 1963, Liu et al., 2022a], NTK [Du et al., 2019b,a, Zou and Gu, 2019, Allen-Zhu et al., 2019a], or the minimum eigenvalue of the gram matrix [Nguyen, 2021, Bombari et al., 2022]. In Appendix B.3, we provide a comprehensive overview of a recent line of work that improves the over-parameterization condition for ensuring the convergence of NNs. However, the over-parameterization condition for Transformer to achieve global convergence remains elusive from existing literature and we make an initial step towards this question.

# 3 Problem setting

This section includes the problem setting with notations and model formulation of the shallow Transformer that is studied in this paper.

## 3.1 Notation

Vectors (matrices) are symbolized by lowercase (uppercase) boldface letters, e.g., $\boldsymbol{w}$, $\boldsymbol{W}$. We use $\|\cdot\|_{\mathrm{F}}$ and $\|\cdot\|_2$ to represent the Frobenius norm and the spectral norm of a matrix, respectively. The Euclidean norm of a vector is symbolized by $\|\cdot\|_2$. The superscript with brackets is used to represent the element of a vector/matrix, e.g., $w^{(i)}$ is the $i^{\mathrm{th}}$ element of $\boldsymbol{w}$. The superscript without brackets symbolizes the parameters at different training step, e.g., $\boldsymbol{\theta}^t$. We denote by $[N] = \{1, \ldots, N\}$ for short. We use $\sigma_{\min}(\cdot)$ and $\lambda_{\min}(\cdot)$ to represent the minimum singular value and minimum eigenvalue of a matrix. The NTK matrix and hessian matrix of the network are denoted by $\boldsymbol{K}$ and $\boldsymbol{H}$, respectively. The order notation, e.g., $\widetilde{\mathcal{O}}$, $\widetilde{\Omega}$, omits the logarithmic factor. More detailed notation can be found in Table 2 of the appendix.

Let $X \subseteq \mathbb{R}^{d_s \times d}$ be a compact metric space and $Y \subseteq \mathbb{R}$, where $d$ is the dimension of each token, $d_s$ is the total sequence length of the input. The training set $\{(\boldsymbol{X}_n, y_n)\}_{n=1}^N$ is assumed to be iid sampled from an unknown probability measure on $X \times Y$. In this paper, we focus the regression task by employing the squared loss. The goal of our regression task is to find a hypothesis, i.e., a Transformer $f : X \to Y$ in our work, such that $f(\boldsymbol{X}; \boldsymbol{\theta})$ parameterized by $\boldsymbol{\theta}$ is a good approximation of the label $y \in Y$ corresponding to a new sample $\boldsymbol{X} \in X$. We use a vector $\boldsymbol{\theta}$ to denote the collection of all learnable parameters.

Table 1: Common initialization methods with their variances of Gaussian distribution and scaling factors. The choice of $\tau_1 = d_m^{-1/2}$ is based on standard NTK initialization on prior literature [Du et al., 2019b].

| Init. | $\eta_O$ | $\eta_V$ | $\eta_Q$ | $\eta_K$ | $\tau_1$ |
|-------|----------|----------|----------|----------|----------|
| LeCun | $d_m^{-1}$ | $d^{-1}$ | $d^{-1}$ | $d^{-1}$ | $1$ |
| He | $2d_m^{-1}$ | $2d^{-1}$ | $2d^{-1}$ | $2d^{-1}$ | $1$ |
| NTK | $1$ | $1$ | $1$ | $1$ | $d_m^{-1/2}$ |

---

**Algorithm 1:** Gradient descent training

**Input:** data $(\boldsymbol{X}_n, y_n)_{n=1}^N$, step size $\gamma$.
Initialize weights as follows:
$\quad \boldsymbol{\theta}^0 := \{\boldsymbol{W}_Q^0, \boldsymbol{W}_K^0, \boldsymbol{W}_V^0, \boldsymbol{W}_O^0\}$.
**for** $t = 0$ **to** $t' - 1$ **do**
$\quad \boldsymbol{W}_Q^{t+1} = \boldsymbol{W}_Q^t - \gamma \cdot \nabla_{\boldsymbol{W}_Q} \ell(\boldsymbol{\theta}^t),$
$\quad \boldsymbol{W}_K^{t+1} = \boldsymbol{W}_K^t - \gamma \cdot \nabla_{\boldsymbol{W}_K} \ell(\boldsymbol{\theta}^t),$
$\quad \boldsymbol{W}_V^{t+1} = \boldsymbol{W}_V^t - \gamma \cdot \nabla_{\boldsymbol{W}_V} \ell(\boldsymbol{\theta}^t),$
$\quad \boldsymbol{W}_O^{t+1} = \boldsymbol{W}_O^t - \gamma \cdot \nabla_{\boldsymbol{W}_O} \ell(\boldsymbol{\theta}^t).$
**end for**
**Output:** the model based on $\boldsymbol{\theta}^{t'}$.

---

## 3.2 Model formulation of shallow Transformer

Throughout this work, we consider the encoder of Transformer, which can be applied to both regression and classification tasks [Yüksel et al., 2019, Dosovitskiy et al., 2021]. Given an input $\boldsymbol{X} \in \mathbb{R}^{d_s \times d}$, the model is defined as below:

$$\boldsymbol{A}_1 = \text{Self-attention}(\boldsymbol{X}) \triangleq \sigma_s \left( \tau_0 (\boldsymbol{X} \boldsymbol{W}_Q^\top) \left( \boldsymbol{X} \boldsymbol{W}_K^\top \right)^\top \right) \left( \boldsymbol{X} \boldsymbol{W}_V^\top \right), \tag{1.1}$$

$$\boldsymbol{A}_2 = \tau_1 \sigma_r (\boldsymbol{A}_1 \boldsymbol{W}_H), \qquad \boldsymbol{a}_3 = \varphi(\boldsymbol{A}_2), \qquad f(\boldsymbol{X}; \boldsymbol{\theta}) = \boldsymbol{a}_3^\top \boldsymbol{w}_O, \tag{1.2}$$

where the output is $f(\boldsymbol{X}; \boldsymbol{\theta}) \in \mathbb{R}$, $\tau_0$ and $\tau_1$ are two scaling factors. The ingredients of a Transformer with width $d_m$ are defined as follows:

- A *self-attention* mechanism (Eq. (1.1)): $\sigma_s$ is the row-wise softmax function; the learnable parameters are $\boldsymbol{W}_Q, \boldsymbol{W}_K, \boldsymbol{W}_V \in \mathbb{R}^{d_m \times d}$. We employ Gaussian initialization $\boldsymbol{W}_Q^{(ij)} \sim \mathcal{N}(0, \eta_Q)$, $\boldsymbol{W}_K^{(ij)} \sim \mathcal{N}(0, \eta_K)$, $\boldsymbol{W}_V^{(ij)} \sim \mathcal{N}(0, \eta_V)$ with $i \in [d_m]$ and $j \in [d]$. Refer to Table 1 for typical initialization examples.

- A *feed-forward ReLU* layer (in Eq. (1.2)): $\sigma_r$ is the ReLU activation function; the learnable parameter is $\boldsymbol{W}_H \in \mathbb{R}^{d_m \times d_m}$. Following Yang et al. [2022], we combine $\boldsymbol{W}_V$ and $\boldsymbol{W}_H$ together (by setting $\boldsymbol{W}_H = \boldsymbol{I}$) for ease of the analysis. Note that it does not mean its training dynamics are the same as the joint-training of these two adjacent matrices.

- An *average pooling* layer (in Eq. (1.2)): $\varphi$ indicates the column-wise average pooling. Note that the average pooling layer is applied along the sequence length dimension to ensure the final output is a scalar, which is commonly used in practical Vision Transformer or theoretical analysis [Dosovitskiy et al., 2021, Yang, 2020].

- An *output* layer (in Eq. (1.2)) with learnable parameter $\boldsymbol{w}_O \in \mathbb{R}^{d_m}$, initialized by $\boldsymbol{w}_O^{(i)} \sim \mathcal{N}(0, \eta_O)$.

**Remarks:** Proper initialization and scaling are required to ensure the convergence and learnability, as seen in previous work [Jacot et al., 2018, Tirer et al., 2022, Lee et al., 2019]. For our convergence analysis, we consider standard Gaussian initialization with different variances and different scaling factor that includes three typical initialization schemes in practice. In Table 1, we detail the formula of LeCun initialization, He initialization, and NTK initialization.

Given $N$ input samples $\{\boldsymbol{X}_n\}_{n=1}^N$, the corresponding ground truth label, the final output of network, and the output of the last hidden layer, are denoted by:

$$\boldsymbol{y} \triangleq \{y_n\}_{n=1}^N \in \mathbb{R}^N, \quad \boldsymbol{f}(\boldsymbol{\theta}) \triangleq \{f(\boldsymbol{X}_n; \boldsymbol{\theta})\}_{n=1}^N \in \mathbb{R}^N, \quad \boldsymbol{F}_{\text{pre}}(\boldsymbol{\theta}) \triangleq \{\boldsymbol{a}_3(\boldsymbol{X}_n; \boldsymbol{\theta})\}_{n=1}^N \in \mathbb{R}^{N \times d_m}.$$

We consider standard gradient descent (GD) training of Transformer, as illustrated in Algorithm 1. Here the squared loss is expressed as $\ell(\boldsymbol{\theta}) = \frac{1}{2} \|\boldsymbol{f}(\boldsymbol{\theta}) - \boldsymbol{y}\|_2^2$.

## 4 Main results

In this section, we study the convergence guarantee of Transformer training by GD under the squared loss. Firstly, we provide a general analytical framework in Section 4.1 covering different initialization

schemes, where we identify the condition for achieving global convergence. Next, we validate these conditions for several practical initialization schemes under $\tau_0 = d_m^{-1/2}$ in Section 4.2 and $\tau_0 = d_m^{-1}$ in Section 4.3, respectively. We include NTK-based results for self-completeness. Discussion on the convergence results with different scalings, initializations, and architectures is present in Section 4.4.

## 4.1 General framework for convergence analysis

Before presenting our result, we introduce a basic assumption.

**Assumption 1.** *The input data is bounded* $\|\boldsymbol{X}\|_{\mathrm{F}} \leq \sqrt{d_s}C_x$ *with some positive constant* $C_x$.

**Remark:** This assumption is standard as we can always scale the input. The embedding of token is usually assumed to have a unit norm [Li et al., 2023], which is unrelated to $d$.

Now we are ready to present our proposition, with the proof deferred to Appendix C.2. Notice that our proposition holds with high probability under some conditions, which depend on certain initialization schemes and scaling factors. Since our proposition is devoted to a unifying analysis framework under various initialization schemes, we do not include specific probabilities here. The validation of the required conditions and probability is deferred to Sections 4.2 and 4.3, respectively.

**Proposition 1.** *Consider the Transformer with* $d_m \geq N$. *Let* $C_Q, C_K, C_V, C_O$ *be some positive constants, and define the following quantities at initialization for simplification:*

- *The norm of the parameters:*

$$\bar{\lambda}_Q \triangleq \left\|\boldsymbol{W}_Q^0\right\|_2 + C_Q, \ \bar{\lambda}_K \triangleq \left\|\boldsymbol{W}_K^0\right\|_2 + C_K, \ \bar{\lambda}_V \triangleq \left\|\boldsymbol{W}_V^0\right\|_2 + C_V, \ \bar{\lambda}_O \triangleq \left\|\boldsymbol{w}_O^0\right\|_2 + C_O.$$

- *Two auxiliary terms:* $\rho \triangleq N^{1/2}d_s^{3/2}\tau_1 C_x, \quad z \triangleq \bar{\lambda}_O^2 \left(1 + 4\tau_0^2 C_x^4 d_s^2 \bar{\lambda}_V^2 \left(\bar{\lambda}_Q^2 + \bar{\lambda}_K^2\right)\right).$

*Under Assumption 1, we additionally assume that the minimum singular value of* $\boldsymbol{F}_{\mathrm{pre}}^0$, *i.e.,* $\alpha \triangleq \sigma_{\min}\left(\boldsymbol{F}_{\mathrm{pre}}^0\right)$ *satisfies the following condition at initialization:*

$$\alpha^2 \geq 8\rho \max\left(\bar{\lambda}_V C_O^{-1}, \bar{\lambda}_O C_V^{-1}, 2\tau_0 C_x^2 d_s \bar{\lambda}_K \bar{\lambda}_V \bar{\lambda}_O C_Q^{-1}, 2\tau_0 C_x^2 d_s \bar{\lambda}_Q \bar{\lambda}_V \bar{\lambda}_O C_K^{-1}\right) \sqrt{2\ell(\boldsymbol{\theta}^0)}, \quad (2)$$

$$\alpha^3 \geq 32\rho^2 z \sqrt{2\ell(\boldsymbol{\theta}^0)}/\bar{\lambda}_O. \quad (3)$$

*If the step size satisfies* $\gamma \leq 1/C$ *with a constant $C$ depending on* $(\bar{\lambda}_Q, \bar{\lambda}_K, \bar{\lambda}_V, \bar{\lambda}_O, \ell(\boldsymbol{\theta}^0), \rho, \tau_0)$, *then GD converges to a global minimum as follows:*

$$\ell(\boldsymbol{\theta}^t) \leq \left(1 - \gamma\frac{\alpha^2}{2}\right)^t \ell(\boldsymbol{\theta}^0), \quad \forall t \geq 0. \quad (4)$$

**Remark:** The parameter $\alpha$ in Eqs. (2) and (3) controls the convergence rate of global convergence, and the condition will be verified in the next subsection. The step-size $\gamma$ is inversely proportional to $N^{1/2}$ due to the construction of $C$ in Appendix C.2.

**Proof sketch:** The main idea of our convergence analysis is based on the variant of Polyak-Lojasiewicz (PL) inequality [Polyak, 1963, Nguyen, 2021], i.e., $\|\nabla\ell(\boldsymbol{\theta})\|_2^2 \geq 2\lambda_{\min}(\boldsymbol{K})\ell(\boldsymbol{\theta}) \geq 2\lambda_{\min}(\boldsymbol{F}_{\mathrm{pre}}\boldsymbol{F}_{\mathrm{pre}}^\top)\ell(\boldsymbol{\theta})$. Thus, if the minimum singular value of $\boldsymbol{F}_{\mathrm{pre}}$ is strictly greater than 0, then minimizing the gradient on the LHS will drive the loss to zero. To this end, Proposition. 1 can be split into two parts. First, by induction, at every time step, each parameter in the Transformer can be bounded w.h.p; the minimum singular value of $\boldsymbol{F}_{\mathrm{pre}}$ is bounded away for some positive quality at the initialization point. Secondly, we prove that the Lipschitzness of the network gradient, which means the loss function is almost smooth. Combining the above two results, the global convergence can be achieved. Furthermore, based on different initialization and scaling schemes, we are able to validate Eqs. (2) and (3) via the spectral norm estimation of $\bar{\lambda}_Q, \bar{\lambda}_K, \bar{\lambda}_V, \bar{\lambda}_O$ and a positive lower bound for $\boldsymbol{F}_{\mathrm{pre}}$ in the following section.

## 4.2 LeCun and He initialization under the $d_m^{-1/2}$ setting

Here we aim to show that, under the LeCun/He initialization with the setting of $d_m^{-1/2}$, the conditions in Eqs. (2) and (3) will be satisfied with high probability and scaling schemes in Table 1 and hence

lead to global convergence. To derive our result, we need the following assumptions on the input data regarding the rank and dissimilarity of data.

**Assumption 2.** *We assume that the input data $\boldsymbol{X}$ has full row rank.*

**Remark:** This assumption requires that each row $\boldsymbol{X}^{(i,:)}$ is linearly independent for any $i \in [d_s]$, which is fair and attainable in practice. For example, for language tasks, even though there might be some repeated token in $\boldsymbol{X}$, each row in $\boldsymbol{X}$ can be uncorrelated when added with positional embedding. Similarly, in image tasks with Visual Transformer, the raw image is grouped by patch and mapped via a linear layer to construct $\boldsymbol{X}$, thus each row in $\boldsymbol{X}$ can be uncorrelated.

**Assumption 3.** *For any data pair $(\boldsymbol{X}_n, \boldsymbol{X}_{n'})$, with $n \neq n'$ and $n, n' \in [N]$, then we assume that $\mathbb{P}\left(\left|\langle \boldsymbol{X}_n^\top \boldsymbol{X}_n, \boldsymbol{X}_{n'}^\top \boldsymbol{X}_{n'}\rangle\right| \geq t\right) \leq \exp(-t^{\hat{c}})$ with some constant $\hat{c} > 0$.*

**Remark:** We discuss the rationale behind this assumption:

The idea behind Assumption 3 is that different data points admit a small similarity. To be specific, for two data points $\boldsymbol{X}_n$ and $\boldsymbol{X}_{n'}$ with $n \neq n'$, their inner product reflects the similarity of their respective (empirical) covariance matrix. We expect that this similarity is small with a high probability. The spirit of this assumption also exists in previous literature, e.g., Nguyen et al. [2021]. The constant $\hat{c}$ determines the decay of data dissimilarity. A larger $\hat{c}$ results in less separable data. Our assumption has no requirement on $\hat{c}$ such that $\hat{c}$ can be small enough, which allows for a general data distribution.

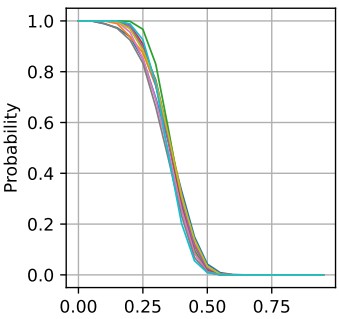

Figure 1: Validation of Asm. 3.

**Verification of assumption.** Here we experimentally validate this assumption under a standard language IMDB dataset [Maas et al., 2011]. We randomly sample 100 (normalized) sentences with embedding and plot the probability (frequency) of $\mathbb{P}\left(\left|\langle \boldsymbol{X}_n^\top \boldsymbol{X}_n, \boldsymbol{X}_{n'}^\top \boldsymbol{X}_{n'}\rangle\right| \geq t\right)$ as $t$ increases. We repeat it over 10 runs and plot the result in Figure 1. We can observe an exponential decay as $t$ increases, which implies that our assumption is fair. Besides, the validation of Assumption 3 on image data, e.g., MNIST by ViT, is deferred to Appendix E.2.

Now we are ready to present our main theorem under the $d_m^{-1/2}$ setting. The proof is deferred to Appendix C.3.

**Theorem 1.** *Under the setting of LeCun/He initialization in Table 1 and Assumptions 1 to 3, suppose $d_m \geq d$, and given $\tau_0 = d_m^{-1/2}$, $d_m = \tilde{\Omega}(N^3)$, then with probability at least $1 - 8e^{-d_m/2} - \delta - \exp\left(-\Omega((N-1)^{-\hat{c}}d_s^{-1})\right)$ for proper $\delta$, the GD training of Transformer converges to a global minimum with sufficiently small step size $\gamma$ as in Eq. (4).*

**Remark:** The probability relates to several randomness sources, e.g., data sampling, dissimilarity of data, and parameter initialization. The quantity $\exp\left(-\Omega((N-1)^{-\hat{c}}d_s^{-1})\right)$ can be small for a small enough $\hat{c}$ as discussed in Assumption 3. Further discussion on our result refers to Section 4.4 for details. Besides, our proof framework is also valid for the $\tau_0 = d_m^{-1}$ setting. We demonstrate that $d_m = \tilde{\Omega}(N^2)$ suffices to achieve global convergence, see Appendix C.3 for details.

**Proof sketch:** To check whether the conditions in Eqs. (2) and (3) hold, the key idea is to provide the lower bound of $\alpha$. Then we upper bound $\bar{\lambda}_Q, \bar{\lambda}_K, \bar{\lambda}_V, \bar{\lambda}_O$ based on concentration inequalities to upper bound the initial loss, one key step is to utilize Gershgorin circle theorem [Gershgorin, 1931] to provide a lower bound for $\alpha$. Lastly, we plug these bound into the condition Eqs. (2) and (3) in order to obtain the requirement for the width $d_m$.

## 4.3 NTK initialization under the $d_m^{-1}$ setting

The NTK theory, as a representative application of the $\tau_0 = d_m^{-1}$ setting, can be also used for analysis of training dynamics. We also include the NTK results in this section for self-completeness. In this section, we first derive the limiting NTK formulation of Transformer under the $d_m^{-1}$ scaling scheme, and then show the global convergence of Transformers.

**Lemma 1.** *Denote* $\boldsymbol{\Phi}^\star =: [\frac{1}{d_s}\boldsymbol{X}_1^\top \mathbf{1}_{d_s}, ..., \frac{1}{d_s}\boldsymbol{X}_N^\top \mathbf{1}_{d_s}]^\top \in \mathbb{R}^{N \times d}$, *then the limiting NTK matrix* $\boldsymbol{K} \in \mathbb{R}^{N \times N}$ *of Transformer under the NTK initialization with* $\tau_0 = d_m^{-1}$ *has the following form:*

$$\boldsymbol{K} = d_s^2 \mathbb{E}_{\boldsymbol{w} \sim \mathcal{N}(\boldsymbol{0}, \boldsymbol{I})} \left( \sigma_r \left( \boldsymbol{\Phi}^\star \boldsymbol{w} \right) \sigma_r \left( \boldsymbol{\Phi}^\star \boldsymbol{w} \right)^\top \right) + d_s^2 \mathbb{E}_{\boldsymbol{w} \sim \mathcal{N}(\boldsymbol{0}, \boldsymbol{I})} \left( \dot{\sigma}_r \left( \boldsymbol{\Phi}^\star \boldsymbol{w} \right) \dot{\sigma}_r \left( \boldsymbol{\Phi}^\star \boldsymbol{w} \right)^\top \right) \left( \boldsymbol{\Phi}^\star \boldsymbol{\Phi}^{\star\top} \right).$$

**Remark:** The formulation of $\boldsymbol{\Phi}^\star$ implies that the self-attention layer degenerates as $\frac{1}{d_s}\mathbf{1}_{d_s \times d_s} \boldsymbol{X} \boldsymbol{W}_V^\top$, i.e., the *dimension missing* effect as mentioned before.

Now we are ready to present our convergence result under the $d_m^{-1}$ scaling with the proof deferred to Appendix C.4.

**Theorem 2.** *Under the setting of NTK initialization in Table 1 and Assumptions 1 to 3 , suppose* $d_m = \tilde{\Omega}(N)$, *then with probability at least* $1 - 8e^{-d_m/2} - \delta - \exp\left(-\Omega((N-1)^{-\hat{c}}d_s^{-1})\right)$, *the GD training of Transformer converges to a global minimum with sufficiently small* $\gamma$ *as in Eq. (4).*

**Proof sketch:** The overall idea is the same as the proof of the previous theorem, i.e., we need to provide the lower bound of $\alpha$. However, in this case, the limit for the output of softmax exists so that we can apply concentration inequality to lower bound the $\alpha$. Lastly, we plug these bound into the condition Eqs. (2) and (3) in order to obtain the requirement for the width $d_m$.

Besides, the stability of NTK during training allows us to build a connection on training dynamics between the Transformer (assuming a squared loss) and the kernel regression predictor. Next, in order to show that the NTK is stable during GD training, below we prove that the spectral norm of Hessian is controlled by the width.

**Theorem 3** (Hessian norm is controlled by the width). *Under Assumption 1 and scaling* $\tau_0 = d_m^{-1}$, *given any fixed* $R > 0$, *and any* $\boldsymbol{\theta}^t \in B(\boldsymbol{\theta}, R) := \{\boldsymbol{\theta} : \|\boldsymbol{\theta} - \boldsymbol{\theta}^0\|_2 \leq R\}$, $\boldsymbol{\theta}^0$ *as the weight at initialization, then with probability at least* $1 - 8e^{-d_m/2}$, *the Hessian spectral norm of Transformer satisfies:* $\|\boldsymbol{H}(\boldsymbol{\theta}^t)\|_2 \leq \mathcal{O}\left(d_m^{-1/2}\right)$.

**Remark:** By [Liu et al., 2020, Proposition 2.3], the small Hessian norm is a sufficient condition for small change of NTK. Thus, Theorem 3 can be an indicator for the stability of NTK. Besides, Theorem 3 supplements the result in [Park and Kim, 2022] which exhibits empirically a relationship between the Hessian norm and the width but lacks theoretical proof.

## 4.4 Discussion on convergence results

We compare the derived results under different scaling schemes, initializations, and architectures.

**Comparison of scaling schemes:** The global convergence can be achieved under both $\tau_0 = d_m^{-1/2}$ and $\tau_0 = d_m^{-1}$. Nevertheless, as suggested by our theory, for a small $d_m$, there is no significant difference between these two scaling schemes on the convergence; but for a large enough $d_m$, the $\tau_0 = d_m^{-1/2}$ scaling admits a faster convergence rate of Transformers than that of $\tau_0 = d_m^{-1}$ due to a more tight estimation of the lower bound of $\alpha$, see Appendix C.6 for details. The intuition behind the lower convergence rate under the setting of large width and $\tau_0 = d_m^{-1}$ is that the input of softmax is close to zero such that softmax roughly degenerates as a pooling layer, losing the ability to fit data. This can be also explained by Lemma 1 from the perspective of *dimension missing*: self-attention $(\boldsymbol{X})$ degenerates as $\frac{1}{d_s}\mathbf{1}_{d_s \times d_s} \boldsymbol{X} \boldsymbol{W}_V^\top$.

The result under the $\tau_0 = d_m^{-1}$ setting requires weaker over-parameterization than the $\tau_0 = d_m^{-1/2}$ setting. Nevertheless, we do not claim that $\tau_0 = d_m^{-1}$ is better than $\tau_0 = d_m^{-1/2}$. This is because, under the over-parameterization regime, the scaling $\tau_0 = d_m^{-1}$ makes the self-attention layer close to a pooling layer. This analysis loses the ability to capture the key characteristics of the self-attention mechanism in Transformers.

Note that the lower bound of the minimum eigenvalue is in the constant order, which is tight (since it matches the upper bound). Based on this, by studying the relationship between $d_m$ and $\lambda_0$, we can prove that quadratic (cubic) over-parameterization is required for $d_m^{-1}$ ($d_m^{-1/2}$) scaling. This quadratic over-parameterization requirement could be relaxed if a better relationship is given while it is still unclear and beyond our proof technique.

**Comparison on initializations:** Though our results achieve the same convergence rate under these initialization schemes, we can still show the *separation* on $\alpha$ that affects the convergence in Eq. (4) among these initialization schemes. To be specific, we verify that under LeCun and He initialization, the lower bound of $\alpha^2$ is tighter than that of NTK initialization, and hence admits faster convergence, see Appendix C.5 for further details. This can be an explanation of the seldom usage of NTK initialization in practice. Besides, the NTK initialization scheme allows for a larger step size than LeCun/He initialization for training. That means, if $\alpha$ is the same in these three initialization schemes, we usually choose a large step size under the NTK initialization scheme, see Appendix E.1.

**Comparison on architectures:** Note that the Transformer defined in Eq. (1.2) includes a self-attention layer, a feedforward ReLU layer, and an output layer. Our result proves that a cubic (quadratic) over-parameterization condition is required for $d_m^{-1/2}$ ($d_m^{-1}$) under LeCun initialization. As a comparison, a three-layer fully-connected ReLU network under LeCun initialization requires $d_m = \tilde{\Omega}(N^3)$ [Nguyen, 2021].

The aforementioned result holds for matrix input. Although not as frequent, some data inputs are naturally in vector form. Two ways to feed the input into Transformer are either formulating along sequence dimension ($d = 1$) or along embedding dimension ($d_s = 1$). The following result shows the separation under these two settings to understand when the Transformer works well or not regarding the input.

**Corollary 1** (Convergence of vector input). *Consider LeCun initialization with $\tau_0 = d_m^{-1}$ scaling, given vector input $\boldsymbol{x} \in \mathbb{R}^{\tilde{d}}$, if one feeds the input to Transformer by setting $d_s = 1, d = \tilde{d}$, then training with GD can converge to a global minimum. On the contrary, if one sets $d_s = \tilde{d}, d = 1$, the conditions in Eqs. (2) and (3) do not hold, the convergence can not be guaranteed by our theory.*

**Remark:** Such a result is motivated by considering least squares. Specifically, given input $\boldsymbol{X} \in \mathbb{R}^{N \times 1}$, then the NTK $\boldsymbol{X}\boldsymbol{X}^\top$ is a rank-one matrix. As a result, when the augmented matrix $[\boldsymbol{X}, \boldsymbol{y}]$ is not rank-one, then there is no solution to the linear system so that GD training can not converge to zero loss. The empirical validation can be found in Figure 3.

**Technical difficulty.** The technical difficulty of our analysis includes handling the softmax function and scaling schemes beyond NTK initialization. Previous convergence analysis, e.g., [Du et al., 2019b, Nguyen, 2021] cannot be applied to our setting because of the following issues. First, different from classical activation functions, e.g., ReLU, in softmax each element of the output depends on all input. To tackle the interplay between dimensions, we build the connection between the output of softmax and $\boldsymbol{X}\boldsymbol{X}^\top$ to disentangle the nonlinear softmax function. By doing so, a lower bound on the minimum singular value of $\boldsymbol{F}_{\text{pre}}$ in Proposition. 1 can be well controlled by $\boldsymbol{X}\boldsymbol{X}^\top$ and the output of softmax.

Regarding different initializations and scaling, previous NTK-based analysis is only valid under the $d_m^{-1}$ setting (the softmax degenerates to an all-one vector) but is inapplicable to the realistic $d_m^{-1/2}$ setting, as discussed in the introduction. To tackle this issue, we analyze the input/output of softmax under LeCun/He initialization and identify the optimization properties of the loss function for global convergence under the finite-width setting.

## 5 Experimental validations

Our experiments are organized as follows: In Section 5.1, we conduct experiments with the model Eq. (1.2) on synthetic data and study the training dynamics. Next, we show convergence results on ViT [Dosovitskiy et al., 2021] on the standard MNIST dataset in Section 5.2. Additional results and detail on the experimental setup are deferred to Appendix E.

### 5.1 Fitting synthetic data

In this section, we corroborate our theoretical findings on the synthetic data. We generate 100 data points where the input $\boldsymbol{X} \in \mathbb{R}^{10 \times 100}$ is sampled from standard Gaussian distribution. The corresponding label $y \in \mathbb{R}$ is generated from standard Gaussian distribution. Squared loss is selected as the criterion. We apply gradient descent on the shallow Transformer defined in Eq. (1.2) with LeCun initialization and $\tau_0 = d_m^{-1/2}$ for 400 epochs with a fixed step size $\gamma = 1$. We test different widths of the network including $d_m = \{10, 100, 1000, 4000\}$ and plot the training progress in

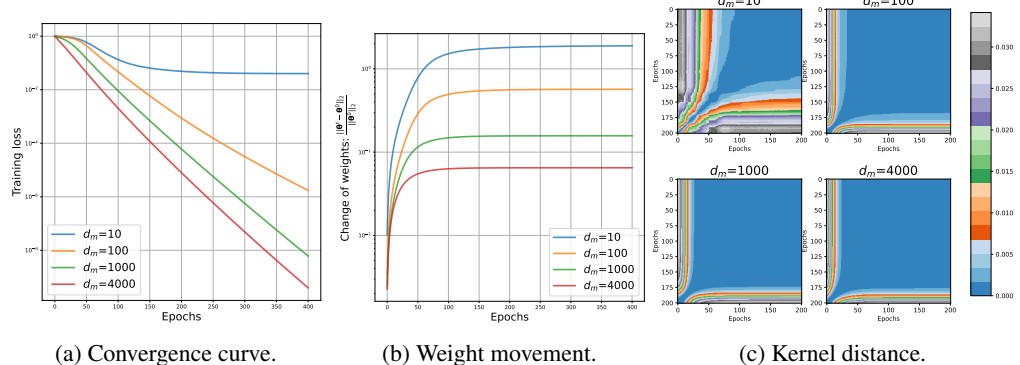

(a) Convergence curve.  (b) Weight movement.  (c) Kernel distance.

Figure 2: Visualization of the training process of Transformers with LeCun initialization and $\tau_0 = d_m^{-1/2}$ scaling on synthetic data. (a) Linear convergence. (b) Rate of change of the weights during training. Observe that the weights change very slowly after the $50^{\text{th}}$ epoch. (c) Evolution of the NTK during the training. The result mirrors the plot (b) and demonstrates how the kernel varies significantly at the beginning of the training and remains approximately constant later. As the width increases, the empirical NTK becomes more stable.

Figure 2. The result shows that the training can converge except for the case with sub-linear width, see the linear convergence rate in Figure 2a and the small movement of the weight in Figure 2b. For these cases, the weights do not change much after 50 epochs while the losses are still decreasing. In Figure 2c, we keep track of the evolution of NTK along each epoch. Specifically, the kernel distance is given by:

$$\text{Distance}\left(\boldsymbol{K}, \widetilde{\boldsymbol{K}}\right) = 1 - \frac{\text{Tr}\left(\boldsymbol{K}\widetilde{\boldsymbol{K}}^\top\right)}{\sqrt{\text{Tr}\left(\boldsymbol{K}\boldsymbol{K}^\top\right)}\sqrt{\text{Tr}\left(\widetilde{\boldsymbol{K}}\widetilde{\boldsymbol{K}}^\top\right)}},$$

which quantitatively compares two kernels by the relative rotation, as used in Fort et al. [2020]. Figure 2c shows that the kernel changes rapidly at the beginning of training while being approximately constant at later stages. The experiment with $\tau_0 = d_m^{-1}$, which is deferred to Appendix E.1, obtains similar results.

Additionally, the experiment with other initialization schemes is deferred to Appendix E.1, where we observe that NTK initialization indeed yields slower convergence than LeCun/He initialization, which is consistent with our theoretical finding.

Next, in order to verify corollary 1, we feed vector input $\boldsymbol{x}_n \in \mathbb{R}^{100}$ into Transformer by setting either $d_s = 1, d = 100$ or $d_s = 100, d = 1$ under LeCun initialization with $\tau_0 = d_m^{-1}$. Figure 3 shows that the training of Transformer with $d_s = 1, d = 100$ is similar to that of two-layer FCNN with the same width $d_m = 100$. However, the case of $d_s = 100, d = 1$ fails to converge, which is consistent with our theoretical finding.

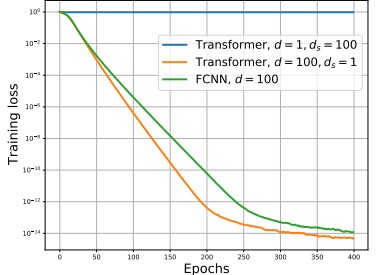

Figure 3: Convergence result on synthetic data with vector input.

## 5.2 Fitting real-world dataset

Beyond synthetic data, in this section, we examine the convergence performance of Vision Transformer (ViT) on classification task on MNIST dataset [LeCun et al., 1998], which includes 10 classes of images with size $28 \times 28$. We use a single layer and single head ViT. The dimension of $d$ is 64. We change the dimension of the query, key, and value from 16 to 1024 and 16384. The network is optimized with SGD with step size 0.1, and momentum 0.9 for 50 epochs. We repeat the experiment for three runs. We present the convergence results over 3 runs in Figure 4a, which shows that when the width is smaller, e.g., 16, both $\tau_0 = d_m^{-1}$ and $\tau_0 = d_m^{-1/2}$ scaling have similar

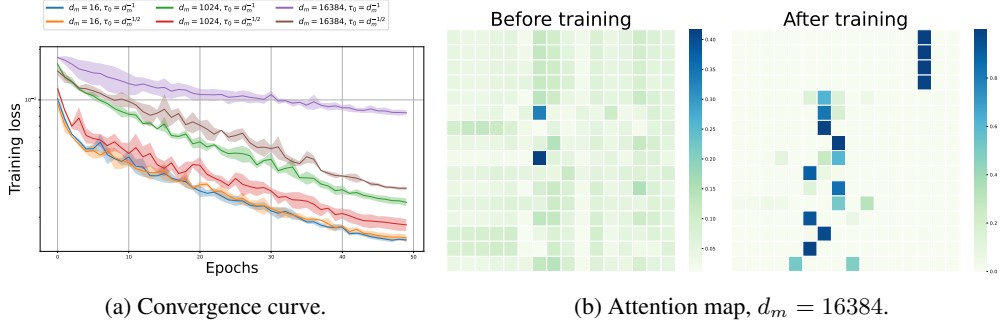

(a) Convergence curve.

(b) Attention map, $d_m = 16384$.

Figure 4: Convergence curve on MNIST dataset with different scaling schemes and different widths in (a). Visualization of attention map in (b).

convergence results. However, as the width increases to 1024 and 16384, the $\tau_0 = d_m^{-1}$ setting admits a slower rate than that of $\tau_0 = d_m^{-1/2}$, especially a extremely slow rate under $d_m = 16384$. This is consistent with our theoretical result on the *dimension missing* effect such that the $\tau_0 = d_m^{-1}$ setting makes Transformer difficult to fit data. Additionally, we visualize the attention map in Figure 4b, i.e., the output of softmax in the self-attention layer before training and after training under $\tau_0 = d_m^{-1/2}$ setting. We could see that the self-attention layer changes a lot during training.

## 6    Conclusion

We present a comprehensive analysis of the global convergence properties of shallow Transformer under various scaling and initialization schemes. Regarding scaling schemes, for a large width setting, the difference on convergence between $\tau_0 = d_m^{-1/2}$ and $\tau_0 = d_m^{-1}$ can be demonstrated both theoretically and empirically. Our theory is able to explain this in a *dimension missing* view. Regarding initialization schemes, our theory prefers to using LeCun and He initialization in Transformer training, which allows for a faster convergence rate than NTK initialization. Though our analysis is limited to shallow Transformers, we believe our framework can be extended to deep Transformers. We provide further discussion into this extension in Appendix C.7. Exploring the convergence properties of deep Transformers is indeed an intriguing avenue for future research.

## Acknowledgements

We thank the reviewers for their constructive feedback. We thank Zhenyu Zhu for the discussion and help in this work. This work has received funding from the Swiss National Science Foundation (SNSF) under grant number 200021_205011. This work was supported by Hasler Foundation Program: Hasler Responsible AI (project number 21043). Corresponding author: Fanghui Liu.

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

## Contents of the Appendix

The Appendix is organized as follows:

- In Appendix A, we summarize the symbols and notation used in this work.

- In Appendix B.1, we provide a theoretical overview for neural tangent kernel (NTK). The background in Sub-Exponential random variables is elaborated in Appendix B.2. More detailed related work on the convergence analysis of nerual networks can be found in Appendix B.3.

- The proofs for the convergence of Transformer are included in Appendix C.

- The derivations and the proofs for the NTK are further elaborated in Appendix D, including the formulation of NTK, minimum eigenvalue of NTK, and the relationship between the Hessian and the width.

- Further details on the experiments are developed in Appendix E.

- Limitations and societal impact of this work are discussed in Appendix F and Appendix G, respectively.

# A  Symbols and Notation

We include the core symbols and notation in Table 2 for facilitating the understanding of our work.

Table 2: Core symbols and notations used in this paper.

| Symbol | Dimension(s) | Definition |
|---|---|---|
| $\mathcal{N}(\mu, \sigma)$ | - | Gaussian distribution of mean $\mu$ and variance $\sigma$ |
| $\|\boldsymbol{w}\|_2$ | - | Euclidean norms of vectors $\boldsymbol{w}$ |
| $\|\boldsymbol{W}\|_2$ | - | Spectral norms of matrix $\boldsymbol{W}$ |
| $\|\boldsymbol{W}\|_{\mathrm{F}}$ | - | Frobenius norms of matrix $\boldsymbol{W}$ |
| $\|\boldsymbol{W}\|_*$ | - | Nuclear norms of matrix $\boldsymbol{W}$ |
| $\lambda_{\min}(\boldsymbol{W}), \lambda_{\max}(\boldsymbol{W})$ | - | Minimum and maximum eigenvalues of matrix $\boldsymbol{W}$ |
| $\sigma_{\min}(\boldsymbol{W}), \sigma_{\max}(\boldsymbol{W})$ | - | Minimum and Maximum singular values of matrix $\boldsymbol{W}$ |
| $\boldsymbol{w}^{(i)}$ | - | $i$-th element of vectors $\boldsymbol{w}$ |
| $\boldsymbol{W}^{(i,j)}$ | - | $(i,j)$-th element of matrix $\boldsymbol{W}$ |
| $\boldsymbol{W}^t$ | - | $\boldsymbol{W}$ at time step $t$ |
| $\circ$ | - | Hadamard product |
| $\sigma_s(\boldsymbol{W})$ | - | Row-wise Softmax activation for matrix $\boldsymbol{W}$ |
| $\sigma_r(\boldsymbol{W})$ | - | Element-wise ReLU activation for matrix $\boldsymbol{W}$ |
| $\mathbb{1}\{\text{event}\}$ | - | Indicator function for event |
| $N$ | - | Size of the dataset |
| $d_m$ | - | Width of intermediate layer |
| $d_s$ | - | Sequence length of the input |
| $d$ | - | Dimension of each token |
| $\eta_Q, \eta_K, \eta_V, \eta_O$ | - | Variance of Gaussian initialization of $\boldsymbol{W}_Q, \boldsymbol{W}_K, \boldsymbol{W}_V, \boldsymbol{w}_O$ |
| $\tau_0, \tau_1$ | - | Scaling factor |
| $\gamma$ | - | Step size |
| $\boldsymbol{X}_n$ | $\mathbb{R}^{d_s \times d}$ | The $n$-th data point |
| $y_n$ | $\mathbb{R}$ | The $n$-th target |
| $\mathcal{D}_X$ | - | Input data distribution |
| $\mathcal{D}_Y$ | - | Target data distribution |
| $\boldsymbol{\beta}_{i,n} := \sigma_s\left(\tau_0 \boldsymbol{X}_n^{(i,:)} \boldsymbol{W}_Q^\top \boldsymbol{W}_K \boldsymbol{X}_n^\top\right)^\top$ | $\mathbb{R}^{d_s}$ | The $i$-th row of the output of softmax of the $n$-th data point |
| $\boldsymbol{y} := [y_1, ..., y_N]^\top$ | $\mathbb{R}^N$ | Ground truth label of the data samples $\{\boldsymbol{X}_n\}_{n=1}^N$ |
| $\boldsymbol{f}(\boldsymbol{\theta}) := [f(\boldsymbol{X}_1; \boldsymbol{\theta}), ..., f(\boldsymbol{X}_N; \boldsymbol{\theta})]^\top$ | $\mathbb{R}^N$ | Network output given data samples $\{\boldsymbol{X}_n\}_{n=1}^N$ |
| $\boldsymbol{F}_{\text{pre}}(\boldsymbol{\theta}) := [\boldsymbol{a}_3(\boldsymbol{X}_1; \boldsymbol{\theta}), ..., \boldsymbol{a}_3(\boldsymbol{X}_N; \boldsymbol{\theta})]^\top$ | $\mathbb{R}^{N \times d_m}$ | Output of the last hidden layer given $\{\boldsymbol{X}_n\}_{n=1}^N$ |
| $f(\boldsymbol{\theta}) := f(\boldsymbol{X}; \boldsymbol{\theta})$ | $\mathbb{R}$ | Network output given data samples $\boldsymbol{X}$ |
| $\boldsymbol{f}_{\text{pre}} := \boldsymbol{a}_3(\boldsymbol{X}; \boldsymbol{\theta})$ | $\mathbb{R}^{d_m}$ | Output of the last hidden layer given $\boldsymbol{X}$ |
| $\boldsymbol{W}_Q, \boldsymbol{W}_K$ | $\mathbb{R}^{d_m \times d}, \mathbb{R}^{d_m \times d}$ | Learnable parameters |
| $\boldsymbol{W}_V, \boldsymbol{w}_O$ | $\mathbb{R}^{d_m \times d}, \mathbb{R}^{d_m}$ | Learnable parameters |
| $\bar{\lambda}_Q \triangleq \left\|\boldsymbol{W}_Q^0\right\|_2 + C_Q, \bar{\lambda}_K \triangleq \left\|\boldsymbol{W}_K^0\right\|_2 + C_K$ | $\mathbb{R}$ | Parameters norm |
| $\bar{\lambda}_V \triangleq \left\|\boldsymbol{W}_V^0\right\|_2 + C_V, \bar{\lambda}_O \triangleq \left\|\boldsymbol{w}_O^0\right\|_2 + C_O$ | $\mathbb{R}$ | Parameters norm |
| $\rho \triangleq N^{1/2} d_s^{3/2} \tau_1 C_x, \quad z \triangleq \bar{\lambda}_O^2\left(1 + 4\tau_0^2 C_x^4 d_s^2 \bar{\lambda}_V^2\left(\bar{\lambda}_Q^2 + \bar{\lambda}_K^2\right)\right)$ | $\mathbb{R}$ | Auxiliary terms |
| $f_i$ | $\mathbb{R}$ | Output of network for input $\boldsymbol{X}_i$ |
| $\mathcal{O}, o, \Omega$ and $\Theta$ | - | Standard Bachmann–Landau order notation |

\* The superscript with bracket represents the element of a vector/matrix, e.g., $\boldsymbol{w}^{(i)}$ is the $i^{\text{th}}$ element of $\boldsymbol{w}$.
\* The superscript without bracket symbolizes the parameters at different training steps, e.g., $\boldsymbol{\theta}^t$.
\* The subscript without bracket symbolizes the variable associated to the $n$-th data sample, e.g., $\boldsymbol{X}_n$.

# B  Theoretical background

## B.1  Preliminary on NTK

In this section, we summarize how training a neural network by minimizing squared loss, i.e., $\ell(\boldsymbol{\theta}^t) = \frac{1}{2}\sum_{n=1}^N (f(\boldsymbol{X}_n; \boldsymbol{\theta}^t) - y_n)^2$, via gradient descent can be characterized by the kernel regression predictor with NTK.

By choosing an infinitesimally small learning rate, we can obtain the following gradient flow:

$$\frac{d\boldsymbol{\theta}^t}{dt} = -\nabla\ell(\boldsymbol{\theta}^t).$$

By substituting the loss into the above equation and using the chain rule, we can find that the network outputs $f(\boldsymbol{\theta}^t) \in \mathbb{R}^N$ admit the following dynamics:

$$\frac{df(\boldsymbol{\theta}^t)}{dt} = -\boldsymbol{K}^t(f(\boldsymbol{\theta}^t) - \boldsymbol{y}), \tag{5}$$

where $\boldsymbol{K}^t = \left(\frac{\partial f(\boldsymbol{\theta}^t)}{\partial \boldsymbol{\theta}}\right)\left(\frac{\partial f(\boldsymbol{\theta}^t)}{\partial \boldsymbol{\theta}}\right)^\top \in \mathbb{R}^{N \times N}$. Jacot et al. [2018], Arora et al. [2019b] have shown that for fully-connected neural networks, under the infinite-width setting and proper initialization, $\boldsymbol{K}^t$ will be stable during training and $\boldsymbol{K}^0$ will converge to a fixed matrix $\boldsymbol{K}^\star \in \mathbb{R}^{N \times N}$, where $(\boldsymbol{K}^\star)^{(ij)} = K^\star(\boldsymbol{X}_i, \boldsymbol{X}_j)$ is the NTK value for the inputs $\boldsymbol{X}_i$ and $\boldsymbol{X}_j$. Then, we rewrite Eq. (5) as:

$$\frac{df(\boldsymbol{\theta}^t)}{dt} = -\boldsymbol{K}^\star(f(\boldsymbol{\theta}^t) - \boldsymbol{y}).$$

This implies the network output for any $\boldsymbol{X} \in \mathbb{R}^{d_s \times d}$ can be calculated by the kernel regression predictor with the associated NTK:

$$f(\boldsymbol{X}) = (K^\star(\boldsymbol{X}, \boldsymbol{X}_1), \cdots, K^\star(\boldsymbol{X}, \boldsymbol{X}_N)) \cdot (\boldsymbol{K}^\star)^{-1}\boldsymbol{y},$$

where $K^\star(\boldsymbol{X}, \boldsymbol{X}_n)$ is the kernel value between test data $\boldsymbol{X}$ and training data $\boldsymbol{X}_n$.

## B.2  Preliminary on Sub-Exponential random variables

Below, we overview the definition of a sub-exponential random variable and few related lemma based on Wainwright [2019]. A random variable $X$ with mean $\mu$ is called sub-exponential random variable if there exist non-negative parameters $(\nu, \alpha)$ such that

$$\mathbb{E}[e^{\lambda(X-\mu)}] \le e^{\frac{\nu^2\lambda^2}{2}} \quad \text{for all } |\lambda| < \frac{1}{\alpha},$$

and we denote by $X \sim SE(\nu, \alpha)$.

**Lemma 2.** *The multiplication of a scalar $s \in \mathbb{R}^+$ and a sub-exponential random variable $X \sim SE(\nu, \alpha)$ is still a sub-exponential random variable: $sX \sim SE(s\nu, s\alpha)$.*

**Lemma 3.** *Given a set of independent sub-exponential random variables $X_i \sim (\nu_i, \alpha_i)$ for $i \in 1, ..., N$, then $\sum_{i=1}^N X_i \sim (\sqrt{\sum_{i=1}^N \nu_i^2}, \max_i \alpha_i)$.*

**Lemma 4.** *Given a sub-exponential random variable $X \sim SE(\nu, \alpha)$ with mean $\mu$, the following inequality holds with probability at least $1 - \delta$:*

$$|X - \mu| < \max\left(\nu\sqrt{2\log\frac{2}{\delta}}, 2\alpha\log\frac{2}{\delta}\right).$$

## B.3  Related work on over-parameterization for convergence analysis

Recent empirical observation shows that neural networks can fit arbitrary labels with zero training loss when applying the first-order methods, e.g., gradient descent (GD) [Zhang et al., 2017]. Due to the highly non-convex and non-smooth instinct of the neural network, a large body of work have attempted to explain such a phenomenon. Early work studied the convergence of stochastic gradient descent (SGD) for training two-layer over-parameterized ReLU neural network with cross-entropy loss [Li and Liang, 2018]. Du et al. [2019b] show that only training the first layer of two-layer ReLU network with square loss by GD can lead to global convergence under the assumption that the Gram matrix is positive definite. Du et al. [2019a] extend the result to deep neural network with smooth activation function and shows that the convergence is guaranteed when the widths of all the hidden layers scale in $\Omega(N^4)$, where $N$ is the number of data points. Meanwhile, Zou and Gu [2019] prove that the condition for the convergence of GD for deep ReLU network is $\Omega(N^8)$, which improves upon Allen-Zhu et al. [2019a] that show the result of $\Omega(N^{24})$. Allen-Zhu et al. [2019a] also provide several convergence analyses under the setting of SGD and various loss functions. The assumption regarding the positive definiteness of the Gram matrix made in Du et al. [2019b] has been rigorously proved in Nguyen et al. [2021]. This facilitates Nguyen [2021] to demonstrate that deep ReLU network under LeCUN initialization with width in the order $\Omega(N^3)$ is enough for global convergence. Recent breakthrough [Bombari et al., 2022] improves previous results by showing that sub-linear layer widths suffice for deep neural network with smooth activation function.

Table 3: Over-parameterization conditions for the convergence analysis of neural network under gradient descent training with squared loss. $L$ is the depth of the network.

|  | Model | Depth | Initialization | Activation | Width |
|---|---|---|---|---|---|
| Allen-Zhu et al. [2019a] | FCNN/CNN | Deep | NTK | ReLU | $\Omega(N^{24}L^{12})$ |
| Du et al. [2019a] | FCNN/CNN | Deep | NTK | Smooth | $\Omega(N^4 2^{\mathcal{O}(L)})$ |
| Oymak and Soltanolkotabi [2020] | FCNN | Shallow | Standard Gaussian | ReLU | $\Omega(N^2)$ |
| Zou and Gu [2019] | FCNN | Deep | He | ReLU | $\Omega(N^8 L^{12})$ |
| Du et al. [2019b] | FCNN | Shallow | NTK | ReLU | $\Omega(N^6)$ |
| Nguyen [2021] | FCNN | Deep | LeCun | ReLU | $\Omega(N^3)$ |
| Chen et al. [2021] | FCNN | Deep | NTK | ReLU | $\Omega(L^{22})$ |
| Song et al. [2021] | FCNN | Shallow | He/Lecun | Smooth | $\Omega(N^{3/2})$ |
| Bombari et al. [2022] | FCNN | Deep | He/LeCun | Smooth | $\Omega(\sqrt{N})$ |
| Allen-Zhu et al. [2019b] | RNN | - | NTK | ReLU | $\Omega(N^c), c > 1$ |
| Hron et al. [2020] | Transformer | Deep | NTK | ReLU | - |
| Yang [2020] | Transformer | Deep | NTK | Softmax+ReLU | - |
| **Our** | Transformer | Shallow | Table 1 | Softmax+ReLU | $\Omega(N)$ |

## C  Proof for convergence analysis

This section is developed for the proof of the convergence result and we outline the flowchart below: Specifically, in Appendix C.1, we provide some auxiliary lemmas. Lemma 5 and Lemma 6 show that the norm of the parameters can be bounded with high probability at initialization. Lemmas 7, 9 and 10 present that the network output and the output of softmax between two adjacent time steps can be upper bounded. The Lipschitzness of network gradient and its norm is bounded in Lemmas 13 and 14. In Appendix C.2, we prove the convergence of the general cases, i.e., Proposition. 1. In Appendix C.3 and C.4 we present the proof for $d_m^{-1/2}$ and $d_m^{-1}$ scaling.

### C.1  Auxiliary lemmas

**Lemma 5** (Corollary 5.35 of Vershynin [2012]). *For a weight matrix $\boldsymbol{W} \in \mathbb{R}^{d_1 \times d_2}$ where each element is sampled independently from $\mathcal{N}(0,1)$, for every $\zeta \geq 0$, with probability at least $1 - 2\exp(-\zeta^2/2)$ one has:*

$$\sqrt{d_1} - \sqrt{d_2} - \zeta \leq \sigma_{\min}(\boldsymbol{W}) \leq \sigma_{\max}(\boldsymbol{W}) \leq \sqrt{d_1} + \sqrt{d_2} + \zeta,$$

*where $\sigma_{\max}(\boldsymbol{W})$ and $\sigma_{\min}(\boldsymbol{W})$ represents the maximum and minimum singular value of $\boldsymbol{W}$, respectively.*

**Lemma 6** (Upper bound of spectral norms of initial weight). *For a weight matrix $\boldsymbol{W} \in \mathbb{R}^{d_m \times d}$ where $d_m > d$, each element is sampled independently from $\mathcal{N}(0,1)$, with probability at least $1 - 2\exp(-d_m/2)$, one has:*

$$\|\boldsymbol{W}\|_2 \leq 3\sqrt{d_m}.$$

*Proof of Lemma 6.* Following Lemma 5, one has:

$$\|\boldsymbol{W}\|_2 \leq \sqrt{d_m} + \sqrt{d} + \zeta.$$

Letting $\zeta = \sqrt{d_m}$ and using the fact that $d_m > d$ complete the proof. □

**Lemma 7.** *Recall from Table 2, $\boldsymbol{\beta}_i = \sigma_s \left(\tau_0 \boldsymbol{X}^{(i,:)} \boldsymbol{W}_Q^\top \boldsymbol{W}_K \boldsymbol{X}^\top\right)^\top$ is the $i$-th row of the output of Softmax, if $\max\left(\left\|\boldsymbol{W}_Q^t\right\|_2, \left\|\boldsymbol{W}_Q^{t'}\right\|_2\right) \leq \bar{\lambda}_Q, \max\left(\left\|\boldsymbol{W}_K^t\right\|_2, \left\|\boldsymbol{W}_K^{t'}\right\|_2\right) \leq \bar{\lambda}_K$, then its difference*

*between $t'$ step and $t$ step has the following upper bound:*

$$\left\| \boldsymbol{\beta}_i^{t'} - \boldsymbol{\beta}_i^t \right\|_2 \leq 2\tau_0 C_x^2 d_s \left( \bar{\lambda}_Q \left\| \boldsymbol{W}_K^{t'} - \boldsymbol{W}_K^t \right\|_2 + \bar{\lambda}_K \left\| \boldsymbol{W}_Q^{t'} - \boldsymbol{W}_Q^t \right\|_2 \right) .$$

*Proof.*

$$\left\| \boldsymbol{\beta}_i^{t'} - \boldsymbol{\beta}_i^t \right\|_2$$
$$\leq \left\| \boldsymbol{\beta}_i^{t'} - \boldsymbol{\beta}_i^t \right\|_1$$
$$\leq 2 \left\| \tau_0 \boldsymbol{X}^{(i,:)} \boldsymbol{W}_Q^{t'\top} \boldsymbol{W}_K^{t'} \boldsymbol{X}^\top - \tau_0 \boldsymbol{X}^{(i,:)} \boldsymbol{W}_Q^{t\top} \boldsymbol{W}_K^t \boldsymbol{X}^\top \right\|_\infty \text{ (By Corollary A.7 in Edelman et al. [2022])}$$
$$= 2 \max_j |\tau_0 \boldsymbol{X}^{(i,:)} \boldsymbol{W}_Q^{t'\top} \boldsymbol{W}_K^{t'} \boldsymbol{X}^{(j,:)\top} - \tau_0 \boldsymbol{X}^{(i,:)} \boldsymbol{W}_Q^{t\top} \boldsymbol{W}_K^t \boldsymbol{X}^{(j,:)\top}|$$
$$\leq 2 \max_j \left( \tau_0 \left\| \boldsymbol{X}^{(i,:)} \right\|_2 \left\| \boldsymbol{W}_Q^{t'\top} \boldsymbol{W}_K^{t'} - \boldsymbol{W}_Q^{t\top} \boldsymbol{W}_K^t \right\|_2 \left\| \boldsymbol{X}^{(j,:)} \right\|_2 \right) \text{ (By Cauchy-Schwarz inequality)}$$
$$\leq 2\tau_0 C_x^2 d_s \left\| \boldsymbol{W}_Q^{t'\top} \boldsymbol{W}_K^{t'} - \boldsymbol{W}_Q^{t\top} \boldsymbol{W}_K^t \right\|_2 \text{ (By Assumption 1)}$$
$$\leq 2\tau_0 C_x^2 d_s \left( \left\| \boldsymbol{W}_Q^{t'} \right\|_2 \left\| \boldsymbol{W}_K^{t'} - \boldsymbol{W}_K^t \right\|_2 + \left\| \boldsymbol{W}_K^t \right\|_2 \left\| \boldsymbol{W}_Q^{t'} - \boldsymbol{W}_Q^t \right\|_2 \right) \text{ (By Cauchy-Schwarz inequality, Triangle inequality)}$$
$$\leq 2\tau_0 C_x^2 d_s \left( \bar{\lambda}_Q \left\| \boldsymbol{W}_K^{t'} - \boldsymbol{W}_K^t \right\|_2 + \bar{\lambda}_K \left\| \boldsymbol{W}_Q^{t'} - \boldsymbol{W}_Q^t \right\|_2 \right) .$$

$\square$

**Lemma 8** (Upper bound the Euclidean norm of the output of softmax). *Suppose $\boldsymbol{v} = \text{Softmax}(\boldsymbol{u}) \in \mathbb{R}^{d_s}$, then one has: $1/\sqrt{d_s} \leq \|\boldsymbol{v}\|_2 \leq 1$ .*

*Proof.* By the inequality of arithmetic and geometric mean:

$$\frac{1}{\sqrt{d_s}} = \frac{\sum_{i=1}^{d_s} \boldsymbol{v}^{(i)}}{\sqrt{d_s}} \leq \|\boldsymbol{v}\|_2 = \sqrt{\sum_{i=1}^{d_s} \left( (\boldsymbol{v}^{(i)})^2 \right)} \leq \sqrt{\left( \sum_{i=1}^{d_s} (\boldsymbol{v}^{(i)}) \right)^2} = 1 .$$

$\square$

**Lemma 9.** *If at $t'$ and $t$ step, $\max \left( \left\| \boldsymbol{W}_Q^t \right\|_2, \left\| \boldsymbol{W}_Q^{t'} \right\|_2 \right) \leq \bar{\lambda}_Q, \max \left( \left\| \boldsymbol{W}_K^t \right\|_2, \left\| \boldsymbol{W}_K^{t'} \right\|_2 \right) \leq \bar{\lambda}_K, \max \left( \left\| \boldsymbol{W}_V^t \right\|_2, \left\| \boldsymbol{W}_V^{t'} \right\|_2 \right) \leq \bar{\lambda}_V$, then the difference between the output of the last hidden layer at $t'$ step and $t$ can be upper bounded by:*

$$\left\| \boldsymbol{F}_{pre}^{t'} - \boldsymbol{F}_{pre}^t \right\|_{\mathrm{F}} \leq \rho \left( \left\| \boldsymbol{W}_V^{t'} - \boldsymbol{W}_V^t \right\|_2 + \bar{\lambda}_V 2\tau_0 C_x^2 d_s \left( \bar{\lambda}_Q \left\| \boldsymbol{W}_K^{t'} - \boldsymbol{W}_K^t \right\|_2 + \bar{\lambda}_K \left\| \boldsymbol{W}_Q^{t'} - \boldsymbol{W}_Q^t \right\|_2 \right) \right) .$$

*Proof.* Note that $\boldsymbol{F}_{\text{pre}} \in \mathbb{R}^{N \times d_m}$ is the output of the last hidden layer given $\{\boldsymbol{X}_n\}_{n=1}^N$ sample as defined in Table 2. We firstly analyze each sample, i.e., $\boldsymbol{f}_{\text{pre}} \in \mathbb{R}^{d_m}$ and we drop the index $n$ for simplification.

$$\left\| \boldsymbol{f}_{\mathrm{pre}}^{t'} - \boldsymbol{f}_{\mathrm{pre}}^{t} \right\|_2 = \tau_1 \left\| \sum_{i=1}^{d_s} \sigma_r \left( \boldsymbol{W}_V^{t'} \boldsymbol{X}^\top \boldsymbol{\beta}_i^{t'} \right) - \sum_{i=1}^{d_s} \sigma_r \left( \boldsymbol{W}_V^{t} \boldsymbol{X}^\top \boldsymbol{\beta}_i^{t} \right) \right\|_2$$

$$\leq \tau_1 \sum_{i=1}^{d_s} \left\| \boldsymbol{W}_V^{t'} \boldsymbol{X}^\top \boldsymbol{\beta}_i^{t'} - \boldsymbol{W}_V^{t} \boldsymbol{X}^\top \boldsymbol{\beta}_i^{t} \right\|_2 \text{ (By Lipschitz continuity of } \sigma_r)$$

$$\leq \tau_1 \sum_{i=1}^{d_s} \left( \left\| (\boldsymbol{W}_V^{t'} - \boldsymbol{W}_V^{t}) \boldsymbol{X}^\top \boldsymbol{\beta}_i^{t'} \right\|_2 + \left\| \boldsymbol{W}_V^{t} \boldsymbol{X} (\boldsymbol{\beta}_i^{t'} - \boldsymbol{\beta}_i^{t}) \right\|_2 \right) \text{ (By Triangle inequality)}$$

$$\leq \tau_1 \sum_{i=1}^{d_s} \left( \left\| \boldsymbol{W}_V^{t'} - \boldsymbol{W}_V^{t} \right\|_2 \left\| \boldsymbol{X} \right\|_2 \left\| \boldsymbol{\beta}_i^{t'} \right\|_2 + \left\| \boldsymbol{W}_V^{t} \right\|_2 \left\| \boldsymbol{X} \right\|_2 \left\| \boldsymbol{\beta}_i^{t'} - \boldsymbol{\beta}_i^{t} \right\|_2 \right) \text{ (By Cauchy-Schwarz inequality)}$$

$$\leq \tau_1 C_x \sqrt{d_s} \sum_{i=1}^{d_s} \left( \left\| \boldsymbol{W}_V^{t'} - \boldsymbol{W}_V^{t} \right\|_2 + \bar{\lambda}_V \left\| \boldsymbol{\beta}_i^{t'} - \boldsymbol{\beta}_i^{t} \right\|_2 \right) \text{ (By Lemma 8 and Assumption 1)}$$

$$\leq \tau_1 C_x d_s^{3/2} \left( \left\| \boldsymbol{W}_V^{t'} - \boldsymbol{W}_V^{t} \right\|_2 + \bar{\lambda}_V 2 \tau_0 C_x^2 d_s \left( \bar{\lambda}_Q \left\| \boldsymbol{W}_K^{t'} - \boldsymbol{W}_K^{t} \right\|_2 + \bar{\lambda}_K \left\| \boldsymbol{W}_Q^{t'} - \boldsymbol{W}_Q^{t} \right\|_2 \right) \right) \text{ (By Lemma 7)} .$$
(6)

Next, we bound the difference given $N$ data sample:

$$\left\| \boldsymbol{F}_{\mathrm{pre}}^{t'} - \boldsymbol{F}_{\mathrm{pre}}^{t} \right\|_{\mathrm{F}} \leq \sqrt{N} \tau_1 C_x d_s^{3/2} \left( \left\| \boldsymbol{W}_V^{t'} - \boldsymbol{W}_V^{t} \right\|_2 + \bar{\lambda}_V 2 \tau_0 C_x^2 d_s \left( \bar{\lambda}_Q \left\| \boldsymbol{W}_K^{t'} - \boldsymbol{W}_K^{t} \right\|_2 + \bar{\lambda}_K \left\| \boldsymbol{W}_Q^{t'} - \boldsymbol{W}_Q^{t} \right\|_2 \right) \right)$$

$$= \rho \left( \left\| \boldsymbol{W}_V^{t'} - \boldsymbol{W}_V^{t} \right\|_2 + \bar{\lambda}_V 2 \tau_0 C_x^2 d_s \left( \bar{\lambda}_Q \left\| \boldsymbol{W}_K^{t'} - \boldsymbol{W}_K^{t} \right\|_2 + \bar{\lambda}_K \left\| \boldsymbol{W}_Q^{t'} - \boldsymbol{W}_Q^{t} \right\|_2 \right) \right) ,$$

where the last equality is by the definition of $\rho$ in Proposition. 1. $\qquad\square$

**Lemma 10.** *If at $t'$ and $t$ step,* $\max \left( \left\| \boldsymbol{W}_Q^{t} \right\|_2, \left\| \boldsymbol{W}_Q^{t'} \right\|_2 \right) \leq \bar{\lambda}_Q, \max \left( \left\| \boldsymbol{W}_K^{t} \right\|_2, \left\| \boldsymbol{W}_K^{t'} \right\|_2 \right) \leq \bar{\lambda}_K, \max \left( \left\| \boldsymbol{W}_V^{t} \right\|_2, \left\| \boldsymbol{W}_V^{t'} \right\|_2 \right) \leq \bar{\lambda}_V, \max \left( \left\| \boldsymbol{w}_O^{t} \right\|_2, \left\| \boldsymbol{w}_O^{t'} \right\|_2 \right) \leq \bar{\lambda}_O,$ *then the difference between the network output at $t'$ step and $t$ step can be upper bounded by:*

$$\left\| \boldsymbol{f}^{t'} - \boldsymbol{f}^{t} \right\|_2 \leq \rho \bar{\lambda}_V \left\| \boldsymbol{w}_O^{t'} - \boldsymbol{w}_O^{t} \right\|_2 + \bar{\lambda}_O \left\| \boldsymbol{F}_{\mathrm{pre}}^{t'} - \boldsymbol{F}_{\mathrm{pre}}^{t} \right\|_{\mathrm{F}} .$$

*Proof.*

$$\left\| \boldsymbol{f}^{t'} - \boldsymbol{f}^{t} \right\|_2 = \left\| \boldsymbol{F}_{\mathrm{pre}}^{t'} \boldsymbol{w}_O^{t'} - \boldsymbol{F}_{\mathrm{pre}}^{t} \boldsymbol{w}_O^{t} \right\|_2 \leq \left\| \boldsymbol{F}_{\mathrm{pre}}^{t'} \right\|_2 \left\| \boldsymbol{w}_O^{t'} - \boldsymbol{w}_O^{t} \right\|_2 + \left\| \boldsymbol{w}_O^{t} \right\|_2 \left\| \boldsymbol{F}_{\mathrm{pre}}^{t'} - \boldsymbol{F}_{\mathrm{pre}}^{t} \right\|_2 ,$$

where we use triangle inequality. Then, the first term of the RHS can be bounded by:

$$\left\| \boldsymbol{F}_{\mathrm{pre}}^{t'} \right\|_2 \leq \left\| \boldsymbol{F}_{\mathrm{pre}}^{t'} \right\|_{\mathrm{F}} = \left\| \begin{bmatrix} \tau_1 \sum_{i=1}^{d_s} \sigma_r \left( \boldsymbol{W}_V^{t'} \boldsymbol{X}_1^\top \boldsymbol{\beta}_{i,1}^{t'} \right) \\ \vdots \\ \tau_1 \sum_{i=1}^{d_s} \sigma_r \left( \boldsymbol{W}_V^{t'} \boldsymbol{X}_N^\top \boldsymbol{\beta}_{i,N}^{t'} \right) \end{bmatrix} \right\|_{\mathrm{F}} \leq \tau_1 \sqrt{N} d_s^{3/2} C_x \left\| \boldsymbol{W}_V^{t'} \right\|_2 = \rho \left\| \boldsymbol{W}_V^{t'} \right\|_2 ,$$

where the last equality is by the definition of $\rho$ in Proposition. 1. $\qquad\square$

**Lemma 11** (Jacobian of Softmax). *Suppose $\boldsymbol{v} = \mathrm{Softmax}(\boldsymbol{u}) \in \mathbb{R}^{d_s}$, then $\frac{\partial \boldsymbol{v}}{\partial \boldsymbol{u}} = \mathrm{diag}(\boldsymbol{v}) - \boldsymbol{v} \boldsymbol{v}^\top$.*

*Proof.* We can reformulate $\boldsymbol{v}$ as: $\boldsymbol{v} = \begin{bmatrix} \frac{\exp \left( \boldsymbol{u}^{(1)} \right)}{\sum_{i=1}^{d_s} \exp \left( \boldsymbol{u}^{(i)} \right)} \\ \vdots \\ \frac{\exp \left( \boldsymbol{u}^{(d_s)} \right)}{\sum_{i=1}^{d_s} \exp \left( \boldsymbol{u}^{(i)} \right)} \end{bmatrix} .$

Then, we have

$$\frac{\partial \boldsymbol{v}^{(j)}}{\partial \boldsymbol{u}^{(k)}} = \frac{\partial \frac{\exp\left(\boldsymbol{u}^{(j)}\right)}{\sum_{i=1}^{d_s} \exp\left(\boldsymbol{u}^{(i)}\right)}}{\partial \boldsymbol{u}^{(k)}} = \begin{cases} \frac{-\exp\left(\boldsymbol{u}^{(j)}\right) - \exp\left(\boldsymbol{u}^{(k)}\right)}{\left(\sum_{i=1}^{d_s} \exp\left(\boldsymbol{u}^{(i)}\right)\right)^2} & \text{if } j \neq k \\ \frac{\exp\left(\boldsymbol{u}^{(k)}\right) \sum_{i=1}^{d_s} \exp\left(\boldsymbol{u}^{(i)}\right) - \left(\exp\left(\boldsymbol{u}^{(k)}\right)\right)^2}{\left(\sum_{i=1}^{d_s} \exp\left(\boldsymbol{u}^{(i)}\right)\right)^2} & \text{if } j = k \end{cases}$$

$$= \begin{cases} -\boldsymbol{v}^{(j)} \boldsymbol{v}^{(k)} & \text{if } j \neq k \\ \boldsymbol{v}^{(k)} - \boldsymbol{v}^{(j)} \boldsymbol{v}^{(k)} & \text{if } j = k \end{cases}$$

Thus

$$\frac{\partial \boldsymbol{v}}{\partial \mathbf{u}} = \operatorname{diag}(\boldsymbol{v}) - \boldsymbol{v}\boldsymbol{v}^\top .$$

$\square$

**Lemma 12** (Upper bound for the loss gradient norm). *If $\left\|\boldsymbol{W}_Q^t\right\|_2 \leq \bar{\lambda}_Q, \left\|\boldsymbol{W}_K^t\right\|_2 \leq \bar{\lambda}_K, \left\|\boldsymbol{W}_V^t\right\|_2 \leq \bar{\lambda}_V, \left\|\boldsymbol{w}_O^t\right\|_2 \leq \bar{\lambda}_O$, then the gradient norm with respect to $\boldsymbol{W}_Q, \boldsymbol{W}_K, \boldsymbol{W}_V, \boldsymbol{w}_O$ can be upper bounded by:*

$$\left\|\nabla_{\boldsymbol{W}_Q} \ell(\boldsymbol{\theta}^\top)\right\|_F \leq 2\rho\tau_0 \bar{\lambda}_K \bar{\lambda}_V \bar{\lambda}_O d_s C_x^2 \left\|\boldsymbol{f}^t - \boldsymbol{y}\right\|_2 , \qquad \left\|\nabla_{\boldsymbol{W}_K} \ell(\boldsymbol{\theta}^\top)\right\|_F \leq 2\rho\tau_0 \bar{\lambda}_Q \bar{\lambda}_V \bar{\lambda}_O d_s C_x^2 \left\|\boldsymbol{f}^t - \boldsymbol{y}\right\|_2 ,$$

$$\left\|\nabla_{\boldsymbol{W}_V} \ell(\boldsymbol{\theta}^\top)\right\|_F \leq \rho\bar{\lambda}_O \left\|\boldsymbol{f}^t - \boldsymbol{y}\right\|_2 , \qquad \left\|\nabla_{\boldsymbol{w}_O} \ell(\boldsymbol{\theta}^\top)\right\|_2 \leq \rho\bar{\lambda}_V \left\|\boldsymbol{f}^t - \boldsymbol{y}\right\|_2 .$$

*Proof.* To simplify the notation, in the proof below, we hide the index $t$. Firstly, consider the gradient w.r.t $\boldsymbol{W}_V$,

$$\left\|\nabla_{\boldsymbol{W}_V} \ell(\boldsymbol{\theta})\right\|_F = \left\|-\sum_{n=1}^N (f(\boldsymbol{X}_n) - y_n)\frac{\partial f(\boldsymbol{X}_n)}{\partial \boldsymbol{W}_V}\right\|_F$$

$$= \tau_1 \left\|\sum_{n=1}^N (f(\boldsymbol{X}_n) - y_n) \sum_{i=1}^{d_s} \left(\boldsymbol{w}_O \circ \dot{\sigma}_r \left(\boldsymbol{W}_V \boldsymbol{X}_n^\top \boldsymbol{\beta}_{i,n}\right)\right) \boldsymbol{\beta}_{i,n}^\top \boldsymbol{X}_n\right\|_F \leq \tau_1 \sum_{n=1}^N |(f(\boldsymbol{X}_n) - y_n)| d_s^{3/2} \bar{\lambda}_O C_x$$

$$\leq \tau_1 \sqrt{N \sum_{n=1}^N |(f(\boldsymbol{X}_n) - y_n)|^2 d_s^{3/2} \bar{\lambda}_O C_x} = \tau_1 d_s^{3/2} \bar{\lambda}_O C_x \sqrt{N} \left\|\boldsymbol{f} - \boldsymbol{y}\right\|_2 = \rho\bar{\lambda}_O \left\|\boldsymbol{f} - \boldsymbol{y}\right\|_2 ,$$

(7)

where the last equality is by the definition of $\rho$ in Proposition. 1. Next, consider the gradient w.r.t $\boldsymbol{w}_O$,

$$\left\|\nabla_{\boldsymbol{w}_O} \ell(\boldsymbol{\theta})\right\|_2 = \left\|-\sum_{n=1}^N (f(\boldsymbol{X}_n) - y_n)\frac{\partial f(\boldsymbol{X}_n)}{\partial \boldsymbol{w}_O}\right\|_2 = \tau_1 \left\|\sum_{n=1}^N (f(\boldsymbol{X}_n) - y_n) \sum_{i=1}^{d_s} \sigma_r \left(\boldsymbol{W}_V \boldsymbol{X}_n^\top \boldsymbol{\beta}_{i,n}\right)\right\|_2$$

$$\leq \tau_1 \sum_{n=1}^N |(f(\boldsymbol{X}_n) - y_n)| d_s^{3/2} \bar{\lambda}_V C_x \leq \tau_1 \sqrt{N \sum_{n=1}^N |(f(\boldsymbol{X}_n) - y_n)|^2 d_s^{3/2} \bar{\lambda}_V C_x} = \rho\bar{\lambda}_V \left\|\boldsymbol{f} - \boldsymbol{y}\right\|_2 ,$$

(8)

Next, consider the gradient w.r.t $\boldsymbol{W}_Q$,

$$
\left\|\nabla_{\boldsymbol{W}_Q} \ell(\boldsymbol{\theta})\right\|_{\mathrm{F}} = \left\|-\sum_{n=1}^{N}(f(\boldsymbol{X}_n) - y_n)\frac{\partial f(\boldsymbol{X}_n)}{\partial \boldsymbol{W}_Q}\right\|_{\mathrm{F}}
$$

$$
= \tau_0\tau_1 \left\|\sum_{n=1}^{N}(f(\boldsymbol{X}_n) - y_n)\sum_{i=1}^{d_s} \boldsymbol{W}_k \boldsymbol{X}_n^\top \left(\mathrm{diag}(\boldsymbol{\beta}_{i,n}) - \boldsymbol{\beta}_{i,n}\boldsymbol{\beta}_{i,n}^\top\right)\boldsymbol{X}_n \boldsymbol{W}_V^\top \left(\boldsymbol{w}_O \circ \dot{\sigma}_r\left(\boldsymbol{W}_V \boldsymbol{X}_n^\top \boldsymbol{\beta}_{i,n}\right)\right)\boldsymbol{X}_n^{(i,:)}\right\|_{\mathrm{F}}
$$

$$
\leq 2\tau_0\tau_1 \sum_{n=1}^{N}|(f(\boldsymbol{X}_n) - y_n)|\, d_s\bar{\lambda}_K\bar{\lambda}_V\bar{\lambda}_O(C_x\sqrt{d_s})^3
$$

$$
\leq 2\tau_0\tau_1 \sqrt{N\sum_{n=1}^{N}|(f(\boldsymbol{X}_n) - y_n)|^2 d_s\bar{\lambda}_K\bar{\lambda}_V\bar{\lambda}_O(C_x\sqrt{d_s})^3}
$$

$$
= 2\rho\tau_0\bar{\lambda}_K\bar{\lambda}_V\bar{\lambda}_O d_s C_x^2 \left\|\boldsymbol{f} - \boldsymbol{y}\right\|_2 .
$$

$$(9)$$

Similarly, for the gradient w.r.t $\boldsymbol{W}_K$, we have:

$$
\left\|\nabla_{\boldsymbol{W}_K} \ell(\boldsymbol{\theta})\right\|_{\mathrm{F}} \leq 2\rho\tau_0\bar{\lambda}_Q\bar{\lambda}_V\bar{\lambda}_O d_s C_x^2 \left\|\boldsymbol{f} - \boldsymbol{y}\right\|_2 . \tag{10}
$$

$\square$

**Lemma 13** (Upper bound for the network function gradient norm)**.** *If* $\left\|\boldsymbol{W}_Q^t\right\|_2 \leq \bar{\lambda}_Q, \|\boldsymbol{W}_K^t\|_2 \leq \bar{\lambda}_K, \|\boldsymbol{W}_V^t\|_2 \leq \bar{\lambda}_V, \|\boldsymbol{w}_O^t\|_2 \leq \bar{\lambda}_O,$ *then one has:*

$$
\left\|\nabla_{\boldsymbol{\theta}} \boldsymbol{f}^t\right\|_2 \leq c_2 ,
$$

*where*

$$
c_2 \triangleq \rho\sqrt{\bar{\lambda}_O^2 + \bar{\lambda}_V^2 + (2\tau_0\bar{\lambda}_K\bar{\lambda}_V\bar{\lambda}_O d_s C_x^2)^2 + (2\tau_0\bar{\lambda}_Q\bar{\lambda}_V\bar{\lambda}_O d_s C_x^2)^2} . \tag{11}
$$

*Proof.* To simplify the notation, in the proof below, we hide the index $t$. Firstly, note that:

$$
\|\nabla_{\boldsymbol{\theta}} \boldsymbol{f}\|_2 \leq \|\nabla_{\boldsymbol{\theta}} \boldsymbol{f}\|_{\mathrm{F}}
$$
$$
= \sqrt{\sum_{n=1}^{N}\left(\|\nabla_{\boldsymbol{w}_O} f(\boldsymbol{X}_n; \boldsymbol{\theta})\|_2^2 + \|\nabla_{\boldsymbol{w}_Q} f(\boldsymbol{X}_n; \boldsymbol{\theta})\|_{\mathrm{F}}^2 + \|\nabla_{\boldsymbol{w}_K} f(\boldsymbol{X}_n; \boldsymbol{\theta})\|_{\mathrm{F}}^2 + \|\nabla_{\boldsymbol{W}_V} f(\boldsymbol{X}_n; \boldsymbol{\theta})\|_{\mathrm{F}}^2\right)}.
$$

$$(12)$$

Then following the step as in Eqs. (7) to (10), each term can be bounded as follows:

$$
\sum_{n=1}^{N}\left(\|\nabla_{\boldsymbol{W}_V} f(\boldsymbol{X}_n; \boldsymbol{\theta})\|_{\mathrm{F}}^2\right) \leq (\rho\bar{\lambda}_O)^2, \qquad\qquad \sum_{n=1}^{N}\left(\|\nabla_{\boldsymbol{w}_O} f(\boldsymbol{X}_n; \boldsymbol{\theta})\|_2\right) \leq (\rho\bar{\lambda}_V)^2 ,
$$
$$
\sum_{n=1}^{N}\left(\|\nabla_{\boldsymbol{w}_Q} f(\boldsymbol{X}_n; \boldsymbol{\theta})\|_{\mathrm{F}}\right) \leq (2\rho\tau_0\bar{\lambda}_K\bar{\lambda}_V\bar{\lambda}_O d_s C_x^2)^2, \quad \sum_{n=1}^{N}\left(\|\nabla_{\boldsymbol{w}_K} f(\boldsymbol{X}_n; \boldsymbol{\theta})\|_{\mathrm{F}}\right) \leq (2\rho\tau_0\bar{\lambda}_Q\bar{\lambda}_V\bar{\lambda}_O d_s C_x^2)^2 .
$$

Plugging these bounds back Eq. (12) finishes the proof. $\square$

**Lemma 14** (Upper bound for the Lipschitzness of the network gradient)**.** *Suppose* $\max\left(\left\|\boldsymbol{W}_Q^t\right\|_2, \left\|\boldsymbol{W}_Q^{t'}\right\|_2\right) \leq \bar{\lambda}_Q, \max\left(\|\boldsymbol{W}_K^t\|_2, \left\|\boldsymbol{W}_K^{t'}\right\|_2\right) \leq \bar{\lambda}_K, \max\left(\|\boldsymbol{W}_V^t\|_2, \left\|\boldsymbol{W}_V^{t'}\right\|_2\right) \leq \bar{\lambda}_V, \max\left(\|\boldsymbol{w}_O^t\|_2, \left\|\boldsymbol{w}_O^{t'}\right\|_2\right) \leq \bar{\lambda}_O,$ *and define* $z \triangleq 2\tau_0 C_x^2 d_s(\bar{\lambda}_Q + \bar{\lambda}_K),$ *then one has*

$$
\left\|\nabla_{\boldsymbol{\theta}} \boldsymbol{f}^{t'} - \nabla_{\boldsymbol{\theta}} \boldsymbol{f}^t\right\|_2 \leq c_3 \left\|\boldsymbol{\theta}^{t'} - \boldsymbol{\theta}^t\right\|_2 , \tag{13}
$$

*where*

$$(c_3)^2 \triangleq N(\tau_1 C_x d_s^{3/2}(1+\bar{\lambda}_V z))^2$$

$$+ N\left\{\tau_1 C_x d_s^{3/2}\left[\bar{\lambda}_O z + \bar{\lambda}_O C_x \sqrt{d_s}(1+\bar{\lambda}_V z)+1\right]\right\}^2$$

$$+ N\left\{\tau_0 \tau_1 C_x d_s\left\{2\bar{\lambda}_K d_s C_x^2\left[\bar{\lambda}_V\left(\bar{\lambda}_O C_x \sqrt{d_s}(1+\bar{\lambda}_V z)+1\right)+\bar{\lambda}_O\right]+\bar{\lambda}_V \bar{\lambda}_O[C_x^2 d_s + 3\bar{\lambda}_K z]\right\}\right\}^2$$

$$+ N\left\{\tau_0 \tau_1 C_x d_s\left\{2\bar{\lambda}_Q d_s C_x^2\left[\bar{\lambda}_V\left(\bar{\lambda}_O C_x \sqrt{d_s}(1+\bar{\lambda}_V z)+1\right)+\bar{\lambda}_O\right]+\bar{\lambda}_V \bar{\lambda}_O[C_x^2 d_s + 3\bar{\lambda}_K z]\right\}\right\}^2 .$$
$$\tag{14}$$

*Proof.* Firstly, note that:

$$\left\|\nabla_{\boldsymbol{\theta}} \boldsymbol{f}^{t'} - \nabla_{\boldsymbol{\theta}} \boldsymbol{f}^t\right\|_2^2$$

$$\leq \sum_{n=1}^N \left(\left\|\nabla_{\boldsymbol{w}_O} f(\boldsymbol{X}_n; \boldsymbol{\theta}^{t'}) - \nabla_{\boldsymbol{w}_O} f(\boldsymbol{X}_n; \boldsymbol{\theta}^t)\right\|_2^2 + \left\|\nabla_{\boldsymbol{W}_V} f(\boldsymbol{X}_n; \boldsymbol{\theta}^{t'}) - \nabla_{\boldsymbol{W}_V} f(\boldsymbol{X}_n; \boldsymbol{\theta}^t)\right\|_{\mathrm{F}}^2\right.$$

$$\left. + \left\|\nabla_{\boldsymbol{W}_Q} f(\boldsymbol{X}_n; \boldsymbol{\theta}^{t'}) - \nabla_{\boldsymbol{W}_Q} f(\boldsymbol{X}_n; \boldsymbol{\theta}^t)\right\|_{\mathrm{F}}^2 + \left\|\nabla_{\boldsymbol{W}_K} f(\boldsymbol{X}_n; \boldsymbol{\theta}^{t'}) - \nabla_{\boldsymbol{W}_K} f(\boldsymbol{X}_n; \boldsymbol{\theta}^t)\right\|_{\mathrm{F}}^2\right) .$$
$$\tag{15}$$

Then, we will prove each term separately. Regarding the first term in Eq. (15), we have

$$\left\|\nabla_{\boldsymbol{w}_O} f(\boldsymbol{X}_n; \boldsymbol{\theta}^{t'}) - \nabla_{\boldsymbol{w}_O} f(\boldsymbol{X}_n; \boldsymbol{\theta}^t)\right\|_2$$

$$= \left\|\boldsymbol{f}_{\mathrm{pre}}^{t'} - \boldsymbol{f}_{\mathrm{pre}}^t\right\|_2$$

$$\leq \tau_1 C_x d_s^{3/2}\left(\left\|\boldsymbol{W}_V^{t'} - \boldsymbol{W}_V^t\right\|_2 + \bar{\lambda}_V 2\tau_0 C_x^2 d_s\left(\bar{\lambda}_Q \left\|\boldsymbol{W}_K^{t'} - \boldsymbol{W}_K^t\right\|_2 + \bar{\lambda}_K \left\|\boldsymbol{W}_Q^{t'} - \boldsymbol{W}_Q^t\right\|_2\right)\right)$$

$$\leq \tau_1 C_x d_s^{3/2}\left(1 + \bar{\lambda}_V 2\tau_0 C_x^2 d_s\left(\bar{\lambda}_Q + \bar{\lambda}_K\right)\right)\left\|\boldsymbol{\theta}^{t'} - \boldsymbol{\theta}^t\right\|_2$$

$$= \tau_1 C_x d_s^{3/2}(1+\bar{\lambda}_V z)\left\|\boldsymbol{\theta}^{t'} - \boldsymbol{\theta}^t\right\|_2 .$$
$$\tag{16}$$

where the first inequality is by Eq. (6), and in the last equality is by the definition of $z$ in the lemma. Regarding the second term in Eq. (15), we have:

$$\left\|\nabla_{\boldsymbol{W}_V} f(\boldsymbol{X}_n; \boldsymbol{\theta}^{t'}) - \nabla_{\boldsymbol{W}_V} f(\boldsymbol{X}_n; \boldsymbol{\theta}^t)\right\|_{\mathrm{F}}$$

$$= \tau_1 \left\|\sum_{i=1}^{d_s}\left(\boldsymbol{w}_O^{t'} \circ \dot{\sigma}_r\left(\boldsymbol{W}_V^{t'} \boldsymbol{X}_n^\top \boldsymbol{\beta}_{i,n}^{t'}\right)\right)\boldsymbol{\beta}_{i,n}^{t'}{}^\top \boldsymbol{X}_n - \sum_{i=1}^{d_s}\left(\boldsymbol{w}_O^t \circ \dot{\sigma}_r\left(\boldsymbol{W}_V^t \boldsymbol{X}_n^\top \boldsymbol{\beta}_{i,n}^t\right)\right)\boldsymbol{\beta}_{i,n}^t{}^\top \boldsymbol{X}_n\right\|_{\mathrm{F}}$$

$$\leq \tau_1 C_x \sqrt{d_s} \sum_{i=1}^{d_s}\left\|\left(\boldsymbol{w}_O^{t'} \circ \dot{\sigma}_r\left(\boldsymbol{W}_V^{t'} \boldsymbol{X}_n^\top \boldsymbol{\beta}_{i,n}^{t'}\right)\right)\boldsymbol{\beta}_{i,n}^{t'}{}^\top - \left(\boldsymbol{w}_O^t \circ \dot{\sigma}_r\left(\boldsymbol{W}_V^t \boldsymbol{X}_n^\top \boldsymbol{\beta}_{i,n}^t\right)\right)\boldsymbol{\beta}_{i,n}^t{}^\top\right\|_{\mathrm{F}}$$

$$\leq \tau_1 C_x \sqrt{d_s} \sum_{i=1}^{d_s}\left[\left\|\boldsymbol{w}_O^{t'} \circ \dot{\sigma}_r\left(\boldsymbol{W}_V^{t'} \boldsymbol{X}_n^\top \boldsymbol{\beta}_{i,n}^{t'}\right)\right\|_2 \left\|\boldsymbol{\beta}_{i,n}^{t'} - \boldsymbol{\beta}_{i,n}^t\right\|_2 + \|\boldsymbol{\beta}_{i,n}^t\|_2 \left\|\boldsymbol{w}_O^{t'} \circ \dot{\sigma}_r\left(\boldsymbol{W}_V^{t'} \boldsymbol{X}_n^\top \boldsymbol{\beta}_{i,n}^{t'}\right) - \tilde{\boldsymbol{w}}_O^t \circ \dot{\sigma}_r\left(\tilde{\boldsymbol{W}}_V^t \boldsymbol{X}_m^\top \tilde{\boldsymbol{\beta}}_{i,n}^t\right)\right\|_2\right]$$

$$\leq \tau_1 C_x \sqrt{d_s} \sum_{i=1}^{d_s}\left[\bar{\lambda}_O \left\|\boldsymbol{\beta}_{i,n}^{t'} - \boldsymbol{\beta}_{i,n}^t\right\|_2 + \left\|\boldsymbol{w}_O^{t'} \circ \dot{\sigma}_r\left(\boldsymbol{W}_V^{t'} \boldsymbol{X}_n^\top \boldsymbol{\beta}_{i,n}^{t'}\right) - \boldsymbol{w}_O^t \circ \dot{\sigma}_r\left(\boldsymbol{W}_V^t \boldsymbol{X}_n^\top \boldsymbol{\beta}_{i,n}^t\right)\right\|_2\right] .$$
$$\tag{17}$$

The term $\left\|\boldsymbol{\beta}_{i,n}^{t'} - \boldsymbol{\beta}_{i,n}^t\right\|_2$ can be bounded by Lemma 7 as follows:

$$\left\|\boldsymbol{\beta}_{i,n}^{t'} - \boldsymbol{\beta}_{i,n}^t\right\|_2 \leq 2\tau_0 C_x^2 d_s\left(\bar{\lambda}_Q \left\|\boldsymbol{W}_K^{t'} - \boldsymbol{W}_K^t\right\|_2 + \bar{\lambda}_K \left\|\boldsymbol{W}_Q^{t'} - \boldsymbol{W}_Q^t\right\|_2\right) \leq z \left\|\boldsymbol{\theta}^{t'} - \boldsymbol{\theta}^t\right\|_2 . \tag{18}$$

The term $\left\| \boldsymbol{w}_O^{t'} \circ \dot{\sigma}_r \left( \boldsymbol{W}_V^{t'} \boldsymbol{X}_n^\top \boldsymbol{\beta}_{i,n}^{t'} \right) - \boldsymbol{w}_O^t \circ \dot{\sigma}_r \left( \boldsymbol{W}_V^t \boldsymbol{X}_n^\top \boldsymbol{\beta}_{i,n}^t \right) \right\|_2$ can be bounded by the triangle inequality, Cauchy–Schwarz inequality, and the same method in Eq. (6) as follows:

$$
\begin{aligned}
&\left\| \boldsymbol{w}_O^{t'} \circ \dot{\sigma}_r \left( \boldsymbol{W}_V^{t'} \boldsymbol{X}_n^\top \boldsymbol{\beta}_{i,n}^{t'} \right) - \boldsymbol{w}_O^t \circ \dot{\sigma}_r \left( \boldsymbol{W}_V^t \boldsymbol{X}_n^\top \boldsymbol{\beta}_{i,n}^t \right) \right\|_2 \\
&\leq \left\| \boldsymbol{w}_O^{t'} \right\|_2 \left\| \dot{\sigma}_r \left( \boldsymbol{W}_V^{t'} \boldsymbol{X}_n^\top \boldsymbol{\beta}_{i,n}^{t'} \right) - \dot{\sigma}_r \left( \boldsymbol{W}_V \boldsymbol{X}_n^\top \boldsymbol{\beta}_{i,n}^t \right) \right\|_2 + \left\| \dot{\sigma}_r \left( \boldsymbol{W}_V^t \boldsymbol{X}_n^\top \boldsymbol{\beta}_{i,n}^t \right) \circ \left( \boldsymbol{w}_O^{t'} - \boldsymbol{w}_O^t \right) \right\|_2 \\
&\leq \bar{\lambda}_O \left\| \boldsymbol{W}_V^{t'} \boldsymbol{X}_n^\top \boldsymbol{\beta}_{i,n}^{t'} - \boldsymbol{W}_V^t \boldsymbol{X}_n^\top \boldsymbol{\beta}_{i,n}^t \right\|_2 + \left\| \boldsymbol{w}_O^{t'} - \boldsymbol{w}_O^t \right\|_2 \\
&\leq \bar{\lambda}_O C_x \sqrt{d_s} \left( \left\| \boldsymbol{W}_V^{t'} - \boldsymbol{W}_V^t \right\|_2 + \bar{\lambda}_V 2\tau_0 C_x^2 d_s \left( \bar{\lambda}_Q \left\| \boldsymbol{W}_K^{t'} - \boldsymbol{W}_K^t \right\|_2 + \bar{\lambda}_K \left\| \boldsymbol{W}_Q^{t'} - \boldsymbol{W}_Q^t \right\|_2 \right) \right) + \left\| \boldsymbol{w}_O^{t'} - \boldsymbol{w}_O^t \right\|_2 \\
&\leq \left[ \bar{\lambda}_O C_x \sqrt{d_s} (1 + \bar{\lambda}_V z) + 1 \right] \left\| \boldsymbol{\theta}^{t'} - \boldsymbol{\theta}^\top \right\|_2 .
\end{aligned}
\tag{19}
$$

Plugging Eq. (18) and Eq. (19) back Eq. (17), we obtain:

$$
\begin{aligned}
&\left\| \nabla_{\boldsymbol{W}_V} f(\boldsymbol{X}_n; \boldsymbol{\theta}^{t'}) - \nabla_{\boldsymbol{W}_V} f(\boldsymbol{X}_n; \boldsymbol{\theta}^t) \right\|_F \\
&\leq \tau_1 C_x d_s^{3/2} \left[ \bar{\lambda}_O z + \bar{\lambda}_O C_x \sqrt{d_s} (1 + \bar{\lambda}_V z) + 1 \right] \left\| \boldsymbol{\theta}^{t'} - \boldsymbol{\theta}^\top \right\|_2 .
\end{aligned}
\tag{20}
$$

Regarding the third term in Eq. (15), let us denote by:

$$
\boldsymbol{U}_{i,n}^{t'} = \boldsymbol{W}_K^{t'} \boldsymbol{X}_n^\top \left( \mathrm{diag}(\boldsymbol{\beta}_{i,n}^{t'}) - \boldsymbol{\beta}_{i,n}^{t'} \boldsymbol{\beta}_{i,n}^{t'\top} \right) \boldsymbol{X}_n, \quad \boldsymbol{h}_{i,n}^{t'} = \boldsymbol{W}_V^{t'\top} \left( \boldsymbol{w}_O^{t'} \circ \dot{\sigma}_r \left( \boldsymbol{W}_V^{t'} \boldsymbol{X}_n^\top \boldsymbol{\beta}_{i,n}^{t'} \right) \right) ,
$$

$$
\boldsymbol{U}_{i,n}^{t} = \boldsymbol{W}_K^{t} \boldsymbol{X}_n^\top \left( \mathrm{diag}(\boldsymbol{\beta}_{i,n}^{t}) - \boldsymbol{\beta}_{i,n}^{t} \boldsymbol{\beta}_{i,n}^{t\top} \right) \boldsymbol{X}_n, \quad \boldsymbol{h}_{i,n}^{t} = \boldsymbol{W}_V^{t\top} \left( \boldsymbol{w}_O^{t} \circ \dot{\sigma}_r \left( \boldsymbol{W}_V^{t} \boldsymbol{X}_n^\top \boldsymbol{\beta}_{i,n}^{t} \right) \right) .
$$

Then:

$$
\begin{aligned}
&\left\| \nabla_{\boldsymbol{W}_Q} f(\boldsymbol{X}_n; \boldsymbol{\theta}^{t'}) - \nabla_{\boldsymbol{W}_Q} f(\boldsymbol{X}_n; \boldsymbol{\theta}^t) \right\|_F = \tau_0 \tau_1 \left\| \sum_{i=1}^{d_s} \boldsymbol{U}_{i,n}^{t'} \boldsymbol{h}_{i,n}^{t'} \boldsymbol{X}_n^{(i,:)} - \sum_{i=1}^{d_s} \boldsymbol{U}_{i,n}^{t} \boldsymbol{h}_{i,n}^{t} \boldsymbol{X}_n^{(i,:)} \right\|_F \\
&\leq \tau_0 \tau_1 C_x \left\| \sum_{i=1}^{d_s} \boldsymbol{U}_{i,n}^{t'} \boldsymbol{h}_{i,n}^{t'} - \sum_{i=1}^{d_s} \boldsymbol{U}_{i,n}^{t} \boldsymbol{h}_{i,n}^{t} \right\|_2 \leq \tau_0 \tau_1 C_x \sum_{i=1}^{d_s} \left( \left\| \boldsymbol{U}_{i,n}^{t'} \right\|_2 \left\| \boldsymbol{h}_{i,n}^{t'} - \boldsymbol{h}_{i,n}^{t} \right\|_2 + \left\| \boldsymbol{h}_{i,n}^{t} \right\|_2 \left\| \boldsymbol{U}_{i,n}^{t'} - \boldsymbol{U}_{i,n}^{t} \right\|_2 \right) .
\end{aligned}
\tag{21}
$$

We bound each term separately:

$$
\left\| \boldsymbol{U}_{i,n}^{t'} \right\|_2 \leq \left\| \boldsymbol{W}_K^{t'} \right\|_2 \left\| \boldsymbol{X}_n \right\|_2^2 \left\| \mathrm{diag}(\boldsymbol{\beta}_{i,n}^{t'}) - \boldsymbol{\beta}_{i,n}^{t'} \boldsymbol{\beta}_{i,n}^{t'\top} \right\|_2 \leq 2 \bar{\lambda}_K d_s C_x^2 .
$$

$$
\left\| \boldsymbol{h}_{i,n}^{t} \right\|_2 \leq \left\| \boldsymbol{W}_V^{t} \right\|_2 \left\| \boldsymbol{w}_O^{t} \right\|_2 \leq \bar{\lambda}_V \bar{\lambda}_O .
$$

$$
\begin{aligned}
&\left\| \boldsymbol{h}_{i,n}^{t'} - \boldsymbol{h}_{i,n}^{t} \right\|_2 \leq \left\| \boldsymbol{W}_V^{t'} \right\|_2 \left\| \boldsymbol{w}_O^{t'} \circ \dot{\sigma}_r \left( \boldsymbol{W}_V^{t'} \boldsymbol{X}_n^\top \boldsymbol{\beta}_{i,n}^{t'} \right) - \boldsymbol{w}_O^{t} \circ \dot{\sigma}_r \left( \tilde{\boldsymbol{W}}_V^{t} \boldsymbol{X}_n^\top \boldsymbol{\beta}_{i,n}^{t} \right) \right\|_2 + \left\| \boldsymbol{W}_V^{t'} - \boldsymbol{W}_V^{t} \right\|_2 \left\| \boldsymbol{w}_O \right\|_2 \\
&\leq \left[ \bar{\lambda}_V \left( \bar{\lambda}_O C_x \sqrt{d_s} (1 + \bar{\lambda}_V z) + 1 \right) + \bar{\lambda}_O \right] \left\| \boldsymbol{\theta}^{t'} - \boldsymbol{\theta}^t \right\|_2 .
\end{aligned}
$$

where the second inequality uses the result in Eq. (19).

$$\left\| \boldsymbol{U}_{i,n}^{t'} - \boldsymbol{U}_{i,n}^{t} \right\|_2$$

$$\leq \left\| \boldsymbol{W}_K^{t'} \boldsymbol{X}_n^{\top} \left( \mathrm{diag}(\boldsymbol{\beta}_{i,n}^{t'}) - \boldsymbol{\beta}_{i,n}^{t'} \boldsymbol{\beta}_{i,n}^{t'\top} \right) - \boldsymbol{W}_K^{t} \boldsymbol{X}_n^{\top} \left( \mathrm{diag}(\boldsymbol{\beta}_{i,n}^{t}) - \boldsymbol{\beta}_{i,n}^{t} \boldsymbol{\beta}_{i,n}^{t\top} \right) \right\|_2 \|\boldsymbol{X}_n\|_2$$

$$\leq C_x \sqrt{d_s} \left[ \left\| \boldsymbol{W}_K^{t'} - \boldsymbol{W}_K^{t} \right\|_2 \|\boldsymbol{X}_n\|_2 \left\| \mathrm{diag}(\boldsymbol{\beta}_{i,n}^{t'}) - \boldsymbol{\beta}_{i,n}^{t'} \boldsymbol{\beta}_{i,n}^{t'\top} \right\|_2 \right.$$

$$\left. + \left\| \boldsymbol{W}_K^{t} \right\|_2 \|\boldsymbol{X}_n\|_2 \left\| \mathrm{diag}(\boldsymbol{\beta}_{i,n}^{t'}) - \boldsymbol{\beta}_{i,n}^{t'} \boldsymbol{\beta}_{i,n}^{t'\top} - \mathrm{diag}(\boldsymbol{\beta}_{i,n}^{t}) - \boldsymbol{\beta}_{i,n}^{t} \boldsymbol{\beta}_{i,n}^{t\top} \right\|_2 \right]$$

$$\leq C_x^2 d_s \left[ \left\| \boldsymbol{\theta}^{t'} - \boldsymbol{\theta}^{t} \right\|_2 + \bar{\lambda}_K \left( \left\| \mathrm{diag}(\boldsymbol{\beta}_{i,n}^{t'}) - \mathrm{diag}(\boldsymbol{\beta}_{i,n}^{t}) \right\|_2 + \left\| \boldsymbol{\beta}_{i,n}^{t'} \boldsymbol{\beta}_{i,n}^{t'\top} - \boldsymbol{\beta}_{i,n}^{t} \boldsymbol{\beta}_{i,n}^{t\top} \right\|_2 \right) \right] \quad (22)$$

$$\leq C_x^2 d_s \left[ \left\| \boldsymbol{\theta}^{t'} - \boldsymbol{\theta}^{t} \right\|_2 + \bar{\lambda}_K \left( \left\| \boldsymbol{\beta}_{i,n}^{t'} - \boldsymbol{\beta}_{i,n}^{t} \right\|_\infty + \left( \left\| \boldsymbol{\beta}_{i,n}^{t'} \right\|_2 + \|\boldsymbol{\beta}_{i,n}^{t}\|_2 \right) \left\| \boldsymbol{\beta}_{i,n}^{t'} - \boldsymbol{\beta}_{i,n}^{t} \right\|_2 \right) \right]$$

$$\leq C_x^2 d_s \left[ \left\| \boldsymbol{\theta}^{t'} - \boldsymbol{\theta}^{t} \right\|_2 + 3\bar{\lambda}_K \left\| \boldsymbol{\beta}_{i,n}^{t'} - \boldsymbol{\beta}_{i,n}^{t} \right\|_2 \right]$$

$$\leq C_x^2 d_s \left[ \left\| \boldsymbol{\theta}^{t'} - \boldsymbol{\theta}^{t} \right\|_2 + 3\bar{\lambda}_K 2\tau_0 C_x^2 d_s \left( \bar{\lambda}_Q \left\| \boldsymbol{W}_K^{t'} - \boldsymbol{W}_K^{t} \right\|_2 + \bar{\lambda}_K \left\| \boldsymbol{W}_Q^{t'} - \boldsymbol{W}_Q^{t} \right\|_2 \right) \right]$$

$$\leq \left[ C_x^2 d_s + 3\bar{\lambda}_K z \right] \left\| \boldsymbol{\theta}^{t'} - \boldsymbol{\theta}^{t} \right\|_2 ,$$

where the last second inequality is by Lemma 7. Plugging back Eq. (21), we obtain:

$$\left\| \nabla_{\boldsymbol{W}_Q} f(\boldsymbol{X}_n; \boldsymbol{\theta}^{t'}) - \nabla_{\boldsymbol{W}_Q} f(\boldsymbol{X}_n; \boldsymbol{\theta}^{t}) \right\|_F$$

$$\leq \tau_0 \tau_1 C_x d_s \left\{ 2\bar{\lambda}_K d_s C_x^2 \left[ \bar{\lambda}_V \left( \bar{\lambda}_O C_x \sqrt{d_s}(1 + \bar{\lambda}_V z) + 1 \right) + \bar{\lambda}_O \right] + \bar{\lambda}_V \bar{\lambda}_O [C_x^2 d_s + 3\bar{\lambda}_K z] \right\} \left\| \boldsymbol{\theta}^{t'} - \boldsymbol{\theta}^{t} \right\|_2 .$$

$$(23)$$

Similarly, the fourth term in Eq. (15) can be bounded by:

$$\left\| \nabla_{\boldsymbol{W}_K} f(\boldsymbol{X}_n; \boldsymbol{\theta}^{t'}) - \nabla_{\boldsymbol{W}_K} f(\boldsymbol{X}_n; \boldsymbol{\theta}^{t}) \right\|_F$$

$$\leq \tau_0 \tau_1 C_x d_s \left\{ 2\bar{\lambda}_Q d_s C_x^2 \left[ \bar{\lambda}_V \left( \bar{\lambda}_O C_x \sqrt{d_s}(1 + \bar{\lambda}_V z) + 1 \right) + \bar{\lambda}_O \right] + \bar{\lambda}_V \bar{\lambda}_O [C_x^2 d_s + 3\bar{\lambda}_K z] \right\} \left\| \boldsymbol{\theta}^{t'} - \boldsymbol{\theta}^{t} \right\|_2 .$$

$$(24)$$

Plugging the upper bound for these four terms back Eq. (15) finishes the proof. $\square$

## C.2 Proof of Proposition. 1

*Proof.* We can reformulate Eq. (1.2) as:

$$f(\boldsymbol{X}) = \tau_1 \boldsymbol{w}_O^{\top} \sum_{i=1}^{d_s} \sigma_r \left( \boldsymbol{W}_V \boldsymbol{X}^{\top} \boldsymbol{\beta}_i \right) = \boldsymbol{w}_O^{\top} \boldsymbol{f}_{\mathrm{pre}} ,$$

with $\boldsymbol{\beta}_i := \sigma_s \left( \tau_0 \boldsymbol{X}^{(i,:)} \boldsymbol{W}_Q^{\top} \boldsymbol{W}_K \boldsymbol{X}^{\top} \right)^{\top} \in \mathbb{R}^{d_s}$. We show by induction for every $t \geq 0$

$$\begin{cases} \left\| \boldsymbol{W}_Q^s \right\|_2 \leq \bar{\lambda}_Q, \|\boldsymbol{W}_K^s\|_2 \leq \bar{\lambda}_K, & s \in [0, t], \\ \left\| \boldsymbol{W}_V^s \right\|_2 \leq \bar{\lambda}_V, \|\boldsymbol{w}_O^s\|_2 \leq \bar{\lambda}_O, & s \in [0, t], \\ \sigma_{\min} \left( \boldsymbol{F}_{\mathrm{pre}}^s \right) \geq \frac{1}{2}\alpha, & s \in [0, t], \\ \ell(\boldsymbol{\theta}^s) \leq \left( 1 - \gamma \frac{\alpha^2}{2} \right)^s \ell(\boldsymbol{\theta}^0), & s \in [0, t]. \end{cases} \quad (25)$$

It is clear that Eq. (25) holds for $t = 0$. Assume that Eq. (25) holds up to iteration $t$. By the triangle inequality, we have

$$\left\| \boldsymbol{W}_Q^{t+1} - \boldsymbol{W}_Q^0 \right\|_F \leq \sum_{s=0}^{t} \left\| \boldsymbol{W}_Q^{s+1} - \boldsymbol{W}_Q^s \right\|_F = \gamma \sum_{s=0}^{t} \left\| \nabla_{\boldsymbol{W}_Q} \ell(\boldsymbol{\theta}^s) \right\|_F$$

$$\leq 2\gamma\rho\tau_0 \bar{\lambda}_K \bar{\lambda}_V \bar{\lambda}_O d_s C_x^2 \sum_{s=0}^{t} \sqrt{2\ell(\boldsymbol{\theta}^s)} \leq 2\gamma\rho\tau_0 \bar{\lambda}_K \bar{\lambda}_V \bar{\lambda}_O d_s C_x^2 \sum_{s=0}^{t} \left( 1 - \gamma \frac{\alpha^2}{2} \right)^{s/2} \sqrt{2\ell(\boldsymbol{\theta}^0)} ,$$

$$(26)$$

where the second inequality follows from the upper bound of the gradient norm in Lemma 12, and the last inequality follows from induction assumption. Let $u := \sqrt{1 - \gamma\alpha^2/2}$, we bound the RHS of the previous expression:

$$\rho 2\tau_0 \bar{\lambda}_K \bar{\lambda}_V \bar{\lambda}_O d_s C_x^2 \frac{2}{\alpha^2}(1-u^2)\frac{1-u^{t+1}}{1-u}\sqrt{2\ell(\boldsymbol{\theta}^0)} \tag{27}$$

$$\leq \rho 2\tau_0 \bar{\lambda}_K \bar{\lambda}_V \bar{\lambda}_O d_s C_x^2 \frac{4}{\alpha^2}\sqrt{2\ell(\boldsymbol{\theta}^0)}, \quad \text{(since } u \in (0,1)) \tag{28}$$

$$\leq C_Q. \quad \text{(by Eq. (2))}. \tag{29}$$

By Weyl's inequality, this implies:

$$\left\|\boldsymbol{W}_Q^{t+1}\right\|_2 \leq \left\|\boldsymbol{W}_Q^0\right\|_2 + C_Q = \bar{\lambda}_Q. \tag{30}$$

Similarly,

$$\left\|\boldsymbol{w}_O^{t+1} - \boldsymbol{w}_O^0\right\|_2 \leq \sum_{s=0}^{t}\left\|\boldsymbol{w}_O^{s+1} - \boldsymbol{w}_O^s\right\|_2 = \gamma\sum_{s=0}^{t}\left\|\nabla_{\boldsymbol{w}_O}\Phi(\boldsymbol{W}^s)\right\|_2 \leq \gamma\rho\bar{\lambda}_V\sum_{s=0}^{t}\sqrt{2\ell(\boldsymbol{\theta}^s)}$$

$$\leq \gamma\rho\bar{\lambda}_V\sum_{s=0}^{t}\left(1 - \gamma\frac{\alpha^2}{2}\right)^{s/2}\sqrt{2\ell(\boldsymbol{\theta}^0)} \leq \rho\bar{\lambda}_V\frac{2}{\alpha^2}(1-u^2)\frac{1-u^{t+1}}{1-u}\sqrt{2\ell(\boldsymbol{\theta}^0)} \leq \rho\bar{\lambda}_V\frac{4}{\alpha^2}\sqrt{2\ell(\boldsymbol{\theta}^0)} \leq C_O. \tag{31}$$

By Weyl's inequality, this implies

$$\left\|\boldsymbol{w}_O^{t+1}\right\|_2 \leq \left\|\boldsymbol{w}_O^0\right\|_2 + C_O = \bar{\lambda}_O. \tag{32}$$

Similarly, we can obtain that

$$\left\|\boldsymbol{W}_K^{t+1} - \boldsymbol{W}_K^0\right\|_{\mathrm{F}} \leq 2\rho\tau_0 \bar{\lambda}_Q \bar{\lambda}_V \bar{\lambda}_O d_s C_x^2 \frac{4}{\alpha^2}\sqrt{2\ell(\boldsymbol{\theta}^0)}, \quad \left\|\boldsymbol{W}_K^{t+1}\right\|_2 \leq \bar{\lambda}_K, \tag{33}$$

$$\left\|\boldsymbol{W}_V^{t+1} - \boldsymbol{W}_V^0\right\|_{\mathrm{F}} \leq \rho\bar{\lambda}_O \frac{4}{\alpha^2}\sqrt{2\ell(\boldsymbol{\theta}^0)}, \quad \left\|\boldsymbol{W}_V^{t+1}\right\|_2 \leq \bar{\lambda}_V. \tag{34}$$

Next, we will prove the fifth inequality in Eq. (25):

$$\left\|\boldsymbol{F}_{\mathrm{pre}}^{t+1} - \boldsymbol{F}_{\mathrm{pre}}^0\right\|_{\mathrm{F}}$$

$$\leq \rho\left\{\left\|\boldsymbol{W}_V^{t+1} - \boldsymbol{W}_V^0\right\|_2 + \left\|\boldsymbol{W}_V^0\right\|_2 2\tau_0 C_x^2\left(\left\|\boldsymbol{W}_Q^{t+1}\right\|_2\left\|\boldsymbol{W}_K^{t+1} - \boldsymbol{W}_K^0\right\|_2 + \left\|\boldsymbol{W}_K^0\right\|_2\left\|\boldsymbol{W}_Q^{t+1} - \boldsymbol{W}_Q^0\right\|_2\right)\right\}$$

$$\leq \sum_{s=0}^{t}\gamma\rho\left\{\rho\bar{\lambda}_O\sqrt{2\ell(\boldsymbol{\theta}^s)} + \bar{\lambda}_V 2\tau_0 C_x^2 d_s[2\rho\tau_0(\bar{\lambda}_K^2 + \bar{\lambda}_Q^2)\bar{\lambda}_V\bar{\lambda}_O C_x^2 d_s\sqrt{2\ell(\boldsymbol{\theta}^s)}]\right\} \quad \text{(by Lemma 12)}$$

$$\leq \rho^2\bar{\lambda}_O\left[1 + 4\tau_0^2 C_x^4 d_s^2 \bar{\lambda}_V^2\left(\bar{\lambda}_Q^2 + \bar{\lambda}_K^2\right)\right]\frac{16}{\alpha^2}\sqrt{2\ell(\boldsymbol{\theta}^0)}$$

$$= \rho^2\frac{z}{\bar{\lambda}_O}\frac{16}{\alpha^2}\sqrt{2\ell(\boldsymbol{\theta}^0)}$$

$$\leq \frac{1}{2}\alpha, \quad \text{(by Eq. (3))}$$

where the first inequality holds by Lemma 9. This result further implies that $\sigma_{\min}\left(\boldsymbol{F}_{\mathrm{pre}}^{t+1}\right) \geq \frac{1}{2}\alpha$ by Weyl's inequality. It remains to prove the last inequality in Eq. (25) holds for step $t+1$.

We start by proving the Lipschitz constant for the gradient of the loss restricted to the interval $[\boldsymbol{\theta}^\top, \boldsymbol{\theta}^{t+1}]$. We define $\boldsymbol{\theta}^{t+\phi} := \boldsymbol{\theta}^\top + \phi(\boldsymbol{\theta}^{t+1} - \boldsymbol{\theta}^\top)$, for $\phi \in [0,1]$. Then, by triangle inequality and Cauchy–Schwarz inequality, we have:

$$\left\|\nabla_{\boldsymbol{\theta}}\ell(\boldsymbol{\theta}^{t+\phi}) - \nabla_{\boldsymbol{\theta}}\ell(\boldsymbol{\theta}^\top)\right\|_2$$

$$= \left\|\nabla_{\boldsymbol{\theta}}\boldsymbol{f}^{t+\phi}\cdot(\boldsymbol{f}^{t+\phi} - \boldsymbol{y}) - \nabla_{\boldsymbol{\theta}}\boldsymbol{f}^t\cdot(\boldsymbol{f}^t - \boldsymbol{y})\right\|_2$$

$$\leq \left\|\boldsymbol{f}^{t+\phi} - \boldsymbol{f}^t\right\|_2\left\|\nabla_{\boldsymbol{\theta}}\boldsymbol{f}^{t+\phi}\right\|_2 + \left\|\nabla_{\boldsymbol{\theta}}\boldsymbol{f}^{t+\phi} - \nabla_{\boldsymbol{\theta}}\boldsymbol{f}^t\right\|_2\left\|\boldsymbol{f}^t - \boldsymbol{y}\right\|_2 \tag{35}$$

$$\leq \left\|\boldsymbol{f}^{t+\phi} - \boldsymbol{f}^t\right\|_2\left\|\nabla_{\boldsymbol{\theta}}\boldsymbol{f}^{t+\phi}\right\|_2 + 2\left\|\nabla_{\boldsymbol{\theta}}\boldsymbol{f}^{t+\phi} - \nabla_{\boldsymbol{\theta}}\boldsymbol{f}^t\right\|_2\ell(\boldsymbol{\theta}^0),$$

where the last inequality is by induction rule. Then, we bound each term separately. Note that:

$$\left\|\boldsymbol{W}_Q^{t+\phi} - \boldsymbol{W}_Q^0\right\|_{\mathrm{F}} \le \left\|\boldsymbol{W}_Q^{t+\phi} - \boldsymbol{W}_Q^t\right\|_{\mathrm{F}} + \sum_{s=0}^{t-1} \left\|\boldsymbol{W}_Q^{s+1} - \boldsymbol{W}_Q^s\right\|_{\mathrm{F}}$$

$$= \phi\gamma \left\|\nabla_{\boldsymbol{W}_Q}\ell(\boldsymbol{\theta}^\top)\right\|_{\mathrm{F}} + \gamma \sum_{s=0}^{t-1} \left\|\nabla_{\boldsymbol{W}_Q}\ell(\boldsymbol{\theta}^s)\right\|_{\mathrm{F}} \le \gamma \sum_{s=0}^{t} \left\|\nabla_{\boldsymbol{W}_Q}\ell(\boldsymbol{\theta}^s)\right\|_{\mathrm{F}} .$$

Then following exact the same step as in Eqs. (26), (27) and (30), we have: $\left\|\boldsymbol{W}_Q^{t+\phi}\right\|_2 \le \bar{\lambda}_Q$. By the same method, we have: $\left\|\boldsymbol{W}_K^{t+\phi}\right\|_2 \le \bar{\lambda}_K, \left\|\boldsymbol{W}_V^{t+\phi}\right\|_2 \le \bar{\lambda}_V, \left\|\boldsymbol{w}_O^{t+\phi}\right\|_2 \le \bar{\lambda}_O$. Now we proceed to bound the first term in Eq. (35).

$$\left\|\boldsymbol{f}^{t+\phi} - \boldsymbol{f}^t\right\|_2$$
$$\le \rho \left\|\boldsymbol{W}_V^{t+\phi}\right\|_2 \left\|\boldsymbol{w}_O^{t+\phi} - \boldsymbol{w}_O^t\right\|_2 + \left\|\boldsymbol{w}_O^t\right\|_2 \left\|\boldsymbol{F}_{\mathrm{pre}}^{t+\phi} - \boldsymbol{F}_{\mathrm{pre}}^t\right\|_2$$
$$\le \rho \left\|\boldsymbol{W}_V^{t+\phi}\right\|_2 \left\|\boldsymbol{w}_O^{t+\phi} - \boldsymbol{w}_O^t\right\|_2$$
$$\quad + \rho \left\|\boldsymbol{w}_O^t\right\|_2 \left(\left\|\boldsymbol{W}_V^{t+\phi} - \boldsymbol{W}_V^t\right\|_2 + \left\|\boldsymbol{W}_V^t\right\|_2 2\tau_0 C_x^2 d_s \left(\left\|\boldsymbol{W}_Q^{t+\phi}\right\|_2 \left\|\boldsymbol{W}_K^{t+\phi} - \boldsymbol{W}_K^t\right\|_2 + \left\|\boldsymbol{W}_K^t\right\|_2 \left\|\boldsymbol{W}_Q^{t+\phi} - \boldsymbol{W}_Q^t\right\|_2\right)\right)$$
$$\le \rho\bar{\lambda}_V \left\|\boldsymbol{\theta}^{t+\phi} - \boldsymbol{\theta}^\top\right\|_2 + \rho\bar{\lambda}_O (1 + \bar{\lambda}_V 2\tau_0 C_x^2 d_s (\bar{\lambda}_Q + \bar{\lambda}_K)) \left\|\boldsymbol{\theta}^{t+\phi} - \boldsymbol{\theta}^\top\right\|_2$$
$$= \rho(\bar{\lambda}_V + \bar{\lambda}_O + \bar{\lambda}_V 2\tau_0 C_x^2 d_s (\bar{\lambda}_Q + \bar{\lambda}_K)) \left\|\boldsymbol{\theta}^{t+\phi} - \boldsymbol{\theta}^\top\right\|_2$$
$$\triangleq c_1 \left\|\boldsymbol{\theta}^{t+\phi} - \boldsymbol{\theta}^\top\right\|_2 ,$$

(36)

where the first inequality is by Lemma 10, the second inequality is by Lemma 9. Next, the second term in Eq. (35) can be bounded by Lemma 13. The third term in Eq. (35) can be bounded by Lemma 14. As a result, Eq. (35) has the following upper bound:

$$\left\|\nabla_{\boldsymbol{\theta}}\ell(\boldsymbol{\theta}^{t+\phi}) - \nabla_{\boldsymbol{\theta}}\ell(\boldsymbol{\theta}^\top)\right\|_2 \le c_1 c_2 \left\|\boldsymbol{\theta}^{t+\phi} - \boldsymbol{\theta}^\top\right\|_2 + 2c_3\ell(\boldsymbol{\theta}^0) \left\|\boldsymbol{\theta}^{t+\phi} - \boldsymbol{\theta}^\top\right\|_2 \triangleq C \left\|\boldsymbol{\theta}^{t+\phi} - \boldsymbol{\theta}^\top\right\|_2 ,$$

(37)

where $c_1, c_2, c_3$ are defined at Eqs. (11), (14) and (36), and we further define $C \triangleq c_1 c_2 + 2c_3\ell(\boldsymbol{\theta}^0)$. Lastly, by applying Lemma 4.3 in Nguyen and Mondelli [2020] and Eq. (37), we have:

$$\ell(\boldsymbol{\theta}^{t+1}) \le \ell(\boldsymbol{\theta}^\top) + \langle\nabla_{\boldsymbol{\theta}}\ell(\boldsymbol{\theta}^\top), \boldsymbol{\theta}^{t+1} - \boldsymbol{\theta}^\top\rangle + \frac{C}{2} \left\|\boldsymbol{\theta}^{t+1} - \boldsymbol{\theta}^\top\right\|_{\mathrm{F}}^2$$

$$= \ell(\boldsymbol{\theta}^\top) - \gamma \left\|\nabla_{\boldsymbol{\theta}}\ell(\boldsymbol{\theta}^\top)\right\|_{\mathrm{F}}^2 + \frac{C}{2}\gamma^2 \left\|\nabla_{\boldsymbol{\theta}}\ell(\boldsymbol{\theta}^\top)\right\|_{\mathrm{F}}^2$$

$$\le \ell(\boldsymbol{\theta}^\top) - \frac{1}{2}\gamma \left\|\nabla_{\boldsymbol{\theta}}\ell(\boldsymbol{\theta}^\top)\right\|_{\mathrm{F}}^2 \quad \text{(By the condition on } \gamma)$$

$$\le \ell(\boldsymbol{\theta}^\top) - \frac{1}{2}\gamma \left\|\nabla_{\boldsymbol{w}_o}\ell(\boldsymbol{\theta}^\top)\right\|_2^2$$

$$= \ell(\boldsymbol{\theta}^\top) - \frac{1}{2}\gamma \left\|(\boldsymbol{F}_{\mathrm{pre}}^t)^\top(\boldsymbol{f}^t - \boldsymbol{y})\right\|_2^2$$

$$\le \ell(\boldsymbol{\theta}^\top) - \frac{1}{2}\gamma(\sigma_{\min}(\boldsymbol{F}_{\mathrm{pre}}^t))^2 \left\|\boldsymbol{f}^t - \boldsymbol{y}\right\|_2^2$$

$$\le (1 - \gamma\frac{\alpha}{2})\ell(\boldsymbol{\theta}^\top) \quad \text{(By induction assumption)},$$

which concludes the proof.

$\square$

## C.3 Proof of Theorem 1

**Lemma 15.** *Let* $\boldsymbol{\Phi} = [\boldsymbol{X}_1^\top \boldsymbol{\beta}_{1,1}, ..., \boldsymbol{X}_N^\top \boldsymbol{\beta}_{1,N}]^\top \in \mathbb{R}^{N \times d}$, *where* $\boldsymbol{\beta}_{1,n} = \sigma_s\left(\tau_0 \boldsymbol{X}_n^{(1,:)} \boldsymbol{W}_Q^\top \boldsymbol{W}_K \boldsymbol{X}_n^\top\right)^\top, n \in [N]$, *then under Assumptions 2 and 3, with probability at*

*least* $1 - \exp\left(-\Omega((N-1)^{-\hat{c}} d_s^{-1})\right)$, *one has:* $\lambda_0 := \lambda_{\min}\left(\mathbb{E}_{\boldsymbol{w} \sim \mathcal{N}(0, \eta_V \mathbb{1}_d)}[\sigma_r(\boldsymbol{\Phi}\boldsymbol{w})\sigma_r(\boldsymbol{\Phi}\boldsymbol{w})^T]\right) \geq$
$\Theta(\eta_V/d_s)$.

*Proof.* Due to Assumption 2, for any data $\boldsymbol{X}$, the matrix $\boldsymbol{X}\boldsymbol{X}^\top$ is positive definite and thus has positive minimum eigenvalue. We denote it as $\lambda_{\min}(\boldsymbol{X}\boldsymbol{X}^\top) \geq C_\lambda$.

According to [Nguyen et al., 2021, Lemma 5.3], using the Hermite expansion of $\sigma_r$, one has:

$$\lambda_0 \geq \eta_V \mu(\sigma_r)^2 \lambda_{\min}(\boldsymbol{\Phi}\boldsymbol{\Phi}^\top), \tag{38}$$

where $\mu(\sigma_r)$ is the 1-st Hermite coefficient of ReLU satisfying $\mu(\sigma_r) > 0$.

Now we proceed to provide a lower bound for $\lambda_{\min}(\boldsymbol{\Phi}\boldsymbol{\Phi}^\top)$. For notational simplicity, define:

$$\boldsymbol{B}_{ij} = \boldsymbol{\beta}_{1,i}\boldsymbol{\beta}_{1,j}^\top \in \mathbb{R}^{d_s \times d_s}, \boldsymbol{C}_{ij} = \boldsymbol{X}_i\boldsymbol{X}_j^\top \in \mathbb{R}^{d_s \times d_s}.$$

Then we can rewrite $\boldsymbol{\Phi}\boldsymbol{\Phi}^\top$ as follows:

$$\boldsymbol{\Phi}\boldsymbol{\Phi}^\top = \begin{bmatrix} \mathrm{Trace}(\boldsymbol{B}_{11}^\top \boldsymbol{C}_{11}) & \mathrm{Trace}(\boldsymbol{B}_{12}^\top \boldsymbol{C}_{12}) & \cdots & \mathrm{Trace}(\boldsymbol{B}_{1N}^\top \boldsymbol{C}_{1N}) \\ \mathrm{Trace}(\boldsymbol{B}_{21}^\top \boldsymbol{C}_{21}) & \mathrm{Trace}(\boldsymbol{B}_{22}^\top \boldsymbol{C}_{22}) & \cdots & \mathrm{Trace}(\boldsymbol{B}_{2N}^\top \boldsymbol{C}_{2N}) \\ \vdots & \vdots & \vdots & \vdots \\ \mathrm{Trace}(\boldsymbol{B}_{N1}^\top \boldsymbol{C}_{N1}) & \mathrm{Trace}(\boldsymbol{B}_{N2}^\top \boldsymbol{C}_{N2}) & \cdots & \mathrm{Trace}(\boldsymbol{B}_{NN}^\top \boldsymbol{C}_{NN}) \end{bmatrix}.$$

By Gershgorin circle theorem [Gershgorin, 1931], there exists $k \in [N]$ such that:

$$\lambda_{\min}(\boldsymbol{\Phi}\boldsymbol{\Phi}^\top) \geq \mathrm{Trace}(\boldsymbol{B}_{kk}^\top \boldsymbol{C}_{kk}) - \sum_{j \neq k} \mathrm{Trace}(\boldsymbol{B}_{kj}^\top \boldsymbol{C}_{kj}). \tag{39}$$

Using Von Neumann's trace inequality [Mirsky, 1975] and noting that $\boldsymbol{B}_{kj}$ is a rank one matrix, one has:

$$\mathrm{Trace}(\boldsymbol{B}_{kj}^\top \boldsymbol{C}_{kj}) \leq \sigma_{\max}(\boldsymbol{B}_{kj})\sigma_{\max}(\boldsymbol{C}_{kj}) = \|\boldsymbol{\beta}_{1,k}\|_2 \|\boldsymbol{\beta}_{1,j}\|_2 \sqrt{\lambda_{\max}(\boldsymbol{C}_{kj}\boldsymbol{C}_{kj}^\top)}$$
$$\leq \|\boldsymbol{\beta}_{1,k}\|_2 \sqrt{\mathrm{Trace}(\boldsymbol{C}_{kj}\boldsymbol{C}_{kj}^\top)} = \|\boldsymbol{\beta}_{1,k}\|_2 \sqrt{\langle \boldsymbol{X}_k^\top \boldsymbol{X}_k, \boldsymbol{X}_j^\top \boldsymbol{X}_j \rangle}, \tag{40}$$

where we use the definition of the inner product between two matrices and $\|\boldsymbol{\beta}_{1,j}\|_2 \leq 1$. By Assumption 2, we have $\lambda_{\min}(\boldsymbol{X}_k \boldsymbol{X}_k^\top) \geq C_\lambda$, where $C_\lambda$ is some positive constant. By setting $t := \|\boldsymbol{\beta}_{1,k}\|_2^2 C_\lambda^2 / (N-1)^2$ in Assumption 3, with probability at least $1 - \exp\left(-\Omega((N-1)^{-\hat{c}})\right)$, one has

$$\sqrt{\langle \boldsymbol{X}_k^\top \boldsymbol{X}_k, \boldsymbol{X}_j^\top \boldsymbol{X}_j \rangle} \leq \|\boldsymbol{\beta}_{1,k}\|_2 C_\lambda (N-1)^{-1}, \quad \forall j \neq k.$$

Plugging back Eq. (39) and Eq. (40), we obtain:

$$\lambda_{\min}(\boldsymbol{\Phi}\boldsymbol{\Phi}^\top) \geq \mathrm{Trace}(\boldsymbol{B}_{kk}^\top \boldsymbol{C}_{kk}) - C_\lambda \|\boldsymbol{\beta}_{1,k}\|_2^2 \geq \lambda_{\min}(\boldsymbol{C}_{kk})\mathrm{Trace}(\boldsymbol{B}_{kk}) - C_\lambda \|\boldsymbol{\beta}_{1,k}\|_2^2$$
$$= \lambda_{\min}(\boldsymbol{X}_k \boldsymbol{X}_k^\top) \|\boldsymbol{\beta}_{1,k}\|_2^2 - C_\lambda \|\boldsymbol{\beta}_{1,k}\|_2^2 \geq \Theta(\|\boldsymbol{\beta}_{1,k}\|_2^2) \geq \Theta(1/d_s), \tag{41}$$

where the last inequality is by the lower bound of $\boldsymbol{\beta}_{1,k}$ in Lemma 8. Lastly, plugging the lower bound of $\lambda_{\min}(\boldsymbol{\Phi}\boldsymbol{\Phi}^\top)$ back Eq. (38) finishes the proof.

$\square$

**Remark:** The estimation of $\lambda_0$ is actually tight because its upper bound is also in a constant order. To be specific, denote $\boldsymbol{G} := \sigma_r(\boldsymbol{\Phi}\boldsymbol{w})\sigma_r(\boldsymbol{\Phi}\boldsymbol{w})^\top$, we have

$$\lambda_0 := \lambda_{\min}(\mathbb{E}_{\boldsymbol{w}}\boldsymbol{G}) \leq \frac{\mathrm{tr}(\mathbb{E}_{\boldsymbol{w}}\boldsymbol{G})}{N} = \frac{\sum_{n=1}^N \mathbb{E}_{\boldsymbol{w}}[\sigma_r(\boldsymbol{\Phi}^{(n,:)}\boldsymbol{w})]^2}{N}, \tag{42}$$

where $\boldsymbol{\Phi}^{(n,:)} = \boldsymbol{\beta}_{1,n}^\top \boldsymbol{X}_n$. Next, by Liao and Couillet [2018] (Sec.A in Supplementary Material):

$$\mathbb{E}_{\boldsymbol{w}}[\sigma_r(\boldsymbol{\Phi}^{(n,:)}\boldsymbol{w})]^2 = \frac{\eta_V}{2\pi} \left\|\boldsymbol{\Phi}^{(n,:)}\right\|^2 \arccos(-1) = \frac{\left\|\boldsymbol{\Phi}^{(n,:)}\right\|^2}{2}. \tag{43}$$

Combine Eq. (42) and Eq. (43), we have

$$\lambda_0 \leq \frac{\sum_{n=1}^{N} \eta_V \|\mathbf{\Phi}^{(n,:)}\|_2^2}{2N} \leq \eta_V d_s C_x^2 \leq \mathcal{O}(1) \,.$$

That means, our estimation on $\lambda_0$ is tight as its upper and lower bounds match with each other.

Now we are ready to present the proof for LeCun initialization under $\tau_0 = d_m^{-1/2}$ scaling.

*Proof.* We select $C_Q = C_K = 1 = C_V = C_O = 1$, then by Lemma 5, with probability at least $1 - 8e^{-d_m/2}$, we have:

$$\begin{aligned}
\bar{\lambda}_V &= \mathcal{O}(\sqrt{d_m/d}), \quad \bar{\lambda}_O = \mathcal{O}(1) \,, \\
\bar{\lambda}_Q &= \mathcal{O}(\sqrt{d_m/d}), \quad \bar{\lambda}_K = \mathcal{O}(\sqrt{d_m/d}) \,.
\end{aligned} \tag{44}$$

Plugging Eq. (44) into Eqs. (2) and (3), it suffices to prove the following equations.

$$\alpha^2 \geq \mathcal{O}\left(\sqrt{N} d_s^{3/2} C_x \sqrt{d_m/d}\right) \sqrt{2\ell(\boldsymbol{\theta}^0)} \tag{45}$$

$$\alpha^3 \geq \mathcal{O}\left(N d_s^3 C_x^2 (1 + 4C_x^4 d_s^2 d_m d^{-2})\right) \sqrt{2\ell(\boldsymbol{\theta}^0)} \tag{46}$$

Next, we will provide the lower bound for $\alpha^2 = \lambda_{\min}((\boldsymbol{F}_{\text{pre}}^0)(\boldsymbol{F}_{\text{pre}}^0)^\top)$. In the following context, we hide the index 0 for simplification. One can note that $\boldsymbol{F}_{\text{pre}}\boldsymbol{F}_{\text{pre}}^\top$ is the summation of PSD matrices, thus, it suffices to lower bound: $\lambda_{\min}(\hat{\boldsymbol{F}}_{\text{pre}}\hat{\boldsymbol{F}}_{\text{pre}}^\top)$, where we introduce the following notations:

$$\begin{aligned}
\hat{\boldsymbol{F}}_{\text{pre}}\hat{\boldsymbol{F}}_{\text{pre}}^\top &= \tau_1^2 \sigma_r(\boldsymbol{\Phi}\boldsymbol{W}_v^\top)\sigma_r(\boldsymbol{\Phi}\boldsymbol{W}_v^\top)^\top \\
\boldsymbol{\Phi} &= [\boldsymbol{X}_1^\top \boldsymbol{\beta}_{1,1}, ..., \boldsymbol{X}_N^\top \boldsymbol{\beta}_{1,N}]^\top \\
\boldsymbol{\beta}_{1,n} &= \sigma_s\left(\tau_0 \boldsymbol{X}_n^{(1,:)} \boldsymbol{W}_Q^\top \boldsymbol{W}_K \boldsymbol{X}_n^\top\right)^\top \quad n \in [N].
\end{aligned} \tag{47}$$

By Matrix-Chernoff inequality, we can obtain that (e.g. Lemma 5.2 of Nguyen et al. [2021]) w.p at least $1 - \delta_1$,

$$\lambda_{\min}(\hat{\boldsymbol{F}}_{\text{pre}}\hat{\boldsymbol{F}}_{\text{pre}}^\top) \geq d_m \lambda_0 / 4, \tag{48}$$

as long as it holds $d_m \geq \tilde{\Omega}(N/\lambda_0)$, where $\lambda_0 = \lambda_{\min}\left(\mathbb{E}_{\boldsymbol{w} \sim \mathcal{N}(0, \eta_V \mathbb{I}_d)}[\sigma_r(\boldsymbol{\Phi}\boldsymbol{w})\sigma_r(\boldsymbol{\Phi}\boldsymbol{w})^\top]\right)$, and $\tilde{\Omega}$ hides logarithmic factors depending on $\delta_1$. Lastly, w.p. at least $1 - \delta_2$, one has $\sqrt{2\ell(\boldsymbol{\theta}^0)} \leq \tilde{\mathcal{O}}(\sqrt{N})$. Plugging back Eqs. (2) and (3), it suffices to prove the following inequality.

$$d_m \lambda_0 / 4 \geq \tilde{\mathcal{O}}(N d_s^{3/2} C_x \sqrt{d_m/d}) \,, \tag{49}$$

$$(d_m \lambda_0 / 4)^{3/2} \geq \tilde{\mathcal{O}}(N^{3/2} d_s^3 C_x^2 (1 + 4C_x^4 d_s^2 d_m d^{-2})) \,, \tag{50}$$

By Lemma 15, with probability at least $1 - \exp\left(-\Omega((N-1)^{-\hat{c}} d_s^{-1})\right)$, one has $\lambda_0 \geq \Theta(\eta_V/d_s) = \Theta(d^{-1} d_s^{-1})$. Thus, when $d_m \geq \tilde{\Omega}(N^3)$, all of the above conditions hold. As a result, the conditions in Eqs. (45) and (46) are satisfied and the convergence of training Transformer is guaranteed as in Eq. (4). Note that one can achieve the same width requirement and probability for He initialization, and the proof bears resemblance to the LeCun initialization.

For the proof under LeCun initialization and $\tau_0 = d_m^{-1}$ scaling, we follow the same strategy. Specifically: As the same in the proof with $\tau_0 = d_m^{-1/2}$ scaling, we select $C_Q = C_K = C_V = C_O = 1$, then plugging Eq. (44) into Eqs. (2) and (3), it suffices to prove the following equations.

$$\alpha^2 \geq \mathcal{O}\left(\sqrt{N} d_s^{3/2} C_x \sqrt{d_m/d}\right) \sqrt{2\ell(\boldsymbol{\theta}^0)} \,, \tag{51}$$

$$\alpha^3 \geq \mathcal{O}\left(N d_s^3 C_x^2 (1 + 4C_x^4 d_s^2 d^{-2})\right) \sqrt{2\ell(\boldsymbol{\theta}^0)} \,. \tag{52}$$

Next, we will provide the lower bound for $\alpha^2 = \lambda_{\min}((\boldsymbol{F}_{\text{pre}}^0)(\boldsymbol{F}_{\text{pre}}^0)^\top)$. In the following context, we hide the index 0 for simplification. One can note that $\boldsymbol{F}_{\text{pre}}\boldsymbol{F}_{\text{pre}}^\top$ is the summation of PSD matrices,

thus, it suffices to lower bound: $\lambda_{\min}(\hat{\boldsymbol{F}}_{\text{pre}}\hat{\boldsymbol{F}}_{\text{pre}}^{\top})$, where we introduce the following notations:

$$
\begin{aligned}
\hat{\boldsymbol{F}}_{\text{pre}}\hat{\boldsymbol{F}}_{\text{pre}}^{\top} &= \tau_1^2 \sigma_r(\boldsymbol{\Phi}\boldsymbol{W}_v^{\top})\sigma_r(\boldsymbol{\Phi}\boldsymbol{W}_v^{\top})^{\top}, \\
\boldsymbol{\Phi} &= [\boldsymbol{X}_1^{\top}\boldsymbol{\beta}_{1,1}, ..., \boldsymbol{X}_N^{\top}\boldsymbol{\beta}_{1,N}]^{\top}, \\
\boldsymbol{\beta}_{1,n} &= \sigma_s\left(\tau_0 \boldsymbol{X}_n^{(1,:)}\boldsymbol{W}_Q^{\top}\boldsymbol{W}_K\boldsymbol{X}_n^{\top}\right)^{\top} \quad n \in [N].
\end{aligned}
\tag{53}
$$

By Eq. (48) w.p at least $1 - \delta_1$, $\lambda_{\min}(\hat{\boldsymbol{F}}_{\text{pre}}\hat{\boldsymbol{F}}_{\text{pre}}^{\top}) \geq d_m\lambda_0/4,$, as long as it holds $d_m \geq \tilde{\Omega}(N/\lambda_0)$, where $\lambda_0 = \lambda_{\min}\left(\mathbb{E}_{\boldsymbol{w} \sim \mathcal{N}(0, \eta_V \mathbb{I}_d)}[\sigma_r(\boldsymbol{\Phi}\boldsymbol{w})\sigma_r(\boldsymbol{\Phi}\boldsymbol{w})^{\top}]\right)$, and $\tilde{\Omega}$ hides logarithmic factors depending on $\delta_1$. Lastly, note that the activation function in the output layer $\sigma_r$ is 1-Lipschitz and is applied to $\sigma_r\left(\boldsymbol{W}_V\boldsymbol{X}^{\top}\boldsymbol{\beta}_i\right)$, where $\boldsymbol{X}^{\top}\boldsymbol{\beta}_i$ is bounded due to the softmax's property in Lemma 8, then by Lemma C.1 of Nguyen and Mondelli [2020], w.p. at least $1 - \delta_3$, one has $\sqrt{2\ell(\boldsymbol{\theta}^0)} \leq \tilde{\mathcal{O}}(\sqrt{N})$. Plugging back Eqs. (2) and (3), it suffices to prove the following inequality.

$$
d_m\lambda_0/4 \geq \tilde{\mathcal{O}}(Nd_s^{3/2}C_x\sqrt{d_m/d}), \tag{54}
$$

$$
(d_m\lambda_0/4)^{3/2} \geq \tilde{\mathcal{O}}(N^{3/2}d_s^3C_x^2(1 + 4C_x^4d_s^2d^{-2})). \tag{55}
$$

By Lemma 15, with probability at least $1 - \exp\left(-\Omega((N-1)^{-\hat{c}}d_s^{-1})\right)$, one has $\lambda_0 \geq \Theta(\eta_V/d_s) = \Theta(d^{-1}d_s^{-1})$. Thus, when $d_m \geq \tilde{\Omega}(N^2)$, all of the above conditions hold. As a result, the conditions in Eqs. (45) and (46) are satisfied and the convergence of training Transformer is guaranteed as in Eq. (4). Note that one can achieve the same width requirement and probability for He initialization, and the proof bears resemblance to the LeCun initialization.

$\square$

## C.4   Proof of Theorem 2 (NTK analysis)

*Proof.* Below, we present the proof for NTK initialization. We select $C_Q = C_K = C_V = C_O = 1$, then by Lemma 5, with probability at least $1 - 8e^{-d_m/2}$, we have:

$$
\begin{aligned}
\bar{\lambda}_V &= \mathcal{O}(\sqrt{d_m} + \sqrt{d}), \quad \bar{\lambda}_O = \mathcal{O}(\sqrt{d_m}), \\
\bar{\lambda}_Q &= \mathcal{O}(\sqrt{d_m} + \sqrt{d}), \quad \bar{\lambda}_K = \mathcal{O}(\sqrt{d_m} + \sqrt{d}).
\end{aligned}
\tag{56}
$$

When $d_m \geq d$, plugging Eq. (56) into Eqs. (2) and (3), it suffices to prove the following equations.

$$
\alpha^2 \geq \mathcal{O}(\sqrt{N}d_s^{5/2}C_x^3)\sqrt{2\ell(\boldsymbol{\theta}^0)}, \tag{57}
$$

$$
\alpha^3 \geq \mathcal{O}(Nd_s^3C_x^2(1 + 4C_x^4d_s^2))\sqrt{2\ell(\boldsymbol{\theta}^0)}. \tag{58}
$$

Next, we will provide the lower bound for $\alpha^2 = \lambda_{\min}((\boldsymbol{F}_{\text{pre}}^0)(\boldsymbol{F}_{\text{pre}}^0)^{\top})$. In the following context, we hide the index 0 for simplification. One can note that $\boldsymbol{F}_{\text{pre}}\boldsymbol{F}_{\text{pre}}^{\top}$ is the summation of PSD matrices, thus, it suffices to lower bound: $\lambda_{\min}(\hat{\boldsymbol{F}}_{\text{pre}}\hat{\boldsymbol{F}}_{\text{pre}}^{\top})$, where we introduce the following notations:

$$
\begin{aligned}
\hat{\boldsymbol{F}}_{\text{pre}}\hat{\boldsymbol{F}}_{\text{pre}}^{\top} &= \tau_1^2 \sigma_r(\boldsymbol{\Phi}\boldsymbol{W}_v^{\top})\sigma_r(\boldsymbol{\Phi}\boldsymbol{W}_v^{\top})^{\top}, \\
\boldsymbol{\Phi} &= [\boldsymbol{X}_1^{\top}\boldsymbol{\beta}_{1,1}, ..., \boldsymbol{X}_N^{\top}\boldsymbol{\beta}_{1,N}]^{\top}, \\
\boldsymbol{\beta}_{1,n} &= \sigma_s\left(\tau_0 \boldsymbol{X}_n^{(1,:)}\boldsymbol{W}_Q^{\top}\boldsymbol{W}_K\boldsymbol{X}_n^{\top}\right)^{\top} \quad n \in [N].
\end{aligned}
\tag{59}
$$

By Eq. (48), w.p at least $1 - \delta$, $\lambda_{\min}(\hat{\boldsymbol{F}}_{\text{pre}}\hat{\boldsymbol{F}}_{\text{pre}}^{\top}) \geq d_m\lambda_0/4$, as long as it holds $d_m \geq \tilde{\Omega}(N/\lambda_0)$, where $\lambda_0 = \lambda_{\min}\left(\mathbb{E}_{\boldsymbol{w} \sim \mathcal{N}(0, \mathbb{I}_d)}[\sigma_r(\boldsymbol{\Phi}\boldsymbol{w})\sigma_r(\boldsymbol{\Phi}\boldsymbol{w})^{\top}]\right)$, and $\tilde{\Omega}$ hides logarithmic factors depending on $\delta_1$. Lastly, w.p. at least $1 - \delta_3$, one has $\sqrt{2\ell(\boldsymbol{\theta}^0)} \leq \tilde{\mathcal{O}}(\sqrt{N})$. Plugging back Eqs. (2) and (3), it suffices to prove the following inequality.

$$
d_m\lambda_0/4 \geq \tilde{\mathcal{O}}(Nd_s^{5/2}C_x^3), \tag{60}
$$

$$
(d_m\lambda_0/4)^{3/2} \geq \tilde{\mathcal{O}}(N^{3/2}d_s^3C_x^2(1 + 4C_x^4d_s^2)). \tag{61}
$$

By Lemma 15, with probability at least $1 - \exp\left(-\Omega((N-1)^{-\hat{c}}d_s^{-1})\right)$, we have $\lambda_0 \geq \Omega(1)$. Thus, all of the above conditions hold when $d_m = \tilde{\Omega}(N)$. As a result, the conditions in Eqs. (57) and (58) are satisfied and the convergence of training Transformer is guaranteed. $\square$

## C.5 Discussion for different initialization schemes

Recall that the convergence result in Theorem 2 shows:

$$\ell(\boldsymbol{\theta}^{\top}) \leq \left(1 - \gamma \frac{\alpha^2}{2}\right)^t \ell(\boldsymbol{\theta}^0).$$

Thus, to discuss the convergence speed for different initialization, we need to check the lower bound for $\alpha^2$. From the proofs for different initialization schemes above, we have the following lower bound for $\alpha^2$, i.e.,

$$\alpha^2 = \lambda_{\min}(\hat{\boldsymbol{F}}_{\text{pre}} \hat{\boldsymbol{F}}_{\text{pre}}^{\top}) \geq \tau_1^2 d_m \lambda_0/4 \geq \tau_1^2 \eta_V d_m \Omega(N/d),$$

with high probability. Plugging the value of $\tau_1$ and $\eta_V$, we observe that for LeCun initialization and He initialization: $\alpha^2 \geq \Omega(d_m N/d)$ while for NTK initialization: $\alpha^2 \geq \Omega(N/d)$. Thus, the convergence speed of LeCun initialization and He initialization is faster than NTK initialization. As a result, faster step-size is required for NTK intialization.

## C.6 Discussion for $\tau_0 = d_m^{-1}$ and $\tau_0 = d_m^{-1/2}$

Eq. (4) indicates that the convergence speed is affected by $\alpha$, there we compare the lower bound for $\alpha$ for these two scaling. In Appendix C.3, we have proved that under the LeCun initialization, one has $\alpha^2 \geq d_m \lambda_0/4 \geq d_m \eta_V \mu(\sigma_r)^2 \Theta(\|\boldsymbol{\beta}_{1,k}\|_2^2)$. Note that this bound holds for these two different scaling, which is inside $\boldsymbol{\beta}$. Thus in the next, we need to see the difference between the lower bound of $\|\boldsymbol{\beta}_{1,k}\|_2^2)$ in the case of these two scalings. Specifically, for $\tau_0 = d_m^{-1/2}$ scaling, we have proved that $\|\boldsymbol{\beta}_{1,k}\|_2^2 \geq 1/d_s$ by Lemma 8. However, for the case of $\tau_0 = d_m^{-1}$ scaling, when the width is large enough, the value inside the softmax tends to zero, as a result, $\|\boldsymbol{\beta}_{1,k}\|_2^2 \approx 1/d_s$. Thus, we can see that as the width increases, the convergence speed of $\tau_0 = d_m^{-1/2}$ could be faster. Lastly, we remark on the difference in the step size for these two scales, which can be seen from the definition of $C$ and its corresponding $c_1, c_2, c_3$ in Appendix C.2.

## C.7 Discussion for extension to deep Transformer and residual Transformer

Extension from our shallow Transformer to deep Transformer is not technically difficult as they share the same analysis framework. Nevertheless, the extension requires several tedious derivations and calculations involving the query, key, and value matrices along different layers. Here we point out the proof roadmap for this extension. The first step is following Proposition. 1 to provide the sufficient condition for the convergence guarantee, e.g., Eqs. (2) and (3). The second step is similar to Theorem 2, where we need to verify the aforementioned assumptions for different initialization. In the second part, the main task is to prove the lower bound of $\alpha := \sigma_{\min}\left(\boldsymbol{F}_{\text{pre}}^0\right)$, where $\boldsymbol{F}_{\text{pre}}$ is the output of the last hidden layer. One can apply concentration inequality to bound the difference between $\sigma_{\min}\left(\boldsymbol{F}_{\text{pre}}^0\right)$ and $\sigma_{\min}\left(\boldsymbol{F}_{\text{pre}}^{\star 0}\right)$, where the latter is the corresponding limit in infinite width. Lastly, one needs to plug the lower bound into the assumptions in order to obtain the width requirement.

Our proof framework is general and can handle the following residual Transformer. Here we give a proof sketch to show how to achieve this. Specifically, we consider the residual block in the self-attention layer:

$$\mathbf{A_1} = \text{Self-attention}(\mathbf{X}) \triangleq \sigma_{\mathbf{s}}\left(\tau_0(\mathbf{X}\mathbf{W_Q^{\top}})\left(\mathbf{X}\mathbf{W_K^{\top}}\right)^{\top}\right)(\mathbf{X}\mathbf{W_V^{\top}}) + \mathbf{X}.$$

As a result, the output becomes

$$f(\boldsymbol{X}) = \tau_1 \boldsymbol{w}_O^{\top} \sum_{i=1}^{d_s} \sigma_r \left(\boldsymbol{W}_V \boldsymbol{X}^{\top} \boldsymbol{\beta}_i + (\boldsymbol{X}^{(i:)})^{\top}\right).$$

To prove the convergence of the above residual Transformer, the first part that will be modified is the proof for Proposition 1. The formula for $\boldsymbol{f}_{\text{pre}}$ becomes as follows:

$$\boldsymbol{f}_{pre} = \tau_1 \boldsymbol{w}_O^{\top} \sum_{i=1}^{d_s} \sigma_r \left(\boldsymbol{W}_V \boldsymbol{X}^{\top} \boldsymbol{\beta}_i + (\boldsymbol{X}^{(i:)})^{\top}\right).$$

Lemmas 7 and 8 remain unchanged. In Lemma 9, only the first step in the proof changes while the remaining part does not change because the term $\boldsymbol{X}^{(i:)^\top}$ in two adjacent time steps cancels out. In Lemma 10, we have

$$||\tau_1 \sum_{i=1}^{d_s} \sigma_r \left( \boldsymbol{W}_V^{t'} \boldsymbol{X}_1^\top \boldsymbol{\beta}_{i,1}^{t'}(\boldsymbol{X}^{(i:)}) \right) ||_2 \le \tau_1 d_s \left( ||\boldsymbol{W}_V^{t'}||_2 d_s^{1/2} C_x + C_x \right).$$

Similarly, in the remaining lemmas, we need to add the additional term $C_x$ for the upper bound of $||\boldsymbol{X}^{(i:)}||_2$.

The second part is the proof for Theorem 1 regarding the lower bound for $\alpha_0$. By Weyl's inequality:

$$\alpha_0 = \sigma_{\min}(\boldsymbol{F}_{\text{pre}}) \ge \sigma_{\min}(\boldsymbol{F}_{\text{pre}}^*) - ||\boldsymbol{F}_{\text{pre}} - \boldsymbol{F}_{\text{pre}}^*||_2,$$

where we denote by $\boldsymbol{F}_{\text{pre}} = [\boldsymbol{f}_{\text{pre}}(\boldsymbol{X}_1), \cdots, \boldsymbol{f}_{\text{pre}}(\boldsymbol{X}_N)]$, and $\boldsymbol{F}_{\text{pre}}^*$ is the corresponding one without the residual connection. Then we can upper bound the second term can be bounded as follows:

$$\begin{aligned}
||\boldsymbol{F}_{\text{pre}} - \boldsymbol{F}_{\text{pre}}^*||_2 &\le ||\boldsymbol{F}_{\text{pre}} - \boldsymbol{F}_{\text{pre}}^*||_F \\
&\le \sqrt{N} ||\tau_1 \boldsymbol{w}_O^\top \sum_{i=1}^{d_s} \sigma_r \left( \boldsymbol{W}_V \boldsymbol{X}^\top \boldsymbol{\beta}_i + (\boldsymbol{X}^{(i:)})^\top \right) - \tau_1 \boldsymbol{w}_O^\top \sum_{i=1}^{d_s} \sigma_r \left( \boldsymbol{W}_V \boldsymbol{X}^\top \boldsymbol{\beta}_i \right) ||_2 \quad (62) \\
&\le \sqrt{N} \tau_1 d_s ||\boldsymbol{w}_O||_2 ||\boldsymbol{W}_V||_2 C_x.
\end{aligned}$$

The remains step follows the same as previous analysis.

## C.8 Linear over-parametrization and attention module behaving as a pooling layer

In this section, we discuss the link between the linear over-parametrization and attention module behaving as a pooling layer under the $d_m^{-1}$ scaling. First, due to the $d_m^{-1}$ scaling, the attention module degenerates to a pooling layer according to the law of large numbers. In this case, the nonlinearity on $\boldsymbol{X}$ disappears and thus the minimum eigenvalue of $\boldsymbol{\Theta}\boldsymbol{\Theta}^\top$ can be estimated via $\boldsymbol{X}\boldsymbol{X}^\top$. Accordingly, this leads to the minimum eigenvalue in the order of $\Omega(N/d)$, and thus linear over-parameterization is enough.

# D   Proof for NTK

In this section, we elaborate the proof for Lemma 1 in Appendix D.1, the proof for Theorem 3 in Appendix D.3, respectively.

## D.1   Proof of Lemma 1

*Proof.* We will compute the inner product of the Jacobian of each weight separately. Firstly, we analyze $\boldsymbol{w}_O$. Let us denote by $\boldsymbol{\beta}_i := \sigma_s \left( \tau_0 \boldsymbol{X}^{(i,:)} \boldsymbol{W}_Q^\top \boldsymbol{W}_K \boldsymbol{X}^\top \right)^\top \in \mathbb{R}^{d_s}$. Then:

$$\frac{\partial f(\boldsymbol{X})}{\partial \boldsymbol{w}_O} = \tau_1 \sum_{i=1}^{d_s} \sigma_r \left( \boldsymbol{W}_V \boldsymbol{X}^\top \boldsymbol{\beta}_i \right).$$

The inner product of the gradient is:

$$\begin{aligned}
\lim_{d_m \to \infty} &\left\langle \frac{\partial f(\boldsymbol{X})}{\partial \boldsymbol{w}_O}, \frac{\partial f(\boldsymbol{X}')}{\partial \boldsymbol{w}_O} \right\rangle \\
&= \tau_1^2 \sum_{i=1,j=1}^{d_s} \lim_{d_m \to \infty} \left( \sigma_r \left( \boldsymbol{W}_V \boldsymbol{X}'^\top \boldsymbol{\beta}_j' \right) \right)^\top \left( \sigma_r \left( \boldsymbol{W}_V \boldsymbol{X}^\top \boldsymbol{\beta}_i \right) \right) \quad (63) \\
&= d_s^2 \mathbb{E}_{\boldsymbol{w} \sim \mathcal{N}(\boldsymbol{0}, \boldsymbol{I})} \left( \sigma_r \left( \boldsymbol{w}^\top \boldsymbol{X}'^\top \boldsymbol{1}_{d_s} \right) \right) \left( \sigma_r \left( \boldsymbol{w}^\top \boldsymbol{X}^\top \boldsymbol{1}_{d_s} \right) \right),
\end{aligned}$$

where the second equality uses the law of large numbers. Secondly, we analyze $\boldsymbol{W}_Q$:

$$\frac{\partial f(\boldsymbol{X})}{\partial W_Q^{(p,q)}} = \tau_0 \tau_1 \sum_{i=1}^{d_s} \left( \boldsymbol{w}_O \circ \dot{\sigma}_r \left( \boldsymbol{W}_V \boldsymbol{X}^\top \boldsymbol{\beta}_i \right) \right)^\top \boldsymbol{W}_V \boldsymbol{X}^\top \left( \text{diag}(\boldsymbol{\beta}_i) - \boldsymbol{\beta}_i \boldsymbol{\beta}_i^\top \right) \boldsymbol{X} \boldsymbol{W}_K^\top \boldsymbol{e}_p \boldsymbol{e}_q^\top \boldsymbol{X}^{(i,:)\top}.$$

Thus:
$$\frac{\partial f(\boldsymbol{X})}{\partial \boldsymbol{W}_Q} = \tau_0\tau_1 \sum_{i=1}^{d_s} \boldsymbol{W}_k \boldsymbol{X}^\top \left(\operatorname{diag}(\boldsymbol{\beta}_i) - \boldsymbol{\beta}_i\boldsymbol{\beta}_i^\top\right) \boldsymbol{X}\boldsymbol{W}_V^\top \left(\boldsymbol{w}_O \circ \dot{\sigma}_r \left(\boldsymbol{W}_V \boldsymbol{X}^\top \boldsymbol{\beta}_i\right)\right) \boldsymbol{X}^{(i,:)}.$$

The inner product of the gradient is:
$$\lim_{d_m\to\infty} \left\langle \frac{\partial f(\boldsymbol{X})}{\partial \boldsymbol{W}_Q}, \frac{\partial f(\boldsymbol{X}')}{\partial \boldsymbol{W}_Q} \right\rangle$$

$$= \lim_{d_m\to\infty} \tau_0^2\tau_1^2 \sum_{i=1,j=1}^{d_s} \operatorname{Trace}\left(\boldsymbol{W}_k \boldsymbol{X}^\top \left(\operatorname{diag}(\boldsymbol{\beta}_i) - \boldsymbol{\beta}_i\boldsymbol{\beta}_i^\top\right) \boldsymbol{X}\boldsymbol{W}_V^\top \left(\boldsymbol{w}_O \circ \dot{\sigma}_r \left(\boldsymbol{W}_V \boldsymbol{X}^\top \boldsymbol{\beta}_i\right)\right) \boldsymbol{X}^{(i,:)}\right.$$

$$\left. \boldsymbol{X}^{(j,:)\top} \left(\boldsymbol{w}_O \circ \dot{\sigma}_r \left(\boldsymbol{W}_V \boldsymbol{X}'^\top \boldsymbol{\beta}'_j\right)\right)^\top \boldsymbol{W}_V \boldsymbol{X}'^\top \left(\operatorname{diag}(\boldsymbol{\beta}'_j) - \boldsymbol{\beta}'_j\boldsymbol{\beta}_j'^\top\right) \boldsymbol{X}'\boldsymbol{W}_k^\top\right)$$

$$= \lim_{d_m\to\infty} \sum_{i=1,j=1}^{d_s} \boldsymbol{X}^{(i,:)}\boldsymbol{X}^{(j,:)\top} \langle \tau_0\tau_1 \left(\boldsymbol{X}^\top \left(\operatorname{diag}(\boldsymbol{\beta}_i) - \boldsymbol{\beta}_i\boldsymbol{\beta}_i^\top\right) \boldsymbol{X}\boldsymbol{W}_V^\top \left(\boldsymbol{w}_O \circ \dot{\sigma}_r \left(\boldsymbol{W}_V \boldsymbol{X}^\top \boldsymbol{\beta}_i\right)\right)\right),$$

$$\tau_0\tau_1 \boldsymbol{W}_k^\top \boldsymbol{W}_k \boldsymbol{X}'^\top \left(\operatorname{diag}(\boldsymbol{\beta}'_j) - \boldsymbol{\beta}'_j\boldsymbol{\beta}_j'^\top\right) \boldsymbol{X}'\boldsymbol{W}_V^\top \left(\boldsymbol{w}_O \circ \dot{\sigma}_r \left(\boldsymbol{W}_V \boldsymbol{X}'^\top \boldsymbol{\beta}'_j\right)\right)\rangle.$$

For the first term of the inner product, we have:
$$\lim_{d_m\to\infty} \tau_0\tau_1 \boldsymbol{X}^\top \left(\operatorname{diag}(\boldsymbol{\beta}_i) - \boldsymbol{\beta}_i\boldsymbol{\beta}_i^\top\right) \boldsymbol{X}\boldsymbol{W}_V^\top \left(\boldsymbol{w}_O \circ \dot{\sigma}_r \left(\boldsymbol{W}_V \boldsymbol{X}^\top \boldsymbol{\beta}_i\right)\right)$$

$$= \lim_{d_m\to\infty} \boldsymbol{X}^\top \left(\operatorname{diag}(\boldsymbol{\beta}_i) - \boldsymbol{\beta}_i\boldsymbol{\beta}_i^\top\right) \boldsymbol{X} \begin{bmatrix} \tau_0\tau_1 \sum_{k=1}^{d_m} W_V^{(k,1)} w_O^{(k)} \dot{\sigma}_r \left(\boldsymbol{W}_V \boldsymbol{X}^\top \boldsymbol{\beta}_i\right)^{(k)} \\ \vdots \\ \tau_0\tau_1 \sum_{k=1}^{d_m} W_V^{(k,d)} w_O^{(k)} \dot{\sigma}_r \left(\boldsymbol{W}_V \boldsymbol{X}^\top \boldsymbol{\beta}_i\right)^{(k)} \end{bmatrix}$$

$$= \boldsymbol{X}^\top \left(\operatorname{diag}(\boldsymbol{1}_{d_s}) - \boldsymbol{1}_{d_s}\boldsymbol{1}_{d_s}^\top\right) \boldsymbol{X} \begin{bmatrix} 0 \\ \vdots \\ 0 \end{bmatrix} = 0,$$

where the second equality is by the law of large numbers and $\mathbb{E}w = 0$ for a random variable $w \sim \mathcal{N}(0,1)$. Thus: $\lim_{d_m\to\infty} \left\langle \frac{\partial f(\boldsymbol{X})}{\partial \boldsymbol{W}_Q}, \frac{\partial f(\boldsymbol{X}')}{\partial \boldsymbol{W}_Q} \right\rangle = 0$. Similarly, $\lim_{d_m\to\infty} \left\langle \frac{\partial f(\boldsymbol{X})}{\partial \boldsymbol{W}_K}, \frac{\partial f(\boldsymbol{X}')}{\partial \boldsymbol{W}_K} \right\rangle = 0$.
Lastly, we analyze $\boldsymbol{W}_V$:
$$\frac{\partial f(\boldsymbol{X})}{\partial W_V^{(p,q)}} = \tau_1 \sum_{i=1}^{d_s} \left(\boldsymbol{w}_O \circ \dot{\sigma}_r \left(\boldsymbol{W}_V \boldsymbol{X}^\top \boldsymbol{\beta}_i\right)\right)^\top \boldsymbol{e}_p \boldsymbol{e}_q^\top \boldsymbol{X}^\top \boldsymbol{\beta}_i.$$

Thus:
$$\frac{\partial f(\boldsymbol{X})}{\partial \boldsymbol{W}_V} = \tau_1 \sum_{i=1}^{d_s} \left(\boldsymbol{w}_O \circ \dot{\sigma}_r \left(\boldsymbol{W}_V \boldsymbol{X}^\top \boldsymbol{\beta}_i\right)\right) \boldsymbol{\beta}_i^\top \boldsymbol{X}.$$

The inner product of the gradient is:
$$\lim_{d_m\to\infty} \left\langle \frac{\partial f(\boldsymbol{X})}{\partial \boldsymbol{W}_V}, \frac{\partial f(\boldsymbol{X}')}{\partial \boldsymbol{W}_V} \right\rangle$$

$$= \lim_{d_m\to\infty} \tau_1^2 \sum_{i=1,j=1}^{d_s} \operatorname{Trace}\left(\left(\boldsymbol{w}_O \circ \dot{\sigma}_r \left(\boldsymbol{W}_V \boldsymbol{X}^\top \boldsymbol{\beta}_i\right)\right) \boldsymbol{\beta}_i^\top \boldsymbol{X}\boldsymbol{X}'^\top \boldsymbol{\beta}'_j \left(\boldsymbol{w}_O \circ \dot{\sigma}_r \left(\boldsymbol{W}_V \boldsymbol{X}'^\top \boldsymbol{\beta}'_j\right)\right)^\top\right)$$

$$= \lim_{d_m\to\infty} \tau_1^2 \sum_{i=1,j=1}^{d_s} \left(\boldsymbol{w}_O \circ \dot{\sigma}_r \left(\boldsymbol{W}_V \boldsymbol{X}'^\top \boldsymbol{\beta}'_j\right)\right)^\top \left(\boldsymbol{w}_O \circ \dot{\sigma}_r \left(\boldsymbol{W}_V \boldsymbol{X}^\top \boldsymbol{\beta}_i\right)\right) \boldsymbol{\beta}_i^\top \boldsymbol{X}\boldsymbol{X}'^\top \boldsymbol{\beta}'_j$$

$$= \lim_{d_m\to\infty} \tau_1^2 \sum_{i=1,j=1}^{d_s} \sum_{k=1}^{d_m} (w_O^{(k)})^2 \dot{\sigma}_r \left(\boldsymbol{W}_V^{(k,:)} \boldsymbol{X}'^\top \boldsymbol{\beta}'_j\right) \dot{\sigma}_r \left(\boldsymbol{W}_V^{(k,:)} \boldsymbol{X}^\top \boldsymbol{\beta}_i\right) \boldsymbol{\beta}_i^\top \boldsymbol{X}\boldsymbol{X}'^\top \boldsymbol{\beta}'_j$$

$$= d_s^2 \mathbb{E}_{\boldsymbol{w}\sim\mathcal{N}(\boldsymbol{0},\boldsymbol{I})} \left(\dot{\sigma}_r \left(\boldsymbol{w}^\top \boldsymbol{X}'^\top \boldsymbol{1}_{d_s}\right)\right) \left(\dot{\sigma}_r \left(\boldsymbol{w}^\top \boldsymbol{X}^\top \boldsymbol{1}_{d_s}\right)\right) \left(\boldsymbol{1}_{d_s}^\top \boldsymbol{X}\boldsymbol{X}'^\top \boldsymbol{1}_{d_s}\right),$$
(64)

where the last equality uses the law of large numbers.

$\square$

## D.2   NTK minimum eigenvalue

**Lemma 16.** *Given $\Phi$ defined in Eq. (53) and $\Phi^\star$ defined in Lemma 1, denote $\lambda_* = \lambda_{\min}(\Phi^\star\Phi^{\star\top})$, and suppose the width satisfies $d_m = \Omega(\frac{N^2\sqrt{\log(2d^2N^2/\delta)}}{\lambda_*^2})$, then with probability at least $1 - \delta$, one has $\left\|\Phi\Phi^\top - \Phi^\star\Phi^{\star\top}\right\|_{\mathrm{F}} \le \frac{\lambda_*}{4}$ and $\lambda_{\min}(\Phi\Phi^\top) \ge \frac{3\lambda_*}{4}$.*

*Proof.* In the following content, the variable with $\star$ indicates the corresponding variable with infinite width $d_m$. According to the definition of $\Phi$ in Eq. (53) and $\Phi^\star$ in Lemma 1, for each entry in $\Phi\Phi^\top$ and $\Phi^\star\Phi^{\star\top}$, we have

$$
\left|(\Phi\Phi^\top - \Phi^\star\Phi^{\star\top})^{(n,r)}\right| = |\boldsymbol{\beta}_{1,n}^\top X_n X_r^\top \boldsymbol{\beta}_{1,r} - \boldsymbol{\beta}_{1,n}^{\star\top} X_n X_r^\top \boldsymbol{\beta}_{1,r}^\star|
$$
$$
\le \left\|X_n X_r^\top(\boldsymbol{\beta}_{1,n} - \boldsymbol{\beta}_{1,n}^\star)\right\|_2 + \left\|X_n X_r^\top(\boldsymbol{\beta}_{1,r} - \boldsymbol{\beta}_{1,r}^\star)\right\|_2 \le C_x^2 d_s(\left\|\boldsymbol{\beta}_{1,n} - \boldsymbol{\beta}_{1,n}^\star\right\|_2 + \left\|\boldsymbol{\beta}_{1,r} - \boldsymbol{\beta}_{1,r}^\star\right\|_2).
$$
$$(65)$$

Next, we will bound $\left\|\boldsymbol{\beta}_{1,n} - \boldsymbol{\beta}_{1,n}^\star\right\|_2$,

$$
\left\|\boldsymbol{\beta}_{1,n} - \boldsymbol{\beta}_{1,n}^\star\right\|_2 = \left\|\sigma_s\left(\tau_0 X_n^{(1,:)} W_Q^\top W_K X_n^\top\right)^\top - \sigma_s\left(\tau_0 X_n^{(1,:)} W_Q^{\star\top} W_K^\star X_n^\top\right)^\top\right\|_2
$$
$$
\le 2C_x^2 d_s \left\|\tau_0 W_Q^\top W_K - \tau_0 W_Q^{\star\top} W_K^\star\right\|_{\mathrm{F}} \quad \text{( By Lemma 7)}.
$$
$$(66)$$

Let first consider the absolute value of each element of $\tau_0 W_Q^\top W_K - \tau_0 W_Q^{\star\top} W_K^\star$, i.e.,

$$
|\tau_0 W_Q^\top W_K - \tau_0 W_Q^{\star\top} W_K^\star|^{(i,j)} = |\tau_0 \sum_{q=1}^{d_m} W_Q^{(q,i)} W_k^{(q,j)} - \tau_0 \sum_{q=1}^{d_m} W_Q^{\star(q,i)} W_k^{\star(q,j)}|.
$$

Since $W_Q^{(q,i)} W_k^{(q,j)} \sim SE(\sqrt{2}\eta_{QK}, \sqrt{2}\eta_{QK})$ is sub-exponential random variable, by Lemma 2 and Lemma 3, $\tau_0 \sum_{q=1}^{d_m} W_Q^{(q,i)} W_k^{(q,j)} \sim SE(\tau_0\eta_{QK}\sqrt{2d_m}, \sqrt{2}\tau_0\eta_{QK})$, by Bernstein's inequality, when $d_m \ge 2\log(2/\delta)$, the following inequality holds with probability at least $1 - \delta$:

$$
|\tau_0 \sum_{q=1}^{d_m} W_Q^{(q,i)} W_k^{(q,j)} - \tau_0 \sum_{q=1}^{d_m} W_Q^{\star(q,i)} W_k^{\star(q,j)}| \le 2\tau_0\eta_{QK}\sqrt{d_m \log(2/\delta)}.
$$
$$(67)$$

Substituting $\delta = \frac{\delta'}{d^2}$ and applying union bound, we can obtain that when $d_m \ge 2\log\frac{2d^2}{\delta'}$, with probability at least $1 - \delta'$:

$$
\left\|\boldsymbol{\beta}_{1,n} - \boldsymbol{\beta}_{1,n}^\star\right\|_2 \le 4\tau_0\eta_{QK}C_x^2 d_s d\sqrt{d_m \log(2d^2/\delta')}.
$$
$$(68)$$

Substituting back to Eq. (65), the following inequality holds with the same width requirement and probability. $|(\Phi\Phi^\top - \Phi^\star\Phi^{\star\top})^{(n,r)}| \le 8\tau_0\eta_{QK}C_x^4 d_s d\sqrt{d_m \log(2d^2/\delta')}$ Applying the union bound over the index $(n,r)$ for $n \in [N]$ and $r \in [N]$, and substituting $\delta' = \frac{\delta''}{N^2}$, we obtain that when the width $d_m \ge 2\log\frac{2d^2N^2}{\delta''}$, the following inequality holds with probability at least $1 - \delta''$:

$$
\left\|\Phi\Phi^\top - \Phi^\star\Phi^{\star\top}\right\|_{\mathrm{F}} = \sqrt{\sum_{n=1}^N \sum_{r=1}^N |(\Phi\Phi^\top - \Phi^\star\Phi^{\star\top})^{(n,r)}|^2} \le 8N\tau_0\eta_{QK}C_x^4 d_s d\sqrt{d_m \log(2d^2N^2/\delta'')}.
$$

In the case of LeCun initialization when $d_m = \Omega(\frac{N^2\sqrt{\log(2d^2N^2/\delta'')}}{\lambda_0^2})$, $\left\|\Phi\Phi^\top - \Phi^\star\Phi^{\star\top}\right\|_{\mathrm{F}} \le \frac{\lambda_*}{4}$. Lastly, one has:

$$
\lambda_{\min}(\Phi\Phi^\top) \ge \lambda_* - \left\|\Phi\Phi^\top - \Phi^\star\Phi^{\star\top}\right\|_2 \ge \lambda_* - \left\|\Phi\Phi^\top - \Phi^\star\Phi^{\star\top}\right\|_{\mathrm{F}} \ge \frac{3\lambda_*}{4}.
$$

where the first inequality is by Weyl's inequality. One can easily check that the same result also holds for He initialization and NTK initialization. $\qquad\square$

**Lemma 17.** *Given the $\Phi^\star$ defined in Lemma 1, then when $N \ge \Omega(d^4)$, with probability at least $1 - e^{-d}$ one has: $\lambda_{\min}(\Phi^\star\Phi^{\star\top}) \ge \Theta(N/d)$.*

*Proof.* Firstly, let us define $\boldsymbol{Z}^\star = \boldsymbol{\Phi}^\star \boldsymbol{\Sigma}^{-1/2}$, then:

$$\lambda_{\min}(\boldsymbol{\Phi}^\star \boldsymbol{\Phi}^{\star\top}) = \lambda_{\min}(\boldsymbol{\Phi}^{\star\top} \boldsymbol{\Phi}^\star) = \lambda_{\min}(\boldsymbol{\Sigma}^{1/2} \boldsymbol{Z}^{\star\top} \boldsymbol{Z}^\star \boldsymbol{\Sigma}^{1/2\top})$$

$$\geq \lambda_{\min}(\boldsymbol{Z}^{\star\top} \boldsymbol{Z}^\star) \lambda_{\min}(\boldsymbol{\Sigma}) \geq \lambda_{\min}(\boldsymbol{Z}^{\star\top} \boldsymbol{Z}^\star) C_\Sigma \,,$$

where the last inequality is by Assumption 4). Note that we can reformulate $\boldsymbol{Z}^\star$ by plugging $\boldsymbol{\Phi}^\star$ as follows:

$$\boldsymbol{Z}^\star = \boldsymbol{\Phi}^\star \boldsymbol{\Sigma}^{-1/2} = \begin{bmatrix} \frac{1}{d_s} \sum_{i=1}^{d_s} \boldsymbol{X}_1^{(i,:)} \boldsymbol{\Sigma}^{-1/2} \\ \vdots \\ \frac{1}{d_s} \sum_{i=1}^{d_s} \boldsymbol{X}_N^{(i,:)} \boldsymbol{\Sigma}^{-1/2} \end{bmatrix} .$$

Recall that in Assumption 4, we define the random vector $\boldsymbol{x} = \boldsymbol{X}_n^{(i,:)\top}$ and write the covariance matrix for each feature of $\boldsymbol{x}$ as $\boldsymbol{\Sigma} = \mathbb{E}[\boldsymbol{x}\boldsymbol{x}^\top]$. Here we further define the random vector $\boldsymbol{z} = \boldsymbol{\Sigma}^{-1/2}\boldsymbol{x}$. Clearly, the covariance matrix for each feature of $\boldsymbol{z}$ is $\boldsymbol{\Sigma}_z = \mathbb{E}[\boldsymbol{z}\boldsymbol{z}^\top] = \mathbb{I}_d$. Therefore, by Proposition 4 of Zhu et al. [2022]), we have $\lambda_{\min}(\boldsymbol{Z}^{\star\top} \boldsymbol{Z}^\star) \geq \Theta(N/d)$, which finishes the proof.

$\square$

**Lemma 18.** *Suppose the number of samples $N \geq \Omega(d^4)$ and the width $d_m = \tilde{\Omega}(N^2 \lambda_*^{-2})$, where $\lambda_* = \lambda_{\min}(\boldsymbol{\Phi}^\star \boldsymbol{\Phi}^{\star\top})$, then under Assumption 4, with probability at least $1 - \delta - e^{-d}$, one has $\lambda_0 = \lambda_{\min}\left(\mathbb{E}_{\boldsymbol{w}\sim\mathcal{N}(0,\eta_V \mathbb{I}_d)}[\sigma_r(\boldsymbol{\Phi}\boldsymbol{w})\sigma_r(\boldsymbol{\Phi}\boldsymbol{w})^T]\right) \geq \Theta(\eta_V N/d)$.*

*Proof.* By the Hermite expansion of $\sigma_r$, one has:

$$\lambda_0 \geq \eta_V \mu(\sigma_r)^2 \lambda_{\min}(\boldsymbol{\Phi}\boldsymbol{\Phi}^\top) \,,$$

where $\mu(\sigma_r)$ is the 1-st Hermite coefficient of ReLU satisfying $\mu(\sigma_r) > 0$. By Lemma 16, , when the width satisfies $d_m = \tilde{\Omega}(N^2 \lambda_*^{-2})$, then with probability at least $1 - \delta$, one has $\lambda_{\min}(\boldsymbol{\Phi}\boldsymbol{\Phi}^\top) \geq \frac{3}{4}\lambda_*$. Furthermore, by Lemma 17, when $N \geq \Omega(d^4)$ w.p. at least $1 - e^{-d}$ one has $\lambda_* = \Theta(N/d)$. Thus, with probability at least $1 - \delta - e^{-d}$, one has:

$$\lambda_0 \geq \frac{3}{4}\eta_V \mu(\sigma_r)^2 \lambda_* \geq \Theta(\eta_V N/d)).$$

$\square$

Below, we provide a lower bound for the minimum eigenvalue of NTK, which plays a key role in analyzing convergence, generalization bounds, and memorization capacity [Arora et al., 2019a, Nguyen et al., 2021, Montanari and Zhong, 2022, Bombari et al., 2022].

To prove this, we need the following assumption.

**Assumption 4.** *Let $\boldsymbol{x} = \sum_{i=1}^{d_s}(\boldsymbol{X}^{(i,:)})^\top$, $\boldsymbol{\Sigma} = \mathbb{E}[\boldsymbol{x}\boldsymbol{x}^\top]$, then we assume that $\boldsymbol{\Sigma}$ is positive definite, i.e., $\lambda_{\min}(\boldsymbol{\Sigma}) \geq C_\Sigma$ for some positive constant $C_\Sigma$.*

**Remark:** This assumption implies that the covariance matrix along each feature dimension is positive definite. In statistics and machine learning [Liu et al., 2021, Liang and Rakhlin, 2020, Vershynin, 2018, Hastie et al., 2022], the covariance of $\boldsymbol{x}$ is frequently assumed to be an identity matrix or positive definite matrix.

**Lemma 19** (Minimum eigenvalue of limiting NTK). *Under Assumptions 1 and 4 and scaling $\tau_0 = d_m^{-1}$, when $N \geq \Omega(d^4)$, the minimum eigenvalue of $\boldsymbol{K}$ can be lower bounded with probability at least $1 - e^{-d}$ as: $\lambda_{\min}(\boldsymbol{K}) \geq \mu(\sigma_r)^2 \Omega(N/d)$, where $\mu(\sigma_r)$ represents the first Hermite coefficient of the ReLU activation function.*

*Proof.* By the Hermite expansion of $\sigma_r$, we have:

$$\lambda_{\min}(\boldsymbol{K}) > \lambda_{\min}(\mathbb{E}_{\boldsymbol{w}\sim\mathcal{N}(\boldsymbol{0},\boldsymbol{I})}\left(\sigma_r\left(\boldsymbol{\Phi}^\star \boldsymbol{w}\right)\sigma_r\left(\boldsymbol{\Phi}^\star \boldsymbol{w}\right)^\top\right))$$

$$= \lambda_{\min}\left(\sum_{s=0}^{\infty} \mu_s(\sigma_r)^2 \bigodot_{i=1}^{s}\left(\boldsymbol{\Phi}^\star \boldsymbol{\Phi}^{\star\top}\right)\right) \quad \text{[Nguyen and Mondelli, 2020, Lemma D.3]}$$

$$\geq \mu(\sigma_r)^2 \lambda_{\min}(\boldsymbol{\Phi}^\star \boldsymbol{\Phi}^{\star\top}) \,.$$

Note that when $N \geq \Omega(d^4)$, based on Lemma 17 and the fact that $\boldsymbol{\Phi}^\star \boldsymbol{\Phi}^{\star\top}$ and $\boldsymbol{\Phi}^{\star\top}\boldsymbol{\Phi}^\star$ share the same non-zero eigenvalues, with probability at least $1 - e^{-d}$ one has:

$$\lambda_{\min}(\boldsymbol{K}) \geq \mu(\sigma_r)^2 (\frac{N}{d} - 9N^{2/3}d^{1/3}) = \mu(\sigma_r)^2 \Theta(N/d),$$

which completes the proof. $\qquad\square$

**Remark:** The minimum eigenvalue of the NTK plays an important role in the global convergence, similar to $\alpha$ in Proposition. 1. By defining $\alpha \triangleq \sigma_{\min}\left(\boldsymbol{F}_{\text{pre}}^0\right)$, under the over-parameterized regime with $d_m \geq N$, we have $\alpha^2 = \lambda_{\min}((\boldsymbol{F}_{\text{pre}}^0)^\top \boldsymbol{F}_{\text{pre}}^0)$, which is the exact minimum eigenvalue of the empirical NTK with respect to the weight in the output layer.

### D.3 Proof of Theorem 3

*Proof.* Before starting the proof, we introduce some useful lemmas that are used to analyze the randomness of initialized weight.

**Lemma 20.** *Given an initial weight vector $\boldsymbol{w} \in \mathbb{R}^{d_m}$ where each element is sampled independently from $\mathcal{N}(0, 1)$, for any $\tilde{\boldsymbol{w}}$ such that $\|\tilde{\boldsymbol{w}} - \boldsymbol{w}\|_2 \leq R$, with probability at least $1 - 2\exp(-d_m/2)$, one has:*

$$\|\tilde{\boldsymbol{w}}\|_2 \leq R + 3\sqrt{d_m}. \tag{69}$$

*Proof of Lemma 20.* By triangle inequality and Lemma 6:

$$\|\tilde{\boldsymbol{w}}\|_2 \leq \|\tilde{\boldsymbol{w}} - \boldsymbol{w}\|_2 + \|\boldsymbol{w}\|_2 \leq R + 3\sqrt{d_m}. \tag{70}$$

$\qquad\square$

Now we are ready to start our proof, in the following content, we avoid the tilde symbol $(\tilde{\cdot})$ over the weight for simplicity. Because we aim to study the effect of width ($d_m$), to simplify the proof, we set the embedding dimension of the input as one, we can write the output layer of the network into the following form:

$$f(\boldsymbol{x}) = \tau_0 \sum_{i=1}^{d_s} \sigma_r \left( \sigma_s \left( \tau_1 x^{(i)} \boldsymbol{w}_Q^\top \boldsymbol{w}_K \boldsymbol{x}^\top \right) \left( \boldsymbol{w}_V \boldsymbol{x}^\top \right) \right)^\top \boldsymbol{w}_O \tag{71}$$

The Hessian matrix $\boldsymbol{H}$ of the network can be written as the following structure:

$$\boldsymbol{H} = \begin{pmatrix} \boldsymbol{H}_{Q,Q} & \boldsymbol{H}_{Q,K} & \boldsymbol{H}_{Q,V} & \boldsymbol{H}_{Q,O} \\ \boldsymbol{H}_{K,Q} & \boldsymbol{H}_{K,K} & \boldsymbol{H}_{K,V} & \boldsymbol{H}_{K,O} \\ \boldsymbol{H}_{V,Q} & \boldsymbol{H}_{V,K} & \boldsymbol{H}_{V,V} & \boldsymbol{H}_{V,O} \\ \boldsymbol{H}_{O,Q} & \boldsymbol{H}_{O,K} & \boldsymbol{H}_{O,V} & \boldsymbol{H}_{O,O} \end{pmatrix}, \tag{72}$$

where we partition the Hessian $\boldsymbol{H}$ of the network to each Hessian block e.g., $\boldsymbol{H}_{Q,Q} = \frac{\partial^2 f}{\partial \boldsymbol{w}_Q \partial \boldsymbol{w}_Q}, \boldsymbol{H}_{Q,K} = \frac{\partial^2 f}{\partial \boldsymbol{w}_Q \partial \boldsymbol{w}_K}$, etc. Based on the triangle inequality of the norm, the spectral norm of the Hessian $\boldsymbol{H}$ can be upper bounded by the sum of the spectral norm of each Hessian block. We start by analyzing $\boldsymbol{H}_{Q,Q}$. The time step $t$ is hidden for simplification.

$$\frac{\partial f(\boldsymbol{x})}{\partial w_Q^{(j)}} = \tau_0 \tau_1 \sum_{i=1}^{d_s} x^{(i)} w_K^{(j)} \boldsymbol{w}_V^\top \left( \boldsymbol{w}_O \circ \acute{\sigma}_r \left( \boldsymbol{w}_V \boldsymbol{x}^\top \boldsymbol{\beta}_i \right) \right) \boldsymbol{x}^\top \left( \text{diag}(\boldsymbol{\beta}_i) - \boldsymbol{\beta}_i \boldsymbol{\beta}_i^\top \right) \boldsymbol{x}, \tag{73}$$

where the Jacobian of Softmax is given by Lemma 11. Next, we calculate the second-order derivative. Each element of the Hessian $\boldsymbol{H}_{Q,Q}$ is

$$H_{Q,Q}^{(j,p)} = \frac{\partial^2 f(\boldsymbol{x})}{\partial w_Q^{(j)} \partial w_Q^{(p)}} = \tau_0^2 \tau_1 \sum_{i=1}^{d_s} w_K^{(j)} w_K^{(p)} (x^{(i)})^2 \boldsymbol{w}_V^\top \left( \boldsymbol{w}_O \circ \acute{\sigma}_r \left( \boldsymbol{w}_V \boldsymbol{x}^\top \boldsymbol{\beta}_i \right) \right) \boldsymbol{x}^\top \boldsymbol{\Gamma}_i \boldsymbol{x},$$

where we denote by $\boldsymbol{\Gamma}_i = \left\{ \text{diag}\left(\boldsymbol{\beta}_i \circ \boldsymbol{x} - \boldsymbol{\beta}_i\boldsymbol{\beta}_i^\top \boldsymbol{x}\right) - \left(\text{diag}\left(\boldsymbol{\beta}_i\right) - \boldsymbol{\beta}_i\boldsymbol{\beta}_i^\top\right)\boldsymbol{x}\boldsymbol{\beta}_i^\top - \boldsymbol{\beta}_i\boldsymbol{x}^\top\left(\text{diag}\left(\boldsymbol{\beta}_i\right) - \boldsymbol{\beta}_i\boldsymbol{\beta}_i^\top\right)\boldsymbol{x} \right\}$, $\circ$ symbolizes Hadamard product. According to initialization of $\boldsymbol{w}_O$, $\boldsymbol{w}_V$, $\boldsymbol{w}_Q$, $\boldsymbol{w}_K$ and Lemma 20, with probability at least $1 - 8e^{-d_m/2}$, we have

$$\|\boldsymbol{w}_V\|_2 \le 3\sqrt{\eta_V d_m} + R, \quad \|\boldsymbol{w}_O\|_2 \le 3\sqrt{\eta_O d_m} + R,$$
$$\|\boldsymbol{w}_Q\|_2 \le 3\sqrt{\eta_Q d_m} + R, \quad \|\boldsymbol{w}_K\|_2 \le 3\sqrt{\eta_K d_m} + R.$$

The spectral norm of $\boldsymbol{H}_{Q,Q}$ can be bounded with probability at least $1 - 8e^{-d_m/2}$:

$$\|\boldsymbol{H}_{Q,Q}\|_2 = \|\tau_0^2 \tau_1 \sum_{i=1}^{d_s} (x^{(i)})^2 \boldsymbol{w}_V^\top \left(\boldsymbol{w}_O \circ \dot{\sigma}_r\left(\boldsymbol{w}_V \boldsymbol{x}^\top \boldsymbol{\beta}_i\right)\right) \boldsymbol{x}^\top \boldsymbol{\Gamma}_i \boldsymbol{x} \boldsymbol{w}_K \boldsymbol{w}_K^\top\|_2$$

$$\le |\tau_0^2 \tau_1 \sum_{i=1}^{d_s} (x^{(i)})^2 \boldsymbol{w}_V^\top \left(\boldsymbol{w}_O \circ \dot{\sigma}_r\left(\boldsymbol{w}_V \boldsymbol{x}^\top \boldsymbol{\beta}_i\right)\right) \boldsymbol{x}^\top \boldsymbol{\Gamma}_i \boldsymbol{x}| \|\boldsymbol{w}_K \boldsymbol{w}_K^\top\|_2 \quad \text{(Homogeneity of norm)}$$

$$\le |\tau_0^2 \tau_1 \sum_{i=1}^{d_s} (x^{(i)})^2 \boldsymbol{x}^\top \boldsymbol{\Gamma}_i \boldsymbol{x}| \|\boldsymbol{w}_O\|_2 \|\boldsymbol{w}_V\|_2 \|\boldsymbol{w}_K\|_2^2 \quad \text{(Cauchy–Schwarz inequality)}$$

$$\le \tau_0^2 \tau_1 |\sum_{i=1}^{d_s} (x^{(i)})^2 \boldsymbol{x}^\top \boldsymbol{\Gamma}_i \boldsymbol{x}| \left(3\sqrt{\eta_O d_m} + R\right)\left(3\sqrt{\eta_V d_m} + R\right)(3\sqrt{\eta_Q d_m} + R)(3\sqrt{\eta_K d_m} + R) \quad \text{(Lemma 20)}$$

$$= \mathcal{O}(1/\sqrt{d_m}),$$

where the last equality is by the bound of $\|\boldsymbol{\Gamma}_i\|_2$, by triangle inequality:

$$\|\boldsymbol{\Gamma}_i\|_2 \le \left\|\text{diag}\left(\boldsymbol{\beta}_i \circ \boldsymbol{x} - \boldsymbol{\beta}_i\boldsymbol{\beta}_i^\top \boldsymbol{x}\right)\right\|_2 + \left\|\left(\text{diag}\left(\boldsymbol{\beta}_i\right) - \boldsymbol{\beta}_i\boldsymbol{\beta}_i^\top\right)\boldsymbol{x}\boldsymbol{\beta}_i^\top\right\|_2 + \left\|\boldsymbol{\beta}_i\boldsymbol{x}^\top\left(\text{diag}\left(\boldsymbol{\beta}_i\right) - \boldsymbol{\beta}_i\boldsymbol{\beta}_i^\top\right)\boldsymbol{x}\right\|_2.$$
$$\tag{74}$$

For the first part of Eq. (74), we have

$$\left\|\text{diag}\left(\boldsymbol{\beta}_i \circ \boldsymbol{x} - \boldsymbol{\beta}_i\boldsymbol{\beta}_i^\top \boldsymbol{x}\right)\right\|_2 \le \left\|\text{diag}\left(\boldsymbol{\beta}_i \circ \boldsymbol{x} - \boldsymbol{\beta}_i\boldsymbol{\beta}_i^\top \boldsymbol{x}\right)\right\|_2 \le \left\|\left(\boldsymbol{\beta}_i \circ \boldsymbol{x} - \boldsymbol{\beta}_i\boldsymbol{\beta}_i^\top \boldsymbol{x}\right)\right\|_\infty$$
$$\le \|(\boldsymbol{\beta}_i \circ \boldsymbol{x})\|_\infty + \left\|\boldsymbol{\beta}_i\boldsymbol{\beta}_i^\top \boldsymbol{x}\right\|_\infty \le \|\boldsymbol{x}\|_\infty + |\boldsymbol{\beta}_i^\top \boldsymbol{x}| \le \|\boldsymbol{x}\|_2 + \|\boldsymbol{\beta}_i\|_2 \|\boldsymbol{x}\|_2 \le 2C_x.$$

For the second part of Eq. (74), we have

$$\left\|\left(\text{diag}\left(\boldsymbol{\beta}_i\right) - \boldsymbol{\beta}_i\boldsymbol{\beta}_i^\top\right)\boldsymbol{x}\boldsymbol{\beta}_i^\top\right\|_2 \le \left(\|\boldsymbol{\beta}_i\|_\infty + \left\|\boldsymbol{\beta}_i\boldsymbol{\beta}_i^\top\right\|_2\right)\left\|\boldsymbol{x}\boldsymbol{\beta}_i^\top\right\|_2$$
$$\le 2\left\|\boldsymbol{x}\boldsymbol{\beta}_i^\top\right\|_2 \le 2\|\boldsymbol{x}\|_2 \left\|\boldsymbol{\beta}_i^\top\right\|_2 \le 2C_x.$$

For the third part of Eq. (74), we have

$$\left\|\boldsymbol{\beta}_i\boldsymbol{x}^\top\left(\text{diag}\left(\boldsymbol{\beta}_i\right) - \boldsymbol{\beta}_i\boldsymbol{\beta}_i^\top\right)\boldsymbol{x}\right\|_2 \le \left\|\boldsymbol{\beta}_i\boldsymbol{x}^\top\right\|_2 \left\|\left(\text{diag}\left(\boldsymbol{\beta}_i\right) - \boldsymbol{\beta}_i\boldsymbol{\beta}_i^\top\right)\right\|_2 \|\boldsymbol{x}\|_2.$$

Next, we analyze the Hessian $\boldsymbol{H}_{Q,K}$, where each element is:

$$H_{Q,K}^{(j,p)} = \frac{\partial^2 f}{\partial w_Q^{(j)} \partial w_K^{(p)}} = \frac{\partial^2 f(\boldsymbol{x})}{\partial w_Q^{(j)} \partial w_Q^{(p)}} = \tau_0^2 \tau_1 \sum_{i=1}^{d_s} w_K^{(j)} w_Q^{(p)} (x^{(i)})^2 \boldsymbol{w}_V^\top \left(\boldsymbol{w}_O \circ \dot{\sigma}_r\left(\boldsymbol{w}_V \boldsymbol{x}^\top \boldsymbol{\beta}_i\right)\right) \boldsymbol{x}^\top \boldsymbol{\Gamma}_i \boldsymbol{x}.$$

Due to the symmetry, similar to $\boldsymbol{H}_{Q,Q}$, the spectral norm of $\boldsymbol{H}_{Q,K}$ can be bounded with probability at least $1 - 8e^{-d_m/2}$: $\|\boldsymbol{H}_{Q,K}\|_2 = \mathcal{O}(1/\sqrt{d_m})$.

Next, we analyze the Hessian $\boldsymbol{H}_{Q,V}$, where each element is:

$$H_{Q,V}^{(j,p)} = \frac{\partial^2 f}{\partial w_Q^{(j)} \partial w_V^{(p)}} = \tau_0 \tau_1 \sum_{i=1}^{d_s} w_K^{(j)} x^{(i)} w_O^{(p)} \dot{\sigma}_r\left(w_V^{(p)} \boldsymbol{x}^\top \boldsymbol{\beta}_i\right) \boldsymbol{x}^\top \left(\text{diag}(\boldsymbol{\beta}_i) - \boldsymbol{\beta}_i\boldsymbol{\beta}_i^\top\right) \boldsymbol{x}.$$

The spectral norm of $\boldsymbol{H}_{Q,V}$ can be bounded with probability at least $1 - 8e^{-d_m/2}$:

$$\|\boldsymbol{H}_{Q,V}\|_2 = \|\tau_0\tau_1 \sum_{i=1}^{d_s} x^{(i)}\dot{\sigma}_r\left(w_V^{(p)}\boldsymbol{x}^\top\boldsymbol{\beta}_i\right)\boldsymbol{x}^\top\left(\text{diag}(\boldsymbol{\beta}_i) - \boldsymbol{\beta}_i\boldsymbol{\beta}_i^\top\right)\boldsymbol{x}\boldsymbol{w}_K\boldsymbol{w}_O^\top\|_2$$

$$\leq |\tau_0\tau_1 \sum_{i=1}^{d_s} x^{(i)}\dot{\sigma}_r\left(w_V^{(p)}\boldsymbol{x}^\top\boldsymbol{\beta}_i\right)\boldsymbol{x}^\top\left(\text{diag}(\boldsymbol{\beta}_i) - \boldsymbol{\beta}_i\boldsymbol{\beta}_i^\top\right)\boldsymbol{x}|\|\boldsymbol{w}_K\boldsymbol{w}_O^\top\|_2 \quad \text{(Homogeneity of norm)}$$

$$\leq \tau_0\tau_1 \sum_{i=1}^{d_s} |x_i|\|\text{diag}(\boldsymbol{\beta}_i) - \boldsymbol{\beta}_i\boldsymbol{\beta}_i^\top\|_2\|\boldsymbol{x}\|_2^2\|\boldsymbol{w}_K\|_2\|\boldsymbol{w}_O\|_2 \quad \text{(Cauchy–Schwarz inequality)}$$

$$\leq \tau_0\tau_1 \sum_{i=1}^{d_s} |x_i|\|\text{diag}(\boldsymbol{\beta}_i) - \boldsymbol{\beta}_i\boldsymbol{\beta}_i^\top\|_2\|\boldsymbol{x}\|_2^2\left(3\sqrt{\eta_{QK}d_m} + R\right)\left(3\sqrt{\eta_O d_m} + R\right) \quad \text{(Lemma 20)}$$

$$= \mathcal{O}(1/d_m)\,,$$

where the last step is by Weyl's inequality and the range of the output of Softmax (from zero to one):

$$\left(\min_j(\beta_i^{(j)}) + \|\boldsymbol{\beta}_i\|_2^2\right)^2 \leq \|\text{diag}(\boldsymbol{\beta}_i) - \boldsymbol{\beta}_i\boldsymbol{\beta}_i^\top\|_2^2 \leq \left(\max_j(\beta_i^{(j)}) + \|\boldsymbol{\beta}_i\|_2^2\right)^2 = \mathcal{O}(1)\,.$$

It suffices to bound $\|\boldsymbol{\beta}_i\|_2$, i.e., $\|\sigma_s\left(x^{(i)}\boldsymbol{x}\boldsymbol{w}_Q^\top\boldsymbol{w}_K\right)\|_2$. Since the range of each element of the output of Softmax is from 0 to 1 and the sum of is one, $\|\boldsymbol{\beta}_i\| \leq 1$.

Next, we analyze the Hessian $\boldsymbol{H}_{Q,O}$, where each element is:

$$H_{Q,O}^{(j,p)} = \frac{\partial^2 f}{\partial w_Q^{(j)}\partial w_V^{(p)}} = \tau_0\tau_1 \sum_{i=1}^{d_s} w_K^{(j)}x^{(i)}w_V^{(p)}\dot{\sigma}_r\left(w_V^{(p)}\boldsymbol{x}^\top\boldsymbol{\beta}_i\right)\boldsymbol{x}^\top\left(\text{diag}(\boldsymbol{\beta}_i) - \boldsymbol{\beta}_i\boldsymbol{\beta}_i^\top\right)\boldsymbol{x}\,.$$

The spectral norm of $\boldsymbol{H}_{Q,V}$ can be bounded with probability at least $1 - 8e^{-d_m/2}$:

$$\|\boldsymbol{H}_{Q,O}\|_2 = \|\tau_0\tau_1 \sum_{i=1}^{d_s} x^{(i)}\dot{\sigma}_r\left(w_V^{(p)}\boldsymbol{x}^\top\boldsymbol{\beta}_i\right)\boldsymbol{x}^\top\left(\text{diag}(\boldsymbol{\beta}_i) - \boldsymbol{\beta}_i\boldsymbol{\beta}_i^\top\right)\boldsymbol{x}\boldsymbol{w}_K\boldsymbol{w}_V^\top\|_2$$

$$\leq |\tau_0\tau_1 \sum_{i=1}^{d_s} x^{(i)}\dot{\sigma}_r\left(w_V^{(p)}\boldsymbol{x}^\top\boldsymbol{\beta}_i\right)\boldsymbol{x}^\top\left(\text{diag}(\boldsymbol{\beta}_i) - \boldsymbol{\beta}_i\boldsymbol{\beta}_i^\top\right)\boldsymbol{x}|\|\boldsymbol{w}_K\boldsymbol{w}_V^\top\|_2 \quad \text{(Homogeneity of norm)}$$

$$\leq \tau_0\tau_1 \sum_{i=1}^{d_s} |x_i|\|\text{diag}(\boldsymbol{\beta}_i) - \boldsymbol{\beta}_i\boldsymbol{\beta}_i^\top\|_2\|\boldsymbol{x}\|_2^2\|\boldsymbol{w}_K\|_2\|\boldsymbol{w}_V\|_2 \quad \text{(Cauchy–Schwarz inequality)}$$

$$\leq \tau_0\tau_1 \sum_{i=1}^{d_s} |x_i|\|\text{diag}(\boldsymbol{\beta}_i) - \boldsymbol{\beta}_i\boldsymbol{\beta}_i^\top\|_2\|\boldsymbol{x}\|_2^2\left(3\sqrt{\eta_{QK}d_m} + R\right)\left(3\sqrt{\eta_V d_m} + R\right) \quad \text{(Lemma 20)}$$

$$= \mathcal{O}(1/d_m)\,.$$

Next, we analyze the Hessian $\boldsymbol{H}_{O,V}$. Each element of $\boldsymbol{H}_{O,V}$ is:

$$H_{OV}^{(p,j)} = \frac{\partial^2 f(\boldsymbol{x})}{\partial w_O^{(p)}\partial w_V^{(j)}} = \tau_1 \sum_{i=1}^{d_s} \dot{\sigma}_r\left(w_V^{(p)}\boldsymbol{\beta}_i^\top\boldsymbol{x}\right)\boldsymbol{x}^\top\boldsymbol{\beta}_i\mathbb{1}_{\{j=p\}}\,.$$

Note that $\boldsymbol{H}_{O,V}$ is a diagonal matrix. Consequently, the spectral norm of $\boldsymbol{H}_{O,V}$ is:

$$\|\boldsymbol{H}_{O,V}\|_2 = \max_{j\in[d_m]} |H_{O,V}^{(jj)}| \leq |\tau_1 \sum_{i=1}^{d_s} \boldsymbol{x}^\top\boldsymbol{\beta}_i| \leq \tau_1 \sum_{i=1}^{d_s} \|\boldsymbol{x}\|_2\|\boldsymbol{\beta}_i\|_2 = \mathcal{O}(1/d_m)\,.$$

Next, since each element of $\boldsymbol{H}_{O,O}$ and $\boldsymbol{H}_{V,V}$ is zero, the corresponding spectral norm is zero. Lastly, due to the symmetry of $\boldsymbol{w}_K$ and $\boldsymbol{w}_Q$, we can obtain the same bound for the corresponding Hessian block. Thus, the spectral norm of Hessian $\boldsymbol{H}$ is upper bounded by $\mathcal{O}(1/\sqrt{d_m})$, which completes the proof. $\square$

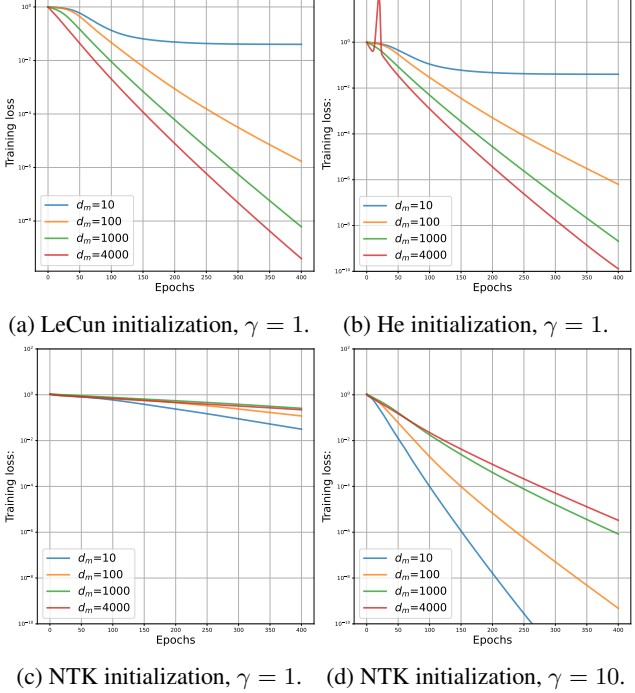

(a) LeCun initialization, $\gamma = 1$.    (b) He initialization, $\gamma = 1$.

(c) NTK initialization, $\gamma = 1$.    (d) NTK initialization, $\gamma = 10$.

Figure 5: Comparison of convergence curve for different initialization schemes. The convergence speed under Lecun/He initialization is generally faster than that of NTK initialization with the same step size.

## E    Additional details on experiments

### E.1    Additional result in Section 5.1

In this section, we present the convergence curve for different initialization schemes following the setup in Section 5.1. The results in Figures 5a to 5c show that when using the same step size $\gamma = 1$ for LeCun and He initialization exhibits similar behavior while NTK initialization leads to slower convergence. This is consistent with our theoretical finding. We also depict the result with larger step size $\gamma = 10$ for NTK initialization in Figure 5d, following our analysis in Appendix C.5.

### E.2    Additional result and set-up in Section 5.2

We have validated Assumption 3 on language data in the main paper. Here we will validate it on image dataset. Specifically, we choose MNIST dataset and consider the embedding in the ViT as $\boldsymbol{X}$, then we plot $\mathbb{P}\left(\left|\left\langle \boldsymbol{X}_n^\top \boldsymbol{X}_n, \boldsymbol{X}_{n'}^\top \boldsymbol{X}_{n'} \right\rangle\right| \geq t\right)$ as $t$ increases in Figure 6, where we could see exponential decay. The architecture of ViT is the same as in Section 5.2.

Additionally, in Figure 7, we can see that $d_m^{-1/2}$ setting achieves faster convergence in training loss and higher test accuracy. In Figure 8, we conduct the experiment on MNIST with regression task by using the same architecture. We take two classes $\{0, 1\}$ in MNIST for binary classification, the output of our regression is taken by $\text{sign}(f)$, and the training error is given by $\sum_{n=1}^{N}[\text{sign}(f(\boldsymbol{X}_n)) - y_n]^2$. We can see that the convergence speed of $d_m^{-1/2}$ is faster than that of $d_m^{-1}$. Overall, these separation results on both classification and regression tasks provide good justification for the $d_m^{-1/2}$ scaling.

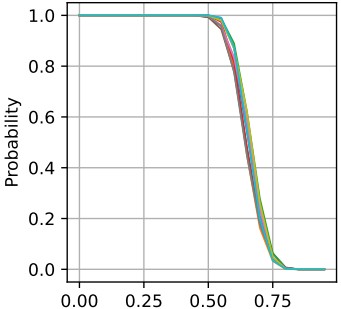

Figure 6: Verification of Assumption 3 on MNIST dataset.

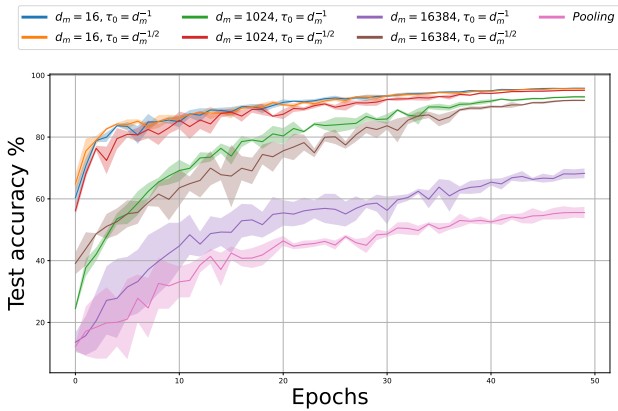

Figure 7: Test accuracy of classification task on MNIST dataset.

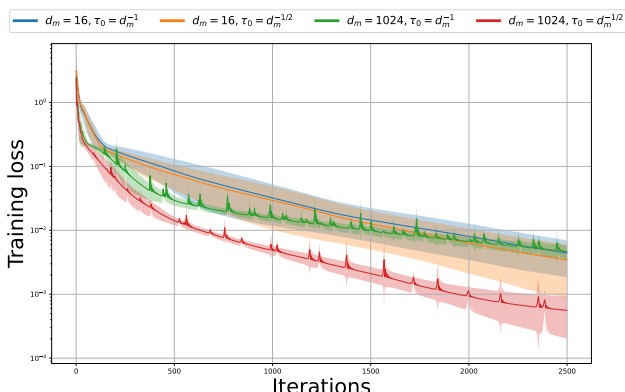

Figure 8: Regression task (with MSE loss) on MNIST dataset.

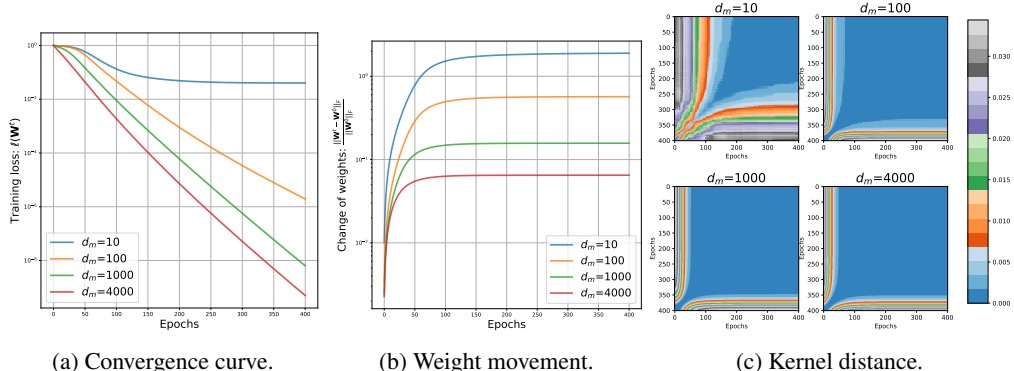

(a) Convergence curve.   (b) Weight movement.   (c) Kernel distance.

Figure 9: Experimental validation of the theoretical results on Transformers with $\tau_0 = d_m^{-1}$ scaling trained on synthetic data. (a) Linear convergence. (b) Rate of change of the weights during training. Observe that the weights change very slowly after the $50^{\text{th}}$ epoch. (c) Evolution of the NTK during the training. The result mirrors the plot (b) and demonstrates how the kernel varies significantly at the beginning of the training and remains approximately constant later. As the width increases, the NTK becomes more stable.

## F   Limitation

Firstly, in this work, we prove the global convergence guarantee for Transformer under different initialization schemes. Nevertheless, we only consider the gradient descent training under squared loss.

Our result does not cover SGD training and other loss functions, e.g., hinge loss and cross-entropy loss. Secondly, the model architecture being analyzed in this work is the encoder of Transformer which is widely used for regression problems or image classification problems. We do not consider the entire Transformer including the decoder, which is designed for sequence-to-sequence modeling. Lastly, our model does not cover the deep Transformer. We believe our analytic framework paves a way for further analyzing large-scale Transformer.

# G   Societal impact

This work offers a convergence analysis for the Transformer, which is a core element within contemporary large language models. Our theoretical analysis with realistic initializations and scaling schemes lays a theoretical foundation for the interested practitioner in the ML community to further study other priorities of Transformer such as generalization and in-context learning. We do not expect any negative societal bias from this work.

