# OpenReview forum: "On the Convergence of Encoder-only Shallow Transformers"
_NeurIPS.cc/2023/Conference — NeurIPS 2023 poster_

### Official Review · Reviewer_UvAq · 2023-07-04

**Soundness:** 3 good
**Presentation:** 3 good
**Contribution:** 2 fair
**Rating:** 5
**Confidence:** 4

**Summary:**

This paper proves the convergence rate of GD on one-layer transformer training. In their setting, the transformer is composed by a single attention layer, followed by relu unit and a fixed fully connected layer. The linear convergence rate is achieved. The proof technique follows the standard neural network proof [Nguyen 2021], where under some conditions of minimum singualr value of gram matrix, they inductively show that the parameter drift is controlled, and hence the GD dynamic is easy to study. The main meat is to show that the minimum singular value of gram matrix is lower bounded.


[Nguyen 2021] Nguyen, Quynh. "On the proof of global convergence of gradient descent for deep relu networks with linear widths." In International Conference on Machine Learning, pp. 8056-8062. PMLR, 2021.

**Strengths:**

To my best knowledge, this is the first convergence proof of transformer. The rate makes sense and proof roadmap is clear to me. The first part of the proof idea is standard, that under proper conditions for initialization and over-parameterization, the model parameter will stay in linear regime, and hence the convergence is easy to control. Then the second part is that they show that the conditions can be satisfied for some initialization scheme, by showing that the minimum singular value of gram matrix is bounded. The most interesting and novel part is to derive such lower bound.

**Weaknesses:**

1. As I mentioned above, the first part of the proof (proposition 1) pretty much follows [Nguyen 2021].

2. The model studied is bit toy. In practice, the residual block of transformer is necessary, but it is missing in this paper.

3. They assume the fully connected layer W_H is fixed to identical matrix, and argue that since W_V and W_H are adjacent, this is equavilent to W_H is trainable, However, in GD dynamic, I do not think they are equivalent. For example, in deep linear neural network, training all layer by GD is not equavilent to only training one matrix. Or if they are equivalent, there should be rigorous proof.

**Questions:**

1. Since original framework in [Nguyen 2021] works when all layers are jointly trained, is it possible to extend your proof to that regime as well?

2. In Corollary 1, you said that when d_s and d is chosen to be some value, the conditions 2 and 3 cannot satisfied, and the convergence will fail. I think this is not a good reasoning. It could be that your framework cannot prove the convergence, but does not mean it cannot actually converge. A more convincing reasoning will be showing the lower bound of the loss, to show the failure of convergence.

**Limitations:**

The technical limitations are stated in weakness part. I do not see any negative social impact of this work.

---

> ### Author Rebuttal · Authors · 2023-08-09
>
> We are thankful to the reviewer UvAq for appreciating that this is the first theoretical work on the convergence of Transformers and the insightful feedback. We address the concerns below.
>
> ---
> > **Q1:** [The first part of the proof (proposition 1) pretty much follows [3].]
>
> **A1:** The framework of the proof for proposition 1 follows the variant of Polyak-Lojasiewicz (PL) inequality in traditional convergence analysis. But the studied structure is generally more complicated than [3] due to the softmax function, and the proof in Lemma 7-11 tailored for Transformer.
>
> More importantly, to give the lower bound for $\alpha_0$ in proposition 1 is another technical difficulty, which was later addressed in Theorem 1 when considering several particular initializations. See the [general response](https://openreview.net/forum?id=8ZveVHfmIE&noteId=hT7qrLWaaA) for detailed clarification about this issue.
>
>
> ---
> > **Q2:** [The model studied is bit toy. In practice, the residual block of Transformer is necessary, but it is missing in this paper.]
>
> **A2:** We are thankful for the suggestion of the reviewer. According to the reviewer’s suggestions, we have already extended our result to a shallow Transformer with residual connections, see [general response](https://openreview.net/forum?id=8ZveVHfmIE&noteId=hT7qrLWaaA).
>
> We hope our analysis of the self-attention module (e.g., scaling, softmax) will lay a foundation for the analysis of more general Transformer architectures.
>
>
> ---
> > **Q3:** [They assume the fully connected layer W_H is fixed to the identical matrix, and argue that since W_V and W_H are adjacent, this is equivalent to W_H is trainable, However, in GD dynamic, I do not think they are equivalent. Since original framework in [3] works when all layers are jointly trained, is it possible to extend your proof to that regime as well?]
>
> **A3:** We agree with the reviewer that these two cases are not equivalent under gradient descent. We adopt one learnable parameter for training just for ease of analysis, as mentioned in line 138. The same technique is also used in [1] for analyzing Transformer. According to your suggestions, we will make it clear in our final version by adding the following sentence in line 139.
>
> "Note that we mix $W_V$ and $W_H$ together for ease of the analysis but it does not mean its training dynamics is the same as we jointly train these two adjacent matrices."
>
> Besides, we think it is possible to extend the proof and we illustrate the high-level idea here. If we consider all layers have been trained. Then we need to consider one more parameter $\boldsymbol{W}_H$.  Two main parts of the analysis have to be changed. The first part is Proposition 1. Note that in line 628, $\boldsymbol{f}$ becomes as follows:
>
> $\boldsymbol{f}= \tau\_1
> \boldsymbol{w}\_O^\top
> \sum_{i=1}^{d\_s}\sigma\_r
> \left(\boldsymbol{W}\_H \boldsymbol{W}\_V\boldsymbol{X}^\top \boldsymbol{\beta}\_i\right)$.
>
> Then lemma 7- 8, 10, and 11 remain unchanged.
>  In Lemma 9, one more step of triangle inequality is required to decouple $\boldsymbol{W}\_H^{t’}$ and  $\boldsymbol{W}\_H^{t}$.
> In Lemma 12, 13, 14, we need an additional bound of the gradient norm and the Lipschitz constant for $\boldsymbol{W}\_H$ using the same technique as we bound the others. Then the proof for Proposition 1 can be finished. The second part that needs to be changed is the minimum singular value of $\boldsymbol{F}\_{pre}$. One can analyze the centering features w.r.t. $\boldsymbol{W}\_H$ following the strategy in [2] and give a lower bound for the minimum singular value of $\boldsymbol{F}\_{pre}$.
>
> ---
>
> > **Q4:** [Fail of convergence in Corollary 1.]
>
> **A4:** Our current proof provides an upper bound of the loss function value. To show “fail of convergence”, we agree with the reviewer that the lower bound of loss function is needed. This is a quite difficult problem, and accordingly, based on the reviewer’s suggestions, we modify the corollary 1 for rigorous illustration as follows:
>
> ---
>
> **Corollary** (Convergence of vector input)
>
> Considering LeCun initialization with $\tau_0 = d_m^{-1}$ scaling,
> given vector input $\boldsymbol{x}\in \mathbb{R} ^{\tilde{d}}$,
> if one feeds the input to Transformer by setting $d_s = 1, d=\tilde{d}$, then
> training with GD can converge to a global minimum. However, if one sets $d_s =\tilde{d}, d=1$,
> the conditions in Eq.2-3 do not hold, **its convergence cannot be guranteed by our theory**.
>
> ---
>
> Though the estimation (lower bound of the loss function) under Transformer is difficult, we find that it is possible to consider a simplified case, i.e., linear regression, which still shares a similar spirit to our problem.
>
> ---
>
> Consider the loss $\ell = 1/2 || \boldsymbol{y} - \boldsymbol{X} \boldsymbol{w} ||_2^2 $, where $\boldsymbol{X} \in \mathbb{R}^{N \times d}$. In this case, the neural tangent kernel (NTK) is $\boldsymbol{X} \boldsymbol{X}^\top$. If the NTK is a rank-one matrix, then the minimum eigenvalue is zero. In this case one can see $\boldsymbol{X} \in \mathbb{R}^{N \times 1}$. Therefore, if the augmented matrix $[\boldsymbol{X}, \boldsymbol{y}]$ is not rank-one, which is standard in practice, then GD training can not converge to zero loss since there is no solution.
>
> ---
>
> This would motivate us to rethink Corollary 1 under the Transformer setting as future work. We will add a remark about this in our revised version.
>
> ---
> We hope that our responses have addressed the concerns of the reviewer. If there are any remaining questions, we are happy to discuss them further.
>
> ---
> ### References
> [1] Yang, et al. "Transformers from an optimization perspective." Neurips 2022.
>
> [2] Nguyen, et al. "Tight bounds on the smallest eigenvalue of the neural tangent kernel for deep relu networks." ICML, 2021.
>
> [3] Nguyen, et al. "On the proof of global convergence of gradient descent for deep relu networks with linear widths." ICML, 2021.

---

> > ### Comment · Reviewer_UvAq · 2023-08-12
> > **Thanks for the rebuttal**
> >
> > I would like to thank the authors for the clarification. The results in the paper is timely but given the dependence on previous work, I decide to maintain my score.

---

### Official Review · Reviewer_sZxG · 2023-07-05

**Soundness:** 3 good
**Presentation:** 2 fair
**Contribution:** 3 good
**Rating:** 6
**Confidence:** 3

**Summary:**

This paper studies transformer networks consisting of one layer, with average pooling and a scalar output.  The authors consider encoder type softmax attention (unmasked) and show convergence to a global minimizer of the loss.
The results hold for the cases that the temperature/scaling inside the softmax $\tau_0$ is either inversely proportional to the overparameterization of the weight matrices, or the square root of it. The results also hold for different types of initialization He/Le-Cun.
Finally, the authors also provide an NTK based analysis.

**Strengths:**

To the best of my knowledge this is the first theoretical work on the convergence of transformers and could potentially lead to further investigate the convergence properties of transformers and bring insights into the way they work.  It is also interesting that the authors study different initialization schemes and provide comparisons between them.

**Weaknesses:**

Except for some questions I have (see below), my main concern is the formulation of the transformer model in this analysis. Specifically,

1. The paper does not study sequence to sequence models as encoder transformers normally are. Thus, I find the formulation of outputting a scalar inconsistent with practice.
2. The ReLU layers do not have any matrix after the application of the ReLU. There are also no residual connections.
3. The pooling layer to the best of my knowledge is used only in ViT models.

I also think that it would be beneficial for the paper, to include more details about proof techniques for one of the cases (this could be possible by reducing some of the text in the rest of the sections).



**Questions:**

1. In lines 635,637, do the authors mean the triangle inequality?
2. Why the authors study these two specific scaling schemes of $d_m^{-1/2}$ and $d_m^{-1}$? It would be interesting to have an analysis dependent on the scaling parameter $\tau_0$.
3. It is not very clear also what step-sizes could be permitted or constants $\hat{c}$. To my understanding the step-size needs to be inversely proportional to the number of data samples? It would be great if the authors could include some examples, in which the probability bounds are not vacuous and the order of the step-size chosen.
4. It would be great if the authors could include also the order of the constants $C_Q,C_K,C_V,C_O$ (if they cannot be chosen arbitrarily) or if they could point out the line in which they are defined.
5. In line 640, how is it implied that $\sigma_{min}(F_{pre}) \geq \alpha/2$?



**Limitations:**

This work has no potential negative societal impact.

---

> ### Author Rebuttal · Authors · 2023-08-09
>
> We are thankful to the reviewer sZxG for appreciating that this is the first theoretical work on the convergence of Transformers and its significance for practice.
>
> ---
>
> > **Q1:** [No result on sequence to sequence (seq2seq) model. The formulation of outputting a scalar inconsistent with practice. The pooling layer to the best of my knowledge is used only in ViT models. No residual connections.]
>
> **A1:** We agree with the reviewer that the current formulation does not consider seq2seq Transformer. Our model formulation as well as theoretical analysis is motivated by ViT models used for image tasks. Besides, according to the reviewer’s suggestions, our result can be extended to a shallow Transformer with residual connection, see [general response](https://openreview.net/forum?id=8ZveVHfmIE&noteId=hT7qrLWaaA).  We hope our analysis of the self-attention module (e.g., scaling, softmax) will lay a foundation for the analysis of more general Transformer architectures.
>
> ---
>
> > **Q2:** [Include more details about proof techniques for one of the cases (this could be possible by reducing some of the text in the rest of the sections).]
>
> **A2:** We are thankful to the reviewer for the suggestion, we will add the proof sketch, as present in the [general response](https://openreview.net/forum?id=8ZveVHfmIE&noteId=hT7qrLWaaA), into the final version.
>
> ---
>
> > **Q3:** [In lines 635,637, do the authors mean the triangle inequality? In line 640, how is it implied that $\sigma_{min}(F_{pre}) \geq \alpha/2$]
>
> **A3:**  We use Weyl’s inequality, (a general version of triangle inequality) and the relationship between spectral norm and Frobenius norm. To be specific,
>
>
> [lemma] [Weyl’s inequality [6]]
>
>
> Given two matrices $X, Y \in \mathbb{R}^{p\times q}$ with the respective singular value ordered as follows: $\sigma_1(X)\geq\ldots\geq\sigma_r(X)$ and $\sigma_1(Y)\geq\ldots\geq\sigma_r(Y)$, where $r=\min(p,q).$ Then we have $\max_{i\in[r]} |\sigma_i(X)-\sigma_i(Y)|\leq ||X-Y||_2.$
>
>
> [end lemma]
>
>
> In lines 635 and 637, we choose $i = 1$ in Weyl’s inequality and use the inequality between spectral norm and Frobenius norm to obtain $||X||_2 \le ||Y||_2+ ||X-Y||_2 \le ||Y||_2 + ||X-Y||_F$. Thus, this can be also considered as a triangle inequality for the spectral norm.
>
>
> In line 640, we choose $i = r$ in Weyl’s inequality and use the inequality between spectral norm and Frobenius norm to obtain $\sigma\_{\min}(X) \ge \sigma\_{\min}(Y) - ||X-Y||_2 \ge \sigma\_{\min}(Y) - ||X-Y||_F$.
>
> ---
>
> > **Q4:** [Why the authors study these two specific scaling schemes of $d_m^{-1/2}$ and $d_m^{-1}$? It would be interesting to have an analysis dependent on the scaling parameter$\tau_0$ .]
>
> **A4:** These two scaling schemes are widely used in practice ($d_m^{-1/2}$ in [4,5]), and theoretical analysis for the neural network limiting ($d_m^{-1}$ in [1,2,3,7]). Our proof framework can facilitate the analysis of general $\tau_0$, e.g., $\tau_0 = d_m^{-a}$ with $a \in [0,1]$.
>
> ---
>
> > **Q5:** [It is not very clear also what step-sizes could be permitted or constants . To my understanding the step-size needs to be inversely proportional to the number of data samples? It would be great if the authors could include some examples, in which the probability bounds are not vacuous and the order of the step-size chosen.]
>
> **A5:** The step size is inversely proportional to $N^{1/2}$, where N is the number of data samples. To see why, in line 652, we define $C =  c_1 c_2 + 2 c_3 \ell(\theta^0)$. According to line 648 , we can see that $c_1$ is proportional to $\rho$, which is defined as $\rho = N^{1/2} d_s^{3/2} \tau_1 C_x $ in line 170.  Therefore, C is proportional to N^{½}. Since in Proposition 1, we require the step-size $\gamma < 1/C$, therefore, the step-size is inversely proportional to $N^{1/2}$. We will add a remark for this in our final version.
> The probability is non-vacuous since in Lemma 6 we select $\zeta = d_m^{-1/2}$ so that the probability is at least $1-2\exp^{-d_m/2}$. In practice, even for $d_m = 16$, then the probability is at least 0.99$. Larger width leads to a higher probability.
>
> ---
>
> > **Q6:** [It would be great if the authors could include also the order of the constants $C_Q, C_K, C_V, C_O $ (if they cannot be chosen arbitrarily) or if they could point out the line in which they are defined.]
>
> **A6:** $C_Q = C_K = C_V = C_O$ is in constant order, as can be seen in line 673 (appendix). We directly choose one for simplicity. We will make it clear in our final version.
>
> ---
>
> We hope that our responses have addressed the concerns of the reviewer. If there are any remaining questions, we are happy to discuss them further.
>
> ---
>
> ### References
> [1] Du, et al. "Gradient Descent Provably Optimizes Over-parameterized Neural Networks." ICLR, 2019.
>
> [2] Jacot, et al. "Neural tangent kernel: Convergence and generalization in neural networks." NeurIPS, 2018.
>
> [3] Hron, et al. "Infinite attention: NNGP and NTK for deep attention networks." ICML, 2020.
>
> [4] Vaswani, et al. "Attention is all you need." NeurIPS, 2017.
>
> [5] Dosovitskiy, et al. "An Image is Worth 16x16 Words: Transformers for Image Recognition at Scale." ICLR, 2021.
>
> [6] Stewart, Gilbert W. Perturbation theory for the singular value decomposition. 1998.
>
> [7] Yang, et al. "Feature learning in infinite-width neural networks." ICML, 2021.

---

> > ### Comment · Reviewer_sZxG · 2023-08-13
> > **Response to authors**
> >
> > I would like to thank the authors for their response. After taking the time rereading the paper, I think that it is important to be rewritten such as to be more friendly to the reader. Specifically, to my opinion
> > 1. Previous results should be at least cited if not restated so that the reader could easily find them.
> > 2. The appendix should be self-contained. For example, terms like $f_{pre}$ and $F_{pre}$ should be restated and not used from the main text.
> > 3. The lemmas in the appendix are entangled with each other, creating a sense of confusion. The statements should be more clear and concise.
> >
> > These are some comments that I hope to help the authors improve the writing of the paper. My main concern remains the formulation of the transformer architecture as well as the technical contribution of the paper raised by reviewer UvAq. However, I am raising my score to 6 given that the authors will create a revised version.

---

> > > ### Author Response · Authors · 2023-08-14
> > > **Response to reviewer sZxG**
> > >
> > > We are thankful to the reviewer sZxG for the prompt response and positive support.  According to your suggestions, we will re-organize related work and appendix for better understanding. To be specific,
> > > -  [Related work] The subsection ‘over-parameterization for convergence analysis’ in the related work will be elaborated with more details to describe Table 3 of the appendix.
> > > - [Self-contained appendix] We will include a more detailed notation in Table 2. We will also restate the meaning of each symbol if it was only mentioned in the main body.
> > > - [Entanglement of the lemmas in appendix]. We will add more details about theorems and lemmas, and a proof flowchart to point out the relationship between each lemma.
> > >
> > > Besides, the technical difficulty and proof sketch will be added to the main text for better understanding, to make our contributions more clear.
> > >
> > > We appreciate the feedback from the reviewer and we are open to more suggestions to improve the readability and contribution of our work.
> > >
> > > Best regards,
> > >
> > >
> > > Authors

---

### Official Review · Reviewer_p3n5 · 2023-07-06

**Soundness:** 2 fair
**Presentation:** 2 fair
**Contribution:** 3 good
**Rating:** 5
**Confidence:** 3

**Summary:**

This paper presents global convergence results of shallow transformers (one self-attention layer followed by an MLP layer). The authors consider two different scalings for the factor $\tau_0$ used in the attention matrix as a function of $d_m$, the number of rows of $W_Q$ and $W_K$. Specifically, they prove that under LeCun/He/NTK initializations for the parameters of the Transformer, scaling  $\tau_0$ as $d_m^{-1/2}$ requires quadratic over parametrization ($d_m = \Omega(N^2) $) to achieve convergence, while the $d_m^{-1}$ requires only linear over parametrization ($d_m = \Omega(N) $), though the later has a worse convergence rate. The paper claims that it is due the the lack of ability to capture pairwise interaction in the  $\tau_0 = d_m^{-1}$ scaling. Numerical experiments are presented on synthetic data to corroborate their theoretical findings for the $d_m^{-1/2}$  scaling, as well as on MNIST, where the impact of choosing $\tau_0 = d_m^{-1/2}$ or $\tau_0 = d_m^{-1}$ on the training dynamics is discussed.

**Strengths:**

1. This papers tackles the significant question of global convergence of Transformers.

2. The model considered is a realistic one, where both the attention module (involving pair-wise interaction between the $X^{i}$), and the MLP (applied independently on each Self-attention$(X)^{i}$ are considered (Eqs (1.1) and (1.2)).

3. The assumptions on the data-distribution are clearly discussed and Assumption 3. is verified in practice. Likewise, the assumptions of the parameters of the Transformer are clearly discussed.

4. Theorems 1 et 2 are strong results proving the global convergence of gradient descent of the shallow Transformer model for the general regression task studied in the paper.

5. The paper is focussed on the choice of the scaling factor inside the attention module, which is a forward steps towards understanding its impact. The theoretical findings as well as the experiments of the paper underlines the advantage of using the $d_m^{-1/2}$ scaling.

6. Numerical experiments are conducted to support the theoretical claims.

**Weaknesses:**

1. Presentation:

a. In my opinion the biggest weakness of the paper is its lack of clarity. The paper is not really well written, hard to read, with typos and missing words (e.g. l. 37 -42, 129, 152, 222). The paper would benefit from being rewritten in a clearer and more pedagogical manner.

2. Theoretical results:

a. I read in Table 3 that SoftMax + ReLU leads to linear over parametrization. From the paper, we understand that this requires the scaling $d_m^{-1}$. However, as the authors remark it, this implies that the SoftMax operator will behave as a pooling layer. The link between the linear over-parametrization and the attention module behaving as a pooling layer is not clear.

b. Lines 68 to 71. It is not clear whether the mentioned ReLU acts on each element of the sequence separately or mixes them together.

3. Experiments:

a. The experiment on the synthetic data in the main paper only focuses on the $d_m^{-1/2}$ scaling. I see that  the authors provide the results for the $d_m^{-1}$ scaling in the appendix. A discussion on the impact of the chosen scaling on the performance of the model on the synthetic data experiment is missing in the main paper.

b. The paper focuses on regression while the experiment on MNIST seems to be a classification task. What plays the role of $y_n$ in this experiment ?

**Questions:**

1. The quartic dependency of $N$ in the dimension $d$ is not discussed in Theorem 2. Is this a realistic assumption ?

2. Could the authors provide insights on why the $d_m^{-1}$ scaling, which leads the Attention module to behave as a pooling layer, requires only linear over parametrization ?

3. What happens if the Attention module is explicitly replaced by a pooling layer, that is $A_1 = ( 1 / d_s ) \times 1_{d_s \times d_s}X W_V^T$?

4. Could the authors comment the results of the synthetic data experiment for the $d_m^{-1}$ scaling ? And compare with the $d_m^{-1/2}$ scaling ?

5. Could the authors provide additional details for the MNIST experiment ?  Is it a classification task ? What is the test-accuracy of the model ? What is the training loss when setting the attention module to the identity ?

6. Is there a setup where we should chose the $d_m^{-1}$ scaling ?

**Limitations:**

Yes (Appendix F).

---

> ### Author Rebuttal · Authors · 2023-08-09
>
> We thank the reviewer p3n5 for the insightful feedback and for appreciating the significance of the paper. We address the concerns below.
>
> ---
> > **Q1:** [The biggest weakness of the paper is its lack of clarity.]
>
> **A1:** We appreciate the reviewer pointing out some typos. According to your suggestions, we have already corrected typos and rephrased the expression of this paper in the final version for better understanding. For example, line 129 is changed as follows:
>
> "We consider the encoder of Transformer, which can be applied to regression or classification ~~task~~ **tasks**."
>
> Apart from this, we may kindly remind the reviewer that
> - Reviewer YNcT agrees that ‘The paper is organized, well-written, and easy to follow in general.’
> - Reviewer n1Tb and 9ERM give the score of 4 (excellent) for the presentation.
>
> We hope the revised version would have a good presentation from the reviewer's side and will appreciate it if the reviewer can reconsider this “**biggest weakness**” for evaluation.
>
> ---
> > **Q2:** [The link between linear over-parametrization and attention module behaving as a pooling layer is not clear.]
>
> **A2:** Due to the $d_m^{-1}$ scaling, the attention module behaves similarly to a pooling layer according to the law of large numbers, as mentioned in the introduction. In this case, the nonlinearity on $X$ disappears and thus the minimum eigenvalue of ${\Phi} {\Phi}^\top$ can be estimated via $XX^{T}$, see lines 698 and 704 for detail.
> Accordingly, this leads to the minimum eigenvalue in the order of $\Theta(N/d)$, and thus linear over-parameterization is enough, see Lemma 17 for detail. We will make it clear in our final version.
>
> ---
> > **Q3:** [Not clear: ReLU acts on each element of the sequence separately or mixes them together (Lines 68 to 71.)]
>
> **A3:** Throughout the paper, ReLU is applied for each element of the sequence. In line 68, the self-attention layer is changed to one ReLU layer, i.e., in Eq. (1.1) we have ${A}_{1} = max(0, {X} {W}_1^{T})$ in an element-wise way.
>
> ---
> > **Q4:** [The quartic dependency of $N$ in the dimension $d$  is not discussed in Theorem 2. Is this a realistic assumption?]
>
> **A4:** It can be achieved in practice in some cases. For example, the embedding of the tokens is usually chosen from 8-D to 1024-D [3]. Therefore, it is realistic when the token embedding is small. More importantly, for $d_m^{-1/2}$ scaling, our analysis only requires $N>d$, which is more general and valid for practical datasets.
>
> ---
> > **Q5:** [What happens if the attention module is replaced by a pooling layer?]
>
> **A5:** We answer this question both empirically and theoretically:
> - In practice: According to the reviewer’s suggestion, we also conduct the classification task (using cross-entropy loss) on MNIST of the shallow Transformer as well as the setting that the attention module is identity. As shown in Fig. 2 (training loss) and Fig. 3 (test accuracy) in the one-page pdf,  training loss increases, and test accuracy drops.
> - In theory: On one hand, the network after replacing the self-attention can be considered as a 2-layer fully-connected ReLU network that still requires linear quadratic-parametrization under LeCUN initialization. On the other hand, we should note that the considered architecture includes a softmax layer and a 2-layer fully-connected ReLU network,  if the self-attention mechanism is substituted by a fully-connected ReLU layer, it will require cubic over-parameterization for global convergence, as discussed in the paper.
>
>
> ---
> > **Q6:** [More experimental details: regression and classification, synthetic data and MNIST, different scalings.]
>
> **A6:** In our work, $y_n$ is the classification label of MNIST but regression on MNIST is still doable via the MSE loss. Taking two classes of MNIST for binary classification as an example, the output of our regression is taken by $sgn(f)$ and the training error is given by $\sum_{n=1}^N [sgn(f(x_n)) - y_n]^2$. This is commonly used in theory via the MSE loss, e.g., [4,5].
> - Regression results: there is no significant difference between these two scaling schemes under this synthetic data. The task is too simple so the optimization process is easy. But for a complex regression task, e.g., regression on MNIST, as shown in Fig.4 in the one-page pdf, the convergence speed of $d_m^{-1/2}$  is faster than $d_m^{-1}$.
>
> - Classification results: According to the reviewer’s suggestion, we also conduct the classification task (using cross-entropy loss) on MNIST of the shallow Transformer. As shown in Fig. 2 (training loss) and Fig. 3 (test accuracy) in the one-page pdf, the $d_m^{-1/2}$ setting achieves faster convergence in training loss and higher test accuracy.
>
> These separation results on both classification and regression tasks provide good justification for the $d_m^{-1/2}$ setting. We will add these details about the experiments in our revised version. The source code has already been sent to AC.
>
> ---
> > **Q7:** [Is there a setup where we should choose $d_m^{-1}$?]
>
> **A7:** Yes. The analysis for the scaling $d_m^{-1}$ in Transformer arises from NTK literature [1,2] from the theoretical side. As mentioned in A6, the $d_m^{-1}$ scaling is a feasible choice under a small $d_m$ setting (or some simple tasks as mentioned in A6) in practice.
>
> We hope that our responses have clarified the questions of the reviewer. If there are any remaining questions, we are happy to discuss them further.
>
> ---
>
> ### References
> [1] Yang, et al. "Feature learning in infinite-width neural networks." ICML, 2021.
>
> [2] Jacot, et al. "Neural tangent kernel: Convergence and generalization in neural networks." NeurIPS, 2018.
>
> [3] see tensorflow website: the guide of **word_embeddings**
>
> [4] Rahaman, et al. "On the spectral bias of neural networks." ICML, 2019.
>
> [5] Liao, et al. "A random matrix analysis of random fourier features: beyond the gaussian kernel, a precise phase transition, and the corresponding double descent." NeurIPS, 2020.

---

> > ### Comment · Reviewer_p3n5 · 2023-08-16
> >
> > Thanks a lot for your rebuttal and clarifications.
> >
> > After re-reading the paper in details, I still believe that there is a lack of clarity in both the formulation and the way the results are presented. These are not just minor typos.
> >
> > **Formulation**
> >
> > Some sentences are not grammatically correct: e.g.:
> >
> > l. 228: it is common to make either the assumption that the covariance of x is an identity matrix or weak assumption, e.g., positive definite.
> >
> > l. 245:  Besides, the stability of NTK during training, which allows us to build the connection to kernel regression predictor.
> >
> > There are other examples. It makes things harder to understand for the reader.
> >
> > **Presentation of the results**
> >
> > I still believe the paper lacks pedagogy in presenting the results. For instance:
> >
> > 1. Proposition 1 has many notations and deserves some comments for the reader.
> >
> > 2. In 4.4 you propose an NTK analysis. But in 4.2 and 4.3 you also consider the NTK initialization under the two scalings for $\tau_0$, which is confusing.
> >
> > 3. You don't specify which scaling you use in Th.3 and 4.
> >
> > 4. The learning algorithm is never properly introduced.
> >
> > Therefore, my opinion is that this paper deserves to be rewritten, where formulation has been revised everywhere and results are presented in a clearer way. I strongly believe that this would be beneficial for the paper which is a good theoretical contribution.

---

> > > ### Author Response · Authors · 2023-08-17
> > > **improve the presentation according to Reviewer p3n5's writing suggestions**
> > >
> > > We thank the reviewer p3n5 for the constructive suggestions on writing. We have already corrected the grammatical errors, and imprecise expressions, in order to improve the readability.
> > >
> > > Indicatively, we list the revisions to the reviewer’s comments below:
> > >
> > > ---
> > >
> > > -  l. 228: it is common to make either the assumption that the covariance of x is an identity matrix or weak assumption, e.g., positive definite.
> > >
> > > -> Frequently, the covariance of x is assumed to be an identity matrix or positive definite.
> > >
> > > -  l. 245: Besides, the stability of NTK during training, which allows us to build the connection to kernel regression predictor.
> > >
> > > -> Besides, the stability of NTK during training allows us to build a connection on training dynamics between the Transformer (assuming a squared loss) and the kernel regression predictor.
> > >
> > > ---
> > >
> > > Regarding the presentation, according to your suggestions, we will introduce more descriptions when presenting our theoretical results. To be specific:
> > >
> > > ---
> > >
> > > **Description of Proposition 1:** We will modify lines 170-171 for clarity as follows:
> > >
> > > Define the following quantities at initialization for simplification:
> > > - The norm of the parameters: $\bar{\lambda}_Q = XXX$, $\bar{\lambda}_K= XXX$, $\bar{\lambda}_V= XXX$, $\bar{\lambda}_O= XXX$
> > > - Two auxiliary terms:  $\rho = XXX$ and $z = XXX$.
> > >
> > > Under Asm. 1, we assume that the minimum singular value of ${\bf F}\_{\text{pre}}^0$, i.e., $\alpha \triangleq \sigma\_{min}({\bf F}\_{\text{pre}}^0$) satisfies the following condition at initialization
> > >
> > > ---
> > >
> > > These variables will be included in Table 2 for better reference. Besides, more remarks will be added for a better understanding, e.g., how $\alpha$ is affected by different initialization schemes and the selection of the step-size.
> > >
> > > ---
> > >
> > > **Confusion on “NTK initialization/scaling” name:** The title of Sec. 4.4 will be modified as follows: 'NTK Analysis of Transformers' -> NTK Analysis with $\tau_0 = d_m^{-1} $' for better understanding.
> > >
> > > ---
> > >
> > > **Scaling used in Thm 3. and 4:**  In these two theorems (as well as Section 4.1), the $\tau_0 = d_m^{-1}$ setting is employed. We will make it clear.
> > >
> > > ---
> > >
> > > **No description on learning algorithms:** We will include the following standard gradient descent algorithm for Transformer training in the revised version. It involves gradient updates for Transformer parameters ${\bf W}_Q$, ${\bf W}_K$, ${\bf W}_V$, ${\bf W}_O$ at the $t$-step: (denote ${\bf \theta}^0 := \\{
> > > {\bf W}\_Q^0, {\bf W}\_K^0, {\bf W}\_V^0, {\bf W}\_O^0
> > > \\}$ for simplicity):
> > >
> > > - ${\bf W}\_Q^{t+1} = {\bf W}\_Q^{t}- \gamma\cdot \nabla\_{{\bf W}\_Q} \ell ({\bf \theta}^t)$
> > > - ${\bf W}\_K^{t+1} = {\bf W}\_K^{t}- \gamma\cdot \nabla\_{{\bf W}\_K} \ell ({\bf \theta}^t)$
> > > - ${\bf W}\_V^{t+1} = {\bf W}\_V^{t}- \gamma\cdot \nabla\_{{\bf W}\_V} \ell ({\bf \theta}^t)$
> > > - ${\bf W}\_O^{t+1} = {\bf W}\_O^{t}- \gamma\cdot \nabla\_{{\bf W}\_O} \ell ({\bf \theta}^t)$
> > >
> > > ---
> > >
> > > Apart from the aforementioned revision, we will add the proof sketch, more details on theoretical results and experimental settings, , which has been mentioned in the previous response and [general response](https://openreview.net/forum?id=8ZveVHfmIE&noteId=hT7qrLWaaA). We will try our best to improve the readability of this paper and hope our clarifications address your concern.
> > >
> > > ---
> > >
> > > We sincerely appreciate the reviewer’s proofreading, which will definitely be beneficial to our paper.

---

> > > > ### Comment · Reviewer_p3n5 · 2023-08-18
> > > > **Increasing my score to 5.**
> > > >
> > > > Dear authors,
> > > >
> > > > Thanks for the clarification. Given your commitment to significantly improve the writing and presentation of the results, I increase my score to 5.

---

> > > > > ### Author Response · Authors · 2023-08-18
> > > > >
> > > > > Dear Reviewer p3n5,
> > > > >
> > > > > Thanks for your support.
> > > > > We will include these writing suggestions in our final version for better readability.
> > > > >
> > > > > Best regards,
> > > > >
> > > > > Authors

---

### Official Review · Reviewer_9ERM · 2023-07-06

**Soundness:** 4 excellent
**Presentation:** 4 excellent
**Contribution:** 4 excellent
**Rating:** 7
**Confidence:** 3

**Summary:**

The transformer network is a popular but complicated architecture, which are widely used in NLP and computer vision. The contribution of this paper is two-folded:
1. it aims to give convergence guarantee for a shallow transformer network (encoder part) under different weight initialisations. Also, the paper concludes that the convergence rate of the NTK initialisation is slower than the original (LeCun/He) initialisation, which is often used in realistic setting;
2.  it shows that a quadratic over-parametrisation guarantees global convergence of shallow transformer in usual setting.

Assumptions are verified by realistic examples. Convergence and training dynamics are also tracked in both synthetic and real-world data sets.

**Strengths:**

This paper gives a standard approach to proof global convergence by bounding the smallest singular value of the weight matrix in the last linear layer in the transformer network from above. The proof is delivered in a clear way, where notations and motivations are clearly stated. The lines seem to be flawless as I cannot find any typos or mistakes.

This paper clearly has its significance in showing global convergence of transformer network in the original (LeCun/He) initialisation for the first time. Also, the over-parametrisation guarantee for global convergence is original, as far as I know. It also serves as a framework for later analysis in transformer.


**Weaknesses:**

The paper only considers the shallow (encoder part of the) transformer network and training in gradient descent, which is still far away from practical architecture. Also, it would be good if I could see the comparison between theoretical and empirical convergence in a plot. Sadly, the plots only support the theorems qualitatively but not quantitatively.

**Questions:**

The quantity $\alpha$ in Proposition 1seems to be sub-optimal. Is it the reason why you do not plot the theoretical decay with the empirical one? Could a log-scale plot be convincing?

Also, in the proof of Lemma 1, you say by LLN we have the second equality in line (67). But the entries in $\beta'_j$ is correlated to these in $\beta_i$. How could you replace each vector respectively with $\frac{1}{d_s}\bm{1}_{d_s}$  by taking the limit before the product?

**Limitations:**

Yes, the authors include a section (Section F in appendix) to address the limitation of the paper.

---

> ### Author Rebuttal · Authors · 2023-08-09
>
> We thank the reviewer 9ERM for the insightful feedback and for appreciating our theoretical analysis of Transformer and its value for the theoretical community. We address the concerns below.
>
> ---
> > **Q1:** [Only the shallow (encoder part of the) Transformer network is considered. ]
>
> **A1:** We agree with the reviewer that our results focus on a relatively simple architecture of Transformers. Nevertheless, even for such a simple architecture, convergence analysis from a theoretical standpoint is still unclear. Our results provide global convergence, as a good starting point, which would pave the way for analyzing practical architectures, as discussed in Appendix C.7.
>
> Besides, in our revised version, we have already included the residual block into our analysis, to capture the fundamental components of Transformer as much as possible, see [general response](https://openreview.net/forum?id=8ZveVHfmIE&noteId=hT7qrLWaaA).
>
> Furthermore, our results on shallow Transformers can still provide some findings and guidelines for practical Transformers, e.g., the impact of initialization schemes, input format, and scaling, see section 4.5 for details.
>
> ---
> > **Q2:** [The comparison between theoretical and empirical convergence. The quantity $\alpha$ in Proposition 1 seems to be sub-optimal. ]
>
> **A2:** We agree with the reviewer that the lower bound $\alpha$ could be loose though the order of $\alpha$ is tight. This is because our derivation involves some constants in the lower bound of $\alpha$. We believe that these constants can be improved under a refined analysis.
>
> According to your suggestions, in Fig.5 of the one-page pdf, we follow the experiment at sec.5.1 and plot the comparison between the empirical convergence and the quality $\left(1-\frac{\gamma \alpha^2}{2}\right)^t \ell (\theta^0)$. We can see that Proposition 1 provides an upper bound for the empirical loss: the order is the same (both of them are straight lines) but the slope is different. A refined analysis (e.g., smaller constant) on the lower bound of $\alpha$ leads to a better estimation.
>
>
> ---
> > **Q3:** [In the proof of Lemma 1, you say by LLN we have the second equality in line (67). But the entries in $\beta'_j \beta_i \frac{1}{d_s}\bf {1}\_{d_s}$ by taking the limit before the product?]
>
> **A3:** The limit is taken after calculating the inner product of the Jacobian, as can be seen in the proof of Lemma 1, e.g., line 787.
>
> ---
>
> We hope that our responses have alrady fixed the issues that the reviewer concerns. We are happy to discuss further questions if any.

---

> > ### Comment · Reviewer_9ERM · 2023-08-13
> >
> > Thank You for Your detailed offical comments and rebuttal. I am positive towards this paper and am happy to keep my rating as 7.

---

### Official Review · Reviewer_n1Tb · 2023-07-07

**Soundness:** 3 good
**Presentation:** 4 excellent
**Contribution:** 3 good
**Rating:** 5
**Confidence:** 2

**Summary:**

This paper proves the global convergence of the shallow transformer under a more realistic setting. While I don't often read theoretical machine learning papers, I found some of the analysis interesting. However, the paper does feel a bit incremental compared to previous work.


**Strengths:**

I would like to commend the authors for their efforts in conducting a more realistic analysis. I also appreciate their inclusion of analysis, discussion, experiments on artificial data, and experiments on real data, which makes the paper more complete.


**Weaknesses:**

The paper does feel a bit incremental at times. It would be helpful to see more discussion of how the analysis could be applied to practical experiments.


**Questions:**

In what way does the proposed analysis different from prior analysis, aparting from a more realistic setting? To achieve the more realist setting, do the authors have to come up with a different analysis technique to do that? It is not clear to me how novel the analysis techniques are.


**Limitations:**

Most of the analysis is done on shallow Transformers as pointed out by the authors. Moreover, the practical implication of the analysis is not yet clear.

---

> ### Author Rebuttal · Authors · 2023-08-09
>
> We thank the reviewer n1Tb for the insightful feedback. We address the concerns below.
>
> ---
>
> > **Q1:** [Most of the analysis is done on shallow Transformer. The practical implication of the analysis is not yet clear.]
>
> **A1:** We agree with the reviewer that this setting is a bit far away from practical Transformers. However, from a theoretical side, even for a shallow Transformer (with a realistic setting), the global convergence analysis is unknown and is still an open question in the deep learning theory community. The contribution on the convergence analysis for Transformer has already been recognized by the rest reviewers.
>
> In our revised version, we have already included the residual block into our analysis, to capture the fundamental components of Transformer as much as possible, see [general response](https://openreview.net/forum?id=8ZveVHfmIE&noteId=hT7qrLWaaA).
> Our theoretical results are still able to provide some guidelines for practical settings:
>
> - Initialization: our theoretical result suggests using He/LeCun initialization instead of NTK initialization because the former initializations can lead to faster convergence.
> - Input format: We show how the data formulation has an impact on the performance of the Transformer. Specifically, given a vector input, if we formulate it along the sequence dimension, we can not guarantee the training converges from theory and it does not converge empirically. If we formulate along the embedding dimension, then the training can converge.
> - Scaling: Our theory identifies why the self-attention layer degenerates as a pooling layer under the $d_m^{-1}$ setting with an infinite-width setting. For small width $d_m$, there is no significant difference for these two scaling schemes on the convergence; But for a large enough $d_m$, then the scaling $d_m^{−1/2}$ admits a faster convergence rate than that of $d_m^{-1}$ because in this case, the self-attention layer degenerates as a pooling layer more or less.
>
> ---
>
> > **Q2:** [In what way does the proposed analysis different from prior analysis, apart from a more realistic setting? To achieve the more realistic setting, do the authors have to come up with a different analysis technique to do that?]
>
> **A2:** We will add a paragraph about the technical difference from prior work in the final version, see the [general response](https://openreview.net/forum?id=8ZveVHfmIE&noteId=hT7qrLWaaA) for details.
>
> ---
>
> We hope that our responses have addressed the concerns of the reviewer. If there are any remaining questions, we are happy to discuss them further.

---

### Official Review · Reviewer_YNcT · 2023-07-07

**Soundness:** 3 good
**Presentation:** 3 good
**Contribution:** 4 excellent
**Rating:** 7
**Confidence:** 4

**Summary:**

The paper presents convergence results for shallow transformer networks for commonly used Gaussian-based initialization schemes in deep learning (LeCun, He) and different scaling regimes under finite overparameterization. In addition, the authors provide limiting neural tangent kernel (NTK) analysis for its minimum Eigenvalue and stability during training. They consider the encoder part of the transformer for the analysis which consists of a single-self attention layer, followed by ReLU, average pooling and a linear layer which maps to a scalar. The analysis is presented for gradient descent (GD) training with square loss applicable to classification/regression, where all parameters apart from the ReLU layer are trainable. The convergence framework for $1\sqrt{d_m}$ scaling uses three assumptions: bounded data, linearly independent tokens in each input example and dissimilarity of two different data examples. The $1/d_m$ scaling regime additionally uses a PDness assumption on the covariance matrix. The paper also provides experiments on synthetic gaussian data and on MNIST data for a Vision Transformer (ViT) tracking convergence under scaling regimes, width and NTK evolution.

**Strengths:**

**Importance and motivation**: Very interesting work, I enjoyed reading it. The paper certainly takes a step and provides novel results towards understanding the training dynamics of shallow transformers and the overparameterization required for global convergence - an important research direction considering their prevalent usage. The authors do a good job in motivating the need to study different scaling regimes in the introduction section as well.

**Organization and writing**: The paper is organized, well-written and easy to follow in general.

**Assumption verification**: I like the important remarks and the experimental evidence authors use to back the assumptions, particularly for assumption 3 where they provide experiments on IMDB and MNIST dataset.

Convergence results discussion in sec 4.5 on slower convergence of $1/d_m$ scaling and NTK initialization being slower than He/LeCun provide good insights into the theoretical results.


**Weaknesses:**

**Missing intuitive explanations**: Standard convex analysis requires PL inequality + smoothness of the loss function to get a linear rate. Per my knowledge, the proof follows a similar(ish) recipe to [1], where they provide linear rates for NNs using the local gradient-Lipschitz property of the loss + a variant of PL, which gives a good analogue to convex analysis (as briefly mentioned already in the related work sec). I think it’d be nice for the reader if the paper contains a longer proof sketch with such intuitive comments/remarks for the proof steps. Some more suggestions on this can be found in the questions section.

**Minor bugs and typos**: There are some minor issues in the proofs and some typos in the paper: please refer to questions.

**Regarding assumption 3**: In its current form, assumption 3 forms a tail bound on the similarity of the respective empirical covariance matrices. This covariance inner product equals similarity of $\mathbf{X}_i, \mathbf{X}_j$ + some cross terms - how well does assumption 3 reflect dissimilarity as mentioned in the paper? It would have been nice had there been plots for similarity of $\mathbf{X}_i, \mathbf{X}_j$ along with the current ones.

**Concerns/questions regarding the results**: I have several queries regarding separability constant $\hat{c}$, lower bound on $N$, etc. Please see the questions section.

**Applicability on contextual token data**: It might be interesting to see if/how the results apply to the commonly used contextual token data setting [2, 3] for theoretical analysis of transformers. Eg: Assm. 2 fails if sparsity [3] isn’t small enough, a small discussion might be something to consider.


**Questions:**

**Some more suggestions on intuitive remarks for proof sketch**:
1. It'd be helpful to mention the condition on gradient that makes the variant of PL and is used to prove convergence.
2. Breaking down (briefly) how the local Lipschitz property of loss is proved.

**Typos/minor bugs**:

1. $ \gamma <=1/C$ in prop. Suggestion: It’d be helpful to specify that $C$ signifies local Lipschitz constant for the gradient.

2. Around line 653 in the appendix: Should it be $\theta^{t+1}, \theta^{t}$ instead of $\mathbf{W}^{t+1}, \mathbf{W}^{t}$? Also, in the same proof, second last step should be $\sigma_{min}^2$ and $||f^t-\mathbf{y}||^2$ - there might be a 2 factor missing in the result as well.

3. Lemma 15 proof, line 661: I don't think $\mathbf{XX}^T$ (correctly mentioned in Assm. 3 - $\mathbf{X}^T\mathbf{X}$) should be called the covariance matrix as the data is placed along rows.

**Results**:


1. For thm. 1 remark: For larger $\hat{c}$, the exp quantity grows - which seems a bit counterintuitive in terms of separability of data.

2. In the $1/d_m$ regime there’s an additional req. On $N$, but not in the $1\sqrt{d_m}$ regime - is there any obvious reasoning for this or is it just a proof artifact

3. Is there any step-size $\gamma$ (or $C$, same thing) difference in $1/d_m$ vs $1/\sqrt{d_m}$? As far as I know, typically in NNs in the latter it is $O(1)$ [5] and $O(d_m)$ in the former [4].

4. It’d be nice to know the authors' thoughts about the tightness on the width condition, especially in the $\sqrt{1/d_m}$ regime.

Limitations section says the result doesn’t cover cross-entropy loss, can such a PL variant + local Lipschitz technique be applied to it? I’ve mostly seen such analysis with square loss [1, 5].


**References**

[1]. Nguyen, Q.N. and Mondelli, M., 2020. Global convergence of deep networks with one wide layer followed by pyramidal topology. Advances in Neural Information Processing Systems, 33, pp.11961-11972.

[2]. Li, H., Wang, M., Liu, S. and Chen, P.Y., 2023. A theoretical understanding of shallow vision transformers: Learning, generalization, and sample complexity. arXiv preprint arXiv:2302.06015.

[3]. Oymak, S., Rawat, A.S., Soltanolkotabi, M. and Thrampoulidis, C., 2023. On the Role of Attention in Prompt-tuning. arXiv preprint arXiv:2306.03435.

[4]. Mousavi-Hosseini, A., Park, S., Girotti, M., Mitliagkas, I. and Erdogdu, M.A., 2022. Neural networks efficiently learn low-dimensional representations with sgd. arXiv preprint arXiv:2209.14863.

[5]. Du, S.S., Zhai, X., Poczos, B. and Singh, A., 2018. Gradient descent provably optimizes over-parameterized neural networks. arXiv preprint arXiv:1810.02054.

**Limitations:**

I couldn’t find the code in the supplementary or any anonymous link to reproduce the results.

---

> ### Author Rebuttal · Authors · 2023-08-09
>
> We thank the reviewer YNcT for the insightful feedback and for appreciating our theoretical analysis of Transformer and its value for the theoretical community. We address the concerns below.
>
> ---
>
> > **Q1:** [Suggestions on intuitive remarks for proof sketch.]
>
> **A1:**  Thanks for the suggestion, we have already discussed about the proof sketch, see the [general response](https://openreview.net/forum?id=8ZveVHfmIE&noteId=hT7qrLWaaA) for details.
> We will add the proof sketch as well as ituitive remarks for better understanding.
>
> ---
>
> > **Q2:** [Typos/minor bugs.]
>
> **A2:** We are thankful to the reviewer for spotting the typos. We have already corrected them in the revised version. For example, in proposition 1, it should be $\gamma \le 1/C$. Around line 653, it should be $\bf \theta^{t+1}$, $\bf \theta^{t}$, $\sigma_{min}^2$, $||f^t-\mathbf{y}||^2$, and a factor of 2. In line 661, we will remove the phase ‘covariance matrix’ to avoid confusion.
>
> ---
>
> > **Q3:** [Assumption 3.]
>
> **A3:**  We plot the figure for $<X_i, X_j>$ in the same MNIST dataset as shown in **Fig.1** in the one-page pdf.
> We can see a similar exponential decay for the inner product of two data points.
>
> ---
>
> > **Q4:** [For thm. 1 remark: For larger \hat{c}, the exp quantity grows - which seems a bit counterintuitive in terms of separability of data.]
>
> **A4:** As $\hat{c}$ increases, the probability in theorem 1 decreases. Meanwhile, from theorem 3, we can see that a larger $\hat{c}$ results in less separable data (line 199 should be corrected). It makes sense as less separable data leads to a lower probability.
>
> Sorry for the confusion because of the typo in line 199 and we have already changed it to "A larger $\hat{c}$ results in less separable data" in the revised version.
>
> ---
>
> > **Q5:** [Applicability on contextual token data.]
>
> **A5:** We are thankful to the reviewer for pointing out these interesting works on the analysis of contextual token data and we believe it would be of great interest to generalize our result to the classification task with contextual token data. We have already included a discussion in our revised version.
>
> When extending our current analysis to contextual token data, we need a refined analysis framework to distinguish that label-relevant (irrelevant) token. Besides, Assumption 2 can be slightly relaxed to the case that $\lambda_{min}(X_k X_k^{T}) > 0$ for any $k$ holds with a high probability. Nevertheless, a new proof framework on the minimum eigenvalue under contextual data is required if we want to get rid of Assumption 2 and its variants.
>
> ---
>
> > **Q6:** [In the regime $d_m^{-1}$ there’s an additional req. On $N$, but not in the regime $d_m^{-1/2}$- is there any obvious reasoning for this or is it just a proof artifact.]
>
> **A6:** This is a proof artifact because the requirement $N > \Omega(d^4)$ is required to provide a lower for the minimum eigenvalue of a specific matrix in the case of  $d_m^{-1}$.
>
> To be specific, we need this assumption to ensure $\lambda\_{\min} ( {\boldsymbol{Z^*}}^\top \boldsymbol{Z^*} )   \ge N/d -9N^{2/3}d^{1/3} = \Theta (N/d)$.
>
> ---
>
> > **Q7:** [Is there any step-size (or $C$ , same thing) difference in $d_m^{-1}$ vs $d_m^{-1/2}$?]
>
> **A7:** Yes, these two different scaling schemes have different $c_1$, $c_2$, $c_3$, see line 652. This leads to different C. We will make it clear in our final version.
>
> ---
>
> > **Q8:** [It’d be nice to know the authors' thoughts about the tightness on the width condition, especially in the regime $d_m^{-1/2}$.]
>
> **A8:** In the regime $d_m^{-1/2}$, the lower bound of the minimum eigenvalue $\lambda_0$ is in the constant order, which is tight.
> Based on this, by studying the relationship between $d_m$ and $\lambda_0$, see Eq. (48) and (49) for details, we can prove that quadratic over-parameterization is required.
>
> This quadratic over-parameterization requirement could be relaxed if a better relationship is given. However this is unclear beyond our proof technique and we will add a remark for this in the revised version.
>
> ---
>
> > **Q9:** [Limitations section says the result doesn’t cover cross-entropy loss, can such a PL variant + local Lipschitz technique be applied to it? I’ve mostly seen such analysis with square loss.]
>
> **A9:** It appears feasible to extend our current analysis from the squared loss to the cross-entropy (CE) loss. The proof idea is still based on the PL variant + local Lipschitz technique but an extra data distance assumption is needed to ensure the global convergence, refer to assumption 4.2 in [2] and assumption 2.1 in [3] for details.
>
> Note that under the CE loss, [2] and [3] deal with fully-connected networks with much stronger over-parameterization than that of the squared loss setting, therefore we think the CE loss requires a stronger quadratic over-parameterization condition for Transformer. We leave this as future work.
>
> ---
>
> > **Q10:** [Code]
>
> **A10:** According to the rebuttal rule, we have sent the code to AC.
>
> ---
>
> ### References
> [1] Nguyen, et al. "Global convergence of deep networks with one wide layer followed by pyramidal topology." NeurIPS, 2020.
>
> [2] Zou, et al. "Gradient descent optimizes over-parameterized deep ReLU networks." Machine learning, 2020
>
> [3] Allen-Zhu, et al, "A convergence theory for deep learning via over-parameterization." ICML, 2019.

---

> > ### Comment · Reviewer_YNcT · 2023-08-21
> >
> > Thank you to the authors for the detailed response to my comments, suggestions and the rebuttal. I am happy to keep my score of 7.

---

### Author Rebuttal · Authors · 2023-08-09

**General Response**

Dear reviewers,

We appreciate your insightful comments. Below, we address three core topics raised by reviewers and then answer individually to the questions of each reviewer.

Concretely, below we make the following responses:
- discuss the extension to the residual Transformer, as pointed out by reviewers sZxG and UvAq
- highlight the technique difficulty when compared to the prior analysis, as requested by reviewers n1Tb and UvAq
- add the proof sketch for better understanding, following the suggestion of reviewers YNcT and sZxG

---

> **Q1:** Extension to residual Transformer (sZxG and UvAq)

**A1:** Our proof framework is general and can handle the following residual Transformer. Here we give a proof sketch to show how to achieve this.

We consider the residual block in the self-attention layer: $ \bf{A}_1= \text{Self-attention}(\bf{X}) \triangleq \sigma_s \left( \tau_0 (\bf{X}{\bf{W}}_Q^\top) \left(\bf{X} \bf{W}_K ^\top\right)^\top \right) \left( \bf{X} \bf{W}_V^\top \right) + \bf{X}$,

which leads to the output $\bf{f}(\bf X)$ in line 628 as $\bf{f}(\bf X)= \tau\_1 \bf{w}\_O^\top \sum\_{i=1}^{d\_s} \sigma\_r
\left(\bf{W}\_V\bf{X}^\top \bf{\beta}\_i + {(\bf{X}^{(i:)})}^\top \right)$.

To prove the convergence of the above residual Transformer, we need:

- To prove the optimization properties of the loss function (e.g., almost smooth), the proof is nearly the same as the current version apart from the additional term to upper bound $||{\bf{X}^{(i:)}}||_2$ in Lemma 9.
- To prove the lower bound of $\alpha_0$, we need to estimate the output difference on the shallow Transformer with/without residual block in Lemma 15. Using the Lipschitz continuity of softmax, ReLU, this difference can be well controlled, and thus the lower bound of $\alpha_0$ can be still achieved with an extra constant.

In the final version, we will detail this convergence result after Sec.C.7 in the appendix. We hope our analysis will lay a foundation for the analysis of more general Transformer architectures.


---

> **Q2:** Technical difficulty (n1Tb and UvAq).

**A2:** Handling the softmax function and the $d_m^{-1}, d_m^{-1/2}$ scaling schemes beyond NTK initialization in Transformer are precisely **the two challenges**. Previous convergence analysis, e.g., [1-2],  cannot be applied to our setting because of the following two issues:
 - Different from classical activation functions, e.g., ReLU, in softmax each element of the output depends on all input. To tackle the interplay between dimensions, we build the connection between $\Phi \Phi^{T}$ and $XX^T$ (see Lemma 15-17) to disentangle the nonlinear softmax function, where $\Phi$ contains information on the output of the softmax. By doing so, the lower bound of the minimum eigenvalue of  $\Phi \Phi^{T}$  can be well controlled by $XX^T$ and the output of softmax. Accordingly, a lower bound on the minimum singular value of ${F}\_{pre}$ in Proposition 1 can be obtained.
- Regarding different initializations and scaling, previous NTK-based analysis is only valid under the $d_m^{-1}$ setting (the softmax degenerates to an all-one vector) but is inapplicable to the realistic $d_m^{-1/2}$ setting, as discussed in the introduction. To tackle this issue, we analyze the input/output of softmax under LeCun/He initialization (see Lemma 15) and identify the optimization properties of the loss function (see the proof of Proposition 1) for global convergence under the finite-width setting.

We will include this discussion in our final version and hope this clarification would address the reviewers’ concerns.

---

> **Q3:** Proof sketch (YNcT and sZxG).

**A3:** Here we present the proof sketch of our global convergence results under various initialization schemes.

[**Proof sketch**] The main idea of our convergence analysis is based on the variant of Polyak-Lojasiewicz (PL) inequality, i.e., (under simple calculation)

$|| \nabla \ell({\theta}) ||\_2^2 \ge 2 \lambda\_{\min}({K}) \ell({\theta}) \ge 2 \lambda\_{\min}({F}\_{pre} {F}_{pre}^{\top}) \ell({\theta})$.

 Thus if the minimum singular value of ${F}_{pre}$ is strictly greater than 0, then minimizing the gradient on the LHS will drive the loss to zero.

To this end, we take Proposition 1 as an example. It can be split into two parts:
- By induction, at every time step, each parameter in Transformer can be bounded w.h.p;  the minimum singular value of ${F}\_{pre}$ is bounded away for some positive quality at the initialization point.
- We prove that the Lipschitzness of the network gradient (Lemma 14), which means the loss function is almost smooth.
By doing so, combining the above two results, the loss function global convergence can be achieved.

Furthermore, in Theorem 1 and Theorem 2, we consider several particular initializations and scaling and aim to provide a positive lower bound for ${F_{pre}}$, satisfying the conditions in Proposition 1.

We will include it in our revised version for better understanding.

---

### References
[1] Du, et al. "Gradient Descent Provably Optimizes Over-parameterized Neural Networks." ICLR, 2018.

[2] Nguyen, et al. "On the proof of global convergence of gradient descent for deep relu networks with linear widths." ICML, 2021.

---

### Decision · Program_Chairs · 2023-09-21

**Decision:**

Accept (poster)

**Comment:**

The paper derives convergence guarantees for a shallow Transformer network (only the encoder part) under different weight initializations/scalings. The analysis relies on the NTK matrix of Transformers derived in prior work and it derives a bound on the minimum eigenvalue of the NTK, similar to the work of Nguyen and Mondelli, 2020. The contribution of this work is mostly theoretical and might not have important practical consequences but I think it's a welcomed contribution for the community.

The reviewers are somewhat positive about the paper. Some concerns have been addressed in the rebuttal, leading some reviewers to increase their scores. However, the paper needs several improvements including adding more intuitive explanations of the analysis and emphasizing more the limitations. For instance, the analysis applies to a shallow encoder only (several theoretical works have studied similar models, it might be worth extending the prior work on this as well). The reviewers also made several criticisms regarding the experimental part. Finally, the title is in my view too general and should be updated to more closely reflect the setting addressed in the paper.

Overall, the paper has some merits so I recommend acceptance, but I strongly encourage the authors to make a somewhat major revision of the paper to address the problems raised by the reviewers. This will undoubtedly increase the impact of the paper so that's also in the interest of the authors.